# Errors-in-variables Fréchet Regression with Low-rank Covariate Approximation

**Dogyoon Song**[*]
Department of EECS
University of Michigan
Ann Arbor, MI 48109, USA
`dogyoons@umich.edu`

**Kyunghee Han**[*]
Department of Math, Stat and Comp Sci
University of Illinois at Chicago
Chicago, IL 60607, USA
`hankh@uic.edu`

## Abstract

Fréchet regression has emerged as a promising approach for regression analysis involving non-Euclidean response variables. However, its practical applicability has been hindered by its reliance on ideal scenarios with abundant and noiseless covariate data. In this paper, we present a novel estimation method that tackles these limitations by leveraging the low-rank structure inherent in the covariate matrix. Our proposed framework combines the concepts of global Fréchet regression and principal component regression, aiming to improve the efficiency and accuracy of the regression estimator. By incorporating the low-rank structure, our method enables more effective modeling and estimation, particularly in high-dimensional and errors-in-variables regression settings. We provide a theoretical analysis of the proposed estimator's large-sample properties, including a comprehensive rate analysis of bias, variance, and additional variations due to measurement errors. Furthermore, our numerical experiments provide empirical evidence that supports the theoretical findings, demonstrating the superior performance of our approach. Overall, this work introduces a promising framework for regression analysis of non-Euclidean variables, effectively addressing the challenges associated with limited and noisy covariate data, with potential applications in diverse fields.

## 1 Introduction

Regression analysis is a fundamental statistical methodology to model the relationship between response variables and explanatory variables (covariates). Linear regression, for example, models the (conditional) expected value of the response variable as a linear function of covariates. Regression models enable researchers and analysts to make predictions, gain insights into how input variables influence the outcomes of interest, and validate hypothetical associations between variables in inferential studies. As a result, regression is widely utilized across various scientific domains, including economics, psychology, biology, and engineering [53, 21, 29].

In recent decades, there has been a growing interest in developing statistical methods capable of handling random objects in non-Euclidean spaces. Examples of these include functional data analysis [41], statistical manifold learning [31], statistical network analysis [34], and object-oriented data analysis [39]. In such contexts, the response variable is defined in a metric space that may lack an algebraic structure, making it challenging to apply global, parametric approaches toward regression as in the classical Euclidean setting. To overcome this challenge, (global) Fréchet regression, which models the relationship by fitting the (conditional) barycenters of the responses as a function of covariates, has been introduced [40]. Notably, when the Euclidean metric is considered, Fréchet

---

[*]Equal contribution. To whom correspondence should be addressed: `hankh@uic.edu`

regression recovers classical Euclidean regression models. For more details on Fréchet regression and its recent developments, we refer readers to [30, 40, 22, 45, 27].

Nevertheless, most existing research on Fréchet regression has focused on ideal scenarios characterized by abundant covariate data that are accurately measured and free of noise. In practical applications, however, high-dimensional data often arise, which are also susceptible to measurement errors and other forms of contamination. These errors can stem from various sources, such as unreliable data collection methods (*e.g.*, low-resolution probes, subjective self-reports) or imperfect data storage and transmission. The high-dimensionality and the presence of measurement errors in covariates pose critical challenges for statistical inference, as regression analysis based on error-prone covariates may result in incorrect associations between variables, yielding misleading conclusions.

To address these limitations, it is crucial to extend the methodology and analysis of Fréchet regression to tackle high-dimensional errors-in-variables problems. In this work, we aim to leverage the low-rank structure in the covariates to enhance the estimation accuracy and computational efficiency of Fréchet regression. Specifically, we explore the extension of principal component regression to handle errors-in-variables regression problems with non-Euclidean response variables.

## 1.1   Contributions

This paper contributes to advancing the (global) Fréchet regression of non-Euclidean response variables, with a particular focus on high-dimensional, errors-in-variables regression.

Firstly, we propose a novel framework, called the *regularized (global) Fréchet regression* (Section 3) that combines the ideas from Fréchet regression [40] and the principal component regression [32]. This framework effectively utilizes the low-rank structure in the matrix of (Euclidean) covariates by extracting its principal components via low-rank matrix approximation. Our proposed method is straightforward to implement, not requiring any knowledge about the error-generating mechanism.

Furthermore, we provide a comprehensive theoretical analysis (Section 4) in three main theorems to establish the effectiveness of the proposed framework. Firstly, we prove the consistency of the proposed estimator for the true global Fréchet regression model (Theorem 1). Secondly, we investigate the convergence rate of the estimator's bias and variance (Theorem 2). Lastly, we derive an upper bound for the distance between the estimates obtained using error-free covariates and those with errors-in-variables covariates (Theorem 3). Collectively, these results demonstrate that our approach effectively addresses model mis-specification and achieves more efficient model estimation by leveraging the low-rank structure of covariates, despite the presence of inherent bias due to unobserved measurement errors.

To validate our theoretical findings, we conduct numerical experiments on synthetic datasets (Section 5). We observe that the proposed method provides more accurate estimates of the regression parameters, especially in high-dimensional settings. Our experimental results emphasize the importance of incorporating the low-rank structure of covariates in Fréchet regression, and provide empirical evidence that aligns with our theoretical analysis.

## 1.2   Related work

**Metric-space-valued variables.** Nonparametric regression models for Riemannian-manifold-valued responses were proposed as a generalization of regression for multivariate outputs by Steinke *et al.* [48, 49]. These works provided a foundation for recent developments in regression analysis of non-Euclidean responses. Later, Hein [30] proposed a Nadaraya-Watson-type kernel estimation of regression model for general metric-space-valued outcomes. Since then, statistical properties of regression models for some special classes of metric-space-valued outcomes, such as distribution functions [23, 52, 28] and matrix-valued responses [57, 20], have been investigated. Recently, many researchers have introduced further advances in Fréchet regression, including [40, 10, 37, 45]. In this study, we use the global Fréchet regression proposed in [40] as the basis for our proposed method.

**Errors-in-variables regression.** Much of earlier work on errors-in-variables (EIV) problems in the statistical literature can be found in [13], which covers the simulation-extrapolation (SIMEX) [16, 11], the attenuation correction method [36], covariate-adjusted model [46, 19], and the deconvolution kernel method [25, 24, 18]. The regression calibration method [47], instrumental variable modeling [12, 43], and the two-phase study design [9, 4] were also proposed when additional data are available

for correcting measurement errors. In the high-dimensional modeling literature, regularization methods for recovering the true covariate structure can also be utilized [38, 7, 17]. Despite a diverse body of literature on high-dimensional learning and robust regression modeling, much of it assumes response spaces to be vector spaces endowed with inner products. In this paper, we tackle EIV problems within the Fréchet regression framework. While previous works have explored regression analysis in non-Euclidean metric spaces, addressing EIV issues in this context remains uncharted.

**Principal component regression.** The principal component regression (PCR) [32] is a statistical technique that regresses response variables on principal component scores of the covariate matrix. The conventional PCR selects a few principal components as the "new" regressors associated with the first leading eigenvalues to explain the highest proportion of variations observed in the original covariate matrix. In functional data analysis, PCR is known to have a shrinkage effect on the model estimate and produce robust prediction performance in functional regression [42, 33]. Recently, Agarwal *et al.* [2] investigated the robustness of PCR in the presence of measurement errors on covariates and the statistical guarantees for learning a good predictive model. Unlike prior statistical analyses of EIV problems that often assume known or estimable noise distributions, PCR leverages inherent low-rank structures in the covariates without requiring a priori knowledge of measurement error distributions. We adopt PCR as a concrete, practical solution to EIV models in non-Euclidean regression, driven by two compelling considerations. Firstly, the prevalence of (approximate) low-rank structures in real-world datasets enhances the practical relevance of our approach. Secondly, we intentionally opt for an approach with minimal assumptions regarding covariate errors to ensure broad applicability.

## 1.3 Organization

In Section 2, we introduce the notation used throughout the paper, and overview the global Fréchet regression framework. Section 3 presents the problem setup, objectives, and our proposed estimator, which we refer to as the regularized Fréchet regression (Definition 4). In Section 4, we discuss theoretical guarantees on the regularized Fréchet regression method in accurately estimating the global Fréchet regression function. Section 5 presents the results of numerical "proof-of-concept" experiments that support the theoretical findings. Finally, we conclude this paper with discussions in Section 6. Due to space constraints, detailed proofs of the theorems as well as additional details and discussions of experiments are provided in the Appendix.

# 2 Preliminaries

## 2.1 Notation

Let $\mathbb{N}$ denote the set of positive integers and $\mathbb{R}$ denote the set of real numbers. Also, let $\mathbb{R}_+ := \{x \in \mathbb{R} : x \geq 0\}$. For $n \in \mathbb{N}$, we let $[n] := \{1, \ldots, n\}$. We mostly use plain letters to denote scalars, vectors, and random variables, but we also use boldface uppercase letters for matrices, and curly letters to denote sets when useful. Note that we may identify a vector with its column matrix representation. For a matrix $\boldsymbol{X}$, we let $\boldsymbol{X}^{-1}$ denote its inverse (if exists) and $\boldsymbol{X}^\dagger$ denote the Moore-Penrose pseudoinverse of $\boldsymbol{X}$. Also, we let $\mathrm{rowsp}\,(\boldsymbol{X})$ and $\mathrm{colsp}\,(\boldsymbol{X})$ denote the row and column spaces of $\boldsymbol{X}$, respectively. Furthermore, we let $\mathrm{spec}\,(\boldsymbol{X})$ denote the set of non-zero singular values of $\boldsymbol{X}$, $\sigma_i(\boldsymbol{X})$ denote the $i$-th largest singular value of $\boldsymbol{X}$, and $\sigma^{(\lambda)}(\boldsymbol{X}) := \inf\{\sigma_i(\boldsymbol{X}) > \lambda : i \in \mathbb{N}\}$ with the convention $\inf \emptyset = \infty$. We let $\boldsymbol{1}_n = (1, 1, \ldots, 1)^\top \in \mathbb{R}^d$ and let $\mathbb{1}$ denote the indicator function. We let $\|\cdot\|$ denote a norm, and set $\|\cdot\| = \|\cdot\|_2$ (the $\ell_2$-norm for vectors, and the spectral norm for matrices) by default unless stated otherwise. For a finite set $\mathcal{D}$, we may identify $\mathcal{D}$ with its empirical measure $\nu_{\mathcal{D}} = \frac{1}{|\mathcal{D}|} \sum_{x \in \mathcal{D}} \delta_x$, where $\delta_x$ denotes the Dirac measure supported on $\{x\}$.

Letting $f, g : \mathbb{R} \to \mathbb{R}$, we write $f(x) = O(g(x))$ as $x \to \infty$ if there exist $M > 0$ and $x_0 > 0$ such that $|f(x)| \leq M \cdot g(x)$ for all $x \geq x_0$. Likewise, we write $f(x) = \Omega(g(x))$ if $g(x) = O(f(x))$. Furthermore, we write $f(x) = o(g(x))$ as $x \to \infty$ if $\lim_{x \to \infty} \frac{f(x)}{g(x)} = 0$. For a sequence of random variables $X_n$, and a sequence $a_n$, we write $X_n = O_p(a_n)$ as $n \to \infty$ if for any $\varepsilon > 0$, there exists $M \in \mathbb{R}_+$ and $N \in \mathbb{N}$ such that $P\left(\left|\frac{X_n}{a_n}\right| > M\right) < \varepsilon$ for all $n \geq N$. Similarly, we write $X_n = o_p(a_n)$ if $\lim_{n \to \infty} P\left(\left|\frac{X_n}{a_n}\right| > \varepsilon\right) = 0$ for all $\varepsilon > 0$.

## 2.2 Global Fréchet regression

Let $(X, Y)$ be a random variable that has a joint distribution $P_{X,Y}$ supported on $\mathbb{R}^p \times \mathcal{M}$, where $\mathbb{R}^p$ is the $p$-dimensional Euclidean space and $\mathcal{M} = (\mathcal{M}, d)$ is a metric space equipped with a distance function $d : \mathcal{M} \times \mathcal{M} \to \mathbb{R}$. We write the marginal distribution of $X$ as $P_X$, and the conditional distribution of $Y$ given $X$ as $P_{Y|X}$.

**Definition 1** (Fréchet regression function). *Let $(X, Y)$ be a random element that takes value in $\mathbb{R}^p \times \mathcal{M}$. The* Fréchet regression function *of $Y$ on $X$ is a function $\varphi^* : \mathbb{R}^p \to \mathcal{M}$ such that*

$$\varphi^*(x) = \arg\min_{y \in \mathcal{M}} \mathbb{E}\big[d^2(Y, y) \,|\, X = x\big], \qquad \forall x \in \operatorname{supp} P_X \subseteq \mathbb{R}^p. \tag{1}$$

We note that $\varphi^*(x)$ is the best predictor of $Y$ given $X = x$, as it minimizes the marginal risk $\mathbb{E}\big[d^2(Y, \varphi^*(X))\big]$ under the squared-distance loss. In the literature, $\varphi^*(x)$ is also known as the conditional Fréchet mean of $Y$ given $X = x$ [26]. It is important to recognize that the existence and uniqueness of the Fréchet regression function are closely tied to the geometric characteristics of $\mathcal{M}$, and are not guaranteed in general [3, 8]. Nonetheless, extensive research has been conducted on the existence and uniqueness of Fréchet means in various metric spaces commonly encountered in practical applications. Examples include the unit circle in $\mathbb{R}^2$ [14], Riemannian manifolds [1, 5], Alexandrov spaces with non-positive curvature [51], metric spaces with upper bounded curvature [58], and Wasserstein space [59, 35].

While modeling and estimating the Fréchet regression function $\varphi^*$ is often of interest, its global (parametric) modeling may not be straightforward, especially when $\mathcal{M}$ lacks a useful algebraic structure, such as an inner product. For instance, in classical linear regression analysis with $\mathcal{M} = \mathbb{R}$, the conditional distribution of $Y$ given $X = x$ is normally distributed with a mean of $\varphi^*(x) = \alpha + \beta^\top x$ and a fixed variance $\sigma^2$, where $\alpha$ and $\beta$ represent the regression coefficients. Similarly, when $\mathcal{M}$ possesses a linear-algebraic structure, one can specify a class of regression functions that quantifies the association between the expected outcome and covariates in an additive and multiplicative manner. However, the lack of an algebraic structure in general metric spaces may prevent us from characterizing the barycenter $\varphi^*(x)$ in the same way classical regression analysis determines the expected value of outcomes with changing covariates.

To address this challenge, Petersen and Müller [40] recently proposed to exploit algebraic structures in the space of covariates, $\mathbb{R}^p$, instead of $\mathcal{M}$. Specifically, they consider a weighted Fréchet mean as

$$\varphi(x) = \arg\min_{y \in \mathcal{M}} \mathbb{E}\big[w(X, x) \cdot d^2(Y, y)\big], \tag{2}$$

where $w : \mathbb{R}^p \times \mathbb{R}^p \to \mathbb{R}$ is a weight function such that $w(\xi, x)$ denotes the influence of $\xi$ at $x$. In particular, Petersen and Müller [40] defined the global Fréchet regression function with a specific choice of $w$ as follows.

**Definition 2** (Global Fréchet regression function). *Let $(X, Y)$ be a random variable in $\mathbb{R}^p \times \mathcal{M}$. Let $\mu = \mathbb{E}(X)$ and $\Sigma = \operatorname{Var}(X)$. The* global Fréchet regression function *of $Y$ on $X$ is a function $\varphi_{\mathrm{glo}} : \mathbb{R}^p \to \mathcal{M}$ such that*

$$\varphi_{\mathrm{glo}}(x) = \arg\min_{y \in \mathcal{M}} \mathbb{E}\big[w_{\mathrm{glo}}(X, x) \cdot d^2(Y, y)\big] \tag{3}$$

*where $w_{\mathrm{glo}}(X, x) = 1 + (X - \mu)^\top \Sigma^{-1}(x - \mu)$.*

When $\mathcal{M}$ is an inner product space (*e.g.*, $\mathcal{M} = \mathbb{R}$), the function $\varphi_{\mathrm{glo}}$ restores the standard linear regression model representation over the domain $\mathbb{R}^p$. For this reason, $\varphi_{\mathrm{glo}}$ is commonly referred to as the *global Fréchet regression model* for metric-space-valued outcomes [40, 37, 54].

**Remark on Definition 2** One might wonder why the term "global" is used to describe $\varphi_{\mathrm{glo}}$ as a Fréchet regression function. The use of the adjective "global" serves to emphasize its distinction from "local" nonparametric regression methods that interpolate data points. Notably, when $\mathcal{M}$ is a Hilbert space, $\varphi_{\mathrm{glo}}$ reduces to the natural linear models. For instance, if $\mathcal{M} = \mathbb{R}$, then it follows that $\varphi_{\mathrm{glo}}(x) = \mathbb{E}\big[w_{\mathrm{glo}}(X, x) \cdot Y\big] = \alpha + \beta^\top(x - \mu)$, where $\alpha = \mathbb{E}[Y]$ and $\beta = \Sigma^{-1} \cdot \mathbb{E}\big[(X - \mu) \cdot Y\big]$. These linear models hold uniformly for the evaluation point $x$. Similarly, in the case of an $L^2$ space equipped with the squared-distance metric $d^2(y, y') = \|y - y'\|_2^2$ induced by the $L^2$ norm, $\varphi_{\mathrm{glo}}$ represents the linear regression model for functional responses. Thus, $\varphi_{\mathrm{glo}}$ establishes a globally defined model that spans the entire space.

# 3 Problem and methodology

## 3.1 Problem formulation

Let $(X, Y)$ be a random variable in $\mathbb{R}^p \times \mathcal{M}$ and $P_{X,Y}$ be their joint distribution. Let $\mathcal{D}_n = \{(X_i, Y_i) : i \in [n]\}$ be an independent and identically distributed (IID) sample drawn from $P_{X,Y}$. Note that we may identify the set $\mathcal{D}_n$ with its discrete measure (empirical distribution), cf. Section 2.1. We consider the problem of estimating the global Fréchet regression function $\varphi_{\text{glo}}$ (see Definition 2) from data $\mathcal{D}_n$. In this setting, a natural estimator of $\varphi_{\text{glo}}$ would be its sample-analogue estimator. With $\widehat{\mu}_{\mathcal{D}_n} = \mathbb{E}_{(X,Y)\sim\mathcal{D}_n}(X) = \frac{1}{n}\sum_{i=1}^n X_i$ and $\widehat{\Sigma}_{\mathcal{D}_n} = \mathrm{Var}_{(X,Y)\sim\mathcal{D}_n}(X) = \frac{1}{n}\sum_{i=1}^n (X_i - \widehat{\mu}_{\mathcal{D}_n}) \cdot (X_i - \widehat{\mu}_{\mathcal{D}_n})^\top$, the sample-analogue estimator $\widehat{\varphi}_{\mathcal{D}_n}$ is defined as

$$\widehat{\varphi}_{\mathcal{D}_n}(x) = \arg\min_{y\in\mathcal{M}} \left\{ \frac{1}{n} \sum_{(X_i,Y_i)\in\mathcal{D}_n} \widehat{w}_{\mathcal{D}_n}(X_i, x) \cdot d^2(Y_i, y)] \right\} \tag{4}$$

where $\widehat{w}_{\mathcal{D}_n}(X, x) = 1 + (X - \widehat{\mu}_{\mathcal{D}_n})^\top \widehat{\Sigma}_{\mathcal{D}_n}^{-1}(x - \widehat{\mu}_{\mathcal{D}_n})$. The statistical properties of $\widehat{\varphi}_{\mathcal{D}_n}$, including the asymptotic distribution, a ridge-type variable selection operation, and total variation regularization method have been investigated [40, 37, 54].

In practice, however, we may only be able to access $\widetilde{\mathcal{D}}_n = \{(Z_i, Y_i) : i \in [n]\}$ instead of $\mathcal{D}_n$, where

$$Z_i = X_i + \varepsilon_i, \qquad i = 1, \ldots, n \tag{5}$$

denotes an error-prone observation of the covariates $X$ by measurement error $\varepsilon$. This formulation corresponds to the classical errors-in-variables problem.

**Objective.** Given a dataset, either $\mathcal{D}_n$ or $\widetilde{\mathcal{D}}_n$, our aim is to produce an estimate $\widehat{\varphi}$ of the global Fréchet regression function $\varphi_{\text{glo}}$ so that the prediction error is minimized. Specifically, we evaluate the performance of $\widehat{\varphi}$ by means of the distance in the response space, $d\big(\widehat{\varphi}(x), \varphi_{\text{glo}}(x)\big)$.

## 3.2 Fréchet regression with covariate principal components

**Singular value thresholding.** Among various low-rank matrix approximation methods, we consider the (hard) singular value thresholding (SVT). For any $\lambda \in \mathbb{R}_+$, we define the map $\mathtt{SVT}^{(\lambda)} : \mathbb{R}^{n\times p} \to \mathbb{R}^{n\times p}$ that removes all singular values that are less than the threshold $\lambda$. To be precise, $\mathtt{SVT}^{(\lambda)}$ can be expressed in terms of the singular value decomposition (SVD) as follows:

$$\boldsymbol{M} = \sum_{i=1}^{\min\{n,p\}} s_i \cdot u_i v_i^\top \text{ is a SVD} \quad \Longrightarrow \quad \mathtt{SVT}^{(\lambda)}(\boldsymbol{M}) = \sum_{i=1}^{\min\{n,p\}} s_i \cdot \mathbb{1}\{s_i > \lambda\} \cdot u_i v_i^\top. \tag{6}$$

**Regularized Fréchet regression.** We introduce a variant of the sample-analog estimator of the global Fréchet regression function based on principal components of the sample covariance. To facilitate the description of our proposed estimator, we introduce additional notation here.

**Definition 3** (Covariate mean/covariance). *For a probability distribution $\nu$ on $\mathbb{R}^p \times \mathcal{M}$, the* covariate mean *and* covariate covariance *with respect to $\nu$ are respectively defined as*

$$\mu_\nu := \mathbb{E}_{(X,Y)\sim\nu}(X) \qquad \text{and} \qquad \Sigma_\nu := \mathrm{Var}_{(X,Y)\sim\nu}(X). \tag{7}$$

Recall that a finite set $\mathcal{D} \subset \mathbb{R}^p \times \mathcal{M}$ may be identified with its empirical distribution; it follows that

$$\mu_{\mathcal{D}} = \frac{1}{|\mathcal{D}|} \sum_{(x_i,y_i)\in\mathcal{D}} x_i \qquad \text{and} \qquad \Sigma_{\mathcal{D}} = \frac{1}{|\mathcal{D}|} \sum_{(x_i,y_i)\in\mathcal{D}} (x_i - \mu_{\mathcal{D}}) \cdot (x_i - \mu_{\mathcal{D}})^\top. \tag{8}$$

**Definition 4** (Regularized Fréchet regression). *Let $\nu$ be a probability distribution on $\mathbb{R}^p \times \mathcal{M}$ and $\lambda \in \mathbb{R}_+$. The $\lambda$-regularized Fréchet regression function for $\nu$ is a map $\varphi_\nu^{(\lambda)} : \mathbb{R}^p \to \mathcal{M}$ such that*

$$\varphi_\nu^{(\lambda)}(x) = \arg\min_{y\in\mathcal{M}} R_\nu^{(\lambda)}(y; x), \quad \text{where} \quad R_\nu^{(\lambda)}(y; x) = \mathbb{E}_{(X,Y)\sim\nu}\left[w_\nu^{(\lambda)}(X, x) \cdot d^2(Y, y)\right]$$

$$\text{and} \quad w_\nu^{(\lambda)}(x', x) = 1 + (x' - \mu_\nu)^\top \left[\mathtt{SVT}^{(\lambda)}(\Sigma_\nu)\right]^\dagger (x - \mu_\nu). \tag{9}$$

When $\mathcal{D}_n = \{(X_i, Y_i) \in \mathbb{R}^p \times \mathcal{M} : i \in [n]\}$ is an IID sample from $P_{X,Y}$, the $\lambda$-regularized estimator $\varphi_{\mathcal{D}_n}^{(\lambda)}$ subsumes the sample-analogue estimator $\widehat{\varphi}_{\mathcal{D}_n}$ in (4) as a special case where $\lambda = 0$.

**Connection to principal component regression.** Here we remark that when $\mathcal{M}$ is a Euclidean space, the regularized Fréchet regression function $\varphi_\nu^{(\lambda)}$ effectively reduces to the principal component regression. Suppose that $\mathcal{M} = \mathbb{R}$ and $\mathcal{D}_n = \{(x_i, y_i) \in \mathbb{R}^p \times \mathbb{R} : i \in [n]\}$ is a given dataset. Then $\varphi_{\mathcal{D}_n}^{(\lambda)}(x) = \overline{y} + \hat{\beta}_\lambda^\top (x - \mu_{\mathcal{D}_n})$ where $\overline{y} = \frac{1}{n}\sum_{i=1}^n y_i$ and $\hat{\beta}_\lambda = [\mathtt{SVT}^{(\lambda)}(\Sigma_{\mathcal{D}_n})]^\dagger \cdot [\frac{1}{n}\sum_{i=1}^n (x_i - \mu_{\mathcal{D}_n}) \cdot (y_i - \overline{y})]$. Observe that $\hat{\beta}_\lambda$ is exactly the regression coefficient of principal component regression applied to the centered dataset $\mathcal{D}_n^{\mathrm{ctr}} = \{(x_i - \mu_{\mathcal{D}_n}, y_i - \overline{y}) : i \in [n]\}$ using $k$ principal components with $k = \max_{a \in [p]} \{\sigma_a(\Sigma_{\mathcal{D}_n^{\mathrm{ctr}}}) \geq \lambda\}$.

# 4  Main results

In this section, we investigate properties of $\varphi_\nu^{(\lambda)}$ for $\lambda \geq 0$, with a focus on two cases: $\nu = \mathcal{D}_n$ and $\nu = \widetilde{\mathcal{D}}_n$, cf. Section 3.1. By denoting the true distribution that generates $(X, Y)$ as $\nu^*$, we can express $\varphi_{\mathrm{glo}}$ as $\varphi_{\nu^*}^{(0)}$. To analyze the discrepancy between the estimator $\varphi_\nu^{(\lambda)}(x)$ and $\varphi_{\mathrm{glo}}(x)$, we examine the relationships depicted in the schematic in Figure 1. Our theoretical findings can be summarized as follows: Even in the presence of covariate noises, $\varphi_{\widetilde{\mathcal{D}}_n}^{(\lambda)}$ with a suitable $\lambda > 0$ can effectively eliminate the noise in $Z$ to estimate $X$, thereby reducing the error in estimating $\varphi_{\mathrm{glo}}$.

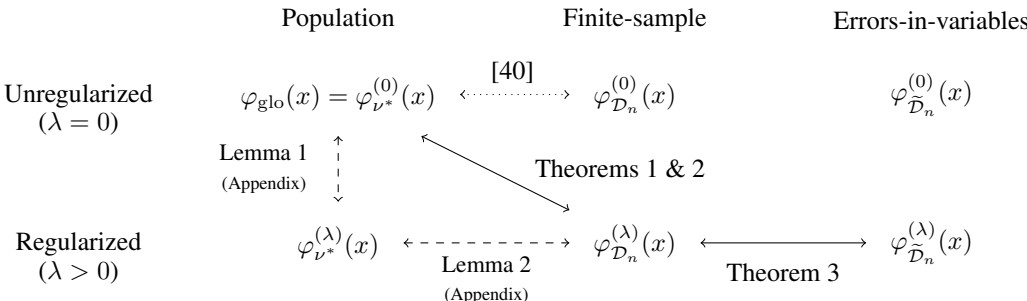

Figure 1: A schematic for the relationship between the regularized Fréchet regression estimators.

## 4.1  Model assumptions and examples

We impose the following assumptions for our analysis.

(C0) (Existence) For any probability distribution $\nu$ and any $\lambda \in \mathbb{R}_+$, the object $\varphi_\nu^{(\lambda)}(x)$ exists (almost surely) and is unique. In particular, $\inf_{y \in \mathcal{M}: d(y, \varphi_{\mathrm{glo}}(x)) > \varepsilon} R(y; x) > R(\varphi_{\mathrm{glo}}(x); x)$ for all $\varepsilon > 0$, where $R(y; x) := R_{\nu^*}^{(0)}(y; x)$.

(C1) (Growth) There exist $D_{\mathtt{g}} > 0$, $C_{\mathtt{g}} > 0$ and $\alpha > 1$, possibly depending on $x$, such that for any probability distribution $\nu$ and any $\lambda \in \mathbb{R}_+$,

$$\begin{cases} d(y, \varphi_\nu^{(\lambda)}(x)) < D_{\mathtt{g}} & \implies & R_\nu^{(\lambda)}(y; x) - R_\nu^{(\lambda)}(\varphi_\nu^{(\lambda)}(x); x) \geq C_{\mathtt{g}} \cdot d(y, \varphi_\nu^{(\lambda)}(x))^\alpha, \\ d(y, \varphi_\nu^{(\lambda)}(x)) \geq D_{\mathtt{g}} & \implies & R_\nu^{(\lambda)}(y; x) - R_\nu^{(\lambda)}(\varphi_\nu^{(\lambda)}(x); x) \geq C_{\mathtt{g}} \cdot D_{\mathtt{g}}^\alpha. \end{cases} \tag{10}$$

(C2) (Bounded entropy) There exists $C_{\mathtt{e}} > 0$, possibly depending on $y$, such that

$$\limsup_{\delta \to 0} \int_0^1 \sqrt{1 + \log \mathfrak{N}(B_d(y, \delta), \delta\varepsilon)} \, \mathrm{d}\varepsilon \leq C_{\mathtt{e}}, \tag{11}$$

where $B_d(y, \delta) := \{y' \in \mathcal{M} : d(y, y') \leq \delta\}$ and $\mathfrak{N}(S, \varepsilon)$ is the $\varepsilon$-covering number[2] of $S$.

---

[2]The formal definition of covering number is provided in Appendix A; see Definition 6.

Assumption (C0) is common to establish the consistency of an M-estimator [55, Chapter 3.2]; in particular, it ensures the weak convergence of the empirical process $R_{\mathcal{D}_n}^{(\lambda)}$ to the population process $R_{\nu^*}^{(\lambda)}$ implying convergence of their minimizers. Furthermore, the conditions on the curvature (C1) and the covering number (C2) control the behavior of the objectives near the minimum in order to obtain rates of convergence; it is worth mentioning that (C2) corresponds to a (locally) bounded entropy for every $y \in \mathcal{M}$, while (P1) in [40] requires the same condition only with $y = \varphi_{\mathrm{glo}}(x)$. These conditions arise from empirical process theory and are also commonly adopted [40, 44, 45].

Here we provide several examples of the space $\mathcal{M}$, in which the conditions (C0), (C1) and (C2) are satisfied. We verify the conditions in Appendix A; see Propositions 1, 2, 3, and 4.

**Example 1.** *Let $\mathcal{M} = (\mathcal{H}, d_{\mathrm{HS}})$ be a finite-dimensional Hilbert space $\mathcal{H}$ equipped with the Hilbert-Schmidt metric $d_{\mathrm{HS}}(y_1, y_2) = \langle y_1 - y_2, y_1 - y_2 \rangle^{1/2}$, e.g., $\mathcal{M} = (\mathbb{R}^r, d_2)$ where $d_2$ is the $\ell^2$-metric.*

**Example 2.** *Let $\mathcal{M}$ be $\mathcal{W}$, the set of probability distributions $G$ on $\mathbb{R}$ such that $\int_{\mathbb{R}} x^2 \, dG(x) < \infty$, equipped with the Wasserstein metric $d_W$ defined as*

$$d_W(G_1, G_2)^2 = \int_0^1 \left( G_1^{-1}(t) - G_2^{-1}(t) \right)^2 \, dt,$$

*where $G_1^{-1}$ and $G_2^{-1}$ are the quantile functions of $G_1$ and $G_2$, respectively. See [40, Section 6].*

**Example 3.** *Let $\mathcal{M} = \left\{ M \in \mathbb{R}^{r \times r} : M = M^T, M \succeq 0 \text{ and } M_{ii} = 1, \forall i \in [r] \right\}$ be the set of correlation matrices of size $r$, equipped with the Frobenius metric, $d_F(M, M') = \|M - M'\|_F$.*

**Example 4.** *Let $\mathcal{M}$ be a (bounded) Riemannian manifold of dimension $r$, and let $d_g$ be the geodesic distance induced by the Riemannian metric.*

## 4.2 Theorem statements

### 4.2.1 Noiseless covariate setting

We first verify the consistency of the $\lambda$-regularized Fréchet regression function as follows.

**Theorem 1** (Consistency). *Suppose that Assumption (C0) holds. If $\mathrm{diam}\,(\mathcal{M}) < \infty$, then for any $\lambda \in \mathbb{R}$ such that $0 \le \lambda < \min\{\sigma_i(\Sigma_{\nu^*}) : \sigma_i(\Sigma_{\nu^*}) > 0\}$, and any $x \in \mathbb{R}^p$,*

$$d\big(\varphi_{\mathcal{D}_n}^{(\lambda)}(x), \varphi_{\nu^*}^{(0)}(x)\big) = o_P(1) \qquad \text{as } n \to \infty. \tag{12}$$

If $\lambda < \sigma^{(0)}(\Sigma_{\nu^*}) = \min\{\sigma_i(\Sigma_{\nu^*}) : \sigma_i(\Sigma_{\nu^*}) > 0\}$, then the regularized estimator $\varphi_{\mathcal{D}_n}^{(\lambda)}(x)$ effectively reduces to the same as the sample-analog estimator $\widehat{\varphi}_{\mathcal{D}_n}(x)$ in (4) in the limit $n \to \infty$. Thus, $\varphi_{\mathcal{D}_n}^{(\lambda)}(x)$ inherits the consistency of $\widehat{\varphi}_{\mathcal{D}_n}$. We provide a detailed proof of Theorem 1 in Appendix B.

In addition to the consistency of $\varphi_{\mathcal{D}_n}^{(\lambda)}$ in the small $\lambda$ limit, we present an analysis for the convergence rate of $\varphi_{\mathcal{D}_n}^{(\lambda)}(x)$ in the following theorem.

**Definition 5.** *The* Mahalanobis seminorm *of $x$ induced by a positive semidefinite matrix $\Sigma$ is $\|x\|_\Sigma := \left( x^\top \Sigma^\dagger x \right)^{1/2}$.*

**Theorem 2** (Rate of convergence). *Suppose that Assumptions (C0)–(C2) hold. If $\mathrm{diam}\,(\mathcal{M}) < \infty$, then for any $\lambda \in \mathbb{R}_+$ and $x \in \mathbb{R}^p$ such that $\|x - \mu_{\nu^*}\|_{\Sigma_{\nu^*}} \le \frac{C_g \cdot D_g^\alpha}{\mathrm{diam}\,(\mathcal{M})^2 \cdot \sqrt{\mathrm{rank}\,\Sigma_{\nu^*}}}$,*

$$d\left(\varphi_{\mathcal{D}_n}^{(\lambda)}(x), \varphi_{\nu^*}^{(0)}(x)\right) = O_P\left( b_\lambda(x)^{\frac{1}{\alpha-1}} + n^{-\frac{1}{2(\alpha-1)}} \right) \quad \text{as} \quad n \to \infty, \tag{13}$$

*where $b_\lambda(x) = \mathrm{rank}\left(\Sigma_{\nu^*} - \Sigma_{\nu^*}^{(\lambda)}\right)^{\frac{1}{2}} \cdot \|x - \mu_{\nu^*}\|_{\Sigma_{\nu^*} - \Sigma_{\nu^*}^{(\lambda)}}$.*

We obtain Theorem 2 by showing a "bias" upper bound $d\big(\varphi_{\nu^*}^{(\lambda)}(x), \varphi_{\nu^*}^{(0)}(x)\big) = O\big(b_\lambda(x)^{\frac{1}{\alpha-1}}\big)$ and a "variance" bound $d\big(\varphi_{\mathcal{D}_n}^{(\lambda)}(x), \varphi_{\nu^*}^{(\lambda)}(x)\big) = O_P\big(n^{-\frac{1}{2(\alpha-1)}}\big)$; see Lemmas 1 and 2 in Appendix C. Here we remark that $b_\lambda(x)$ is a monotone non-decreasing function of $\lambda$, and if $\lambda < \sigma^{(0)}(\Sigma_{\nu^*})$ then $b_\lambda(x) = 0$. Also, the condition on $\|x - \mu_{\nu^*}\|_{\Sigma_{\nu^*}}$ is introduced for a technical reason, and can be removed when $D_g = \infty$. Note that Condition (C1) holds with $D_g = \infty$ and $\alpha = 2$ for Examples 1, 2 and 3. Thus, we have $d\big(\varphi_{\mathcal{D}_n}^{(\lambda)}(x), \varphi_{\nu^*}^{(0)}(x)\big) = O_P\big(b_\lambda(x) + n^{-\frac{1}{2}}\big)$ as $n \to \infty$.

#### 4.2.2 Error-prone covariate setting

Given a set $\mathcal{D}_n = \{(x_i, y_i) : i \in [n]\}$, let $\boldsymbol{X}_{\mathcal{D}_n} := [x_1 \cdots x_n]^\top \in \mathbb{R}^{n \times p}$. We let $\boldsymbol{X} = \boldsymbol{X}_{\mathcal{D}_n}$ and $\boldsymbol{Z} = \boldsymbol{X}_{\widetilde{\mathcal{D}}_n}$ for shorthand, and further, we let $\boldsymbol{X}_{\mathrm{ctr}} = \left(\boldsymbol{I}_n - \frac{1}{n} \mathbf{1}_n \mathbf{1}_n^\top\right) \boldsymbol{X}$ and $\boldsymbol{Z}_{\mathrm{ctr}} = \left(\boldsymbol{I}_n - \frac{1}{n} \mathbf{1}_n \mathbf{1}_n^\top\right) \boldsymbol{Z}$ denote the 'row-centered' matrices.

**Theorem 3** (De-noising covariates)**.** *Suppose that Assumptions (C0) and (C1) hold. Then there exists a constant $C > 0$ such that for any $\lambda \in \mathbb{R}_+$, if $x \in \mu_{\mathcal{D}_n} + \mathrm{rowsp}\,\boldsymbol{X}_{\mathrm{ctr}}$ and*

$$\|x - \mu_{\mathcal{D}_n}\|_{\Sigma_{\mathcal{D}_n}} \leq \frac{1}{2} \left( \frac{C_g \cdot D_g^\alpha}{2 \operatorname{diam}(\mathcal{M})} \cdot \frac{\sigma^{(\lambda)}(\boldsymbol{X}_{\mathrm{ctr}}) \wedge \sigma^{(\lambda)}(\boldsymbol{Z}_{\mathrm{ctr}})}{\|\boldsymbol{Z} - \boldsymbol{X}\|} - 1 \right), \tag{14}$$

*then*

$$d\left(\varphi_{\widetilde{\mathcal{D}}_n}^{(\lambda)}(x), \varphi_{\mathcal{D}_n}^{(\lambda)}(x)\right) \leq C \cdot \left( \frac{\|\boldsymbol{Z} - \boldsymbol{X}\|}{\sigma^{(\lambda)}(\boldsymbol{X}_{\mathrm{ctr}}) \wedge \sigma^{(\lambda)}(\boldsymbol{Z}_{\mathrm{ctr}})} \cdot \frac{2 \cdot \|x - \mu_{\mathcal{D}_n}\|_{\Sigma_{\mathcal{D}_n}} + 1}{C_g} \right)^{\frac{1}{\alpha}}. \tag{15}$$

Note that the condition on $\|x - \mu_{\nu^*}\|_{\Sigma_{\nu^*}}$ in (14) can be removed when $D_g = \infty$. We highlight that the quantity $\frac{\|\boldsymbol{Z} - \boldsymbol{X}\|}{\sigma^{(\lambda)}(\boldsymbol{X}_{\mathrm{ctr}}) \wedge \sigma^{(\lambda)}(\boldsymbol{Z}_{\mathrm{ctr}})}$ acts as the reciprocal of the signal-to-noise ratio. Here, $\|\boldsymbol{Z} - \boldsymbol{X}\|$ quantifies the magnitude of the "noise" in the covariates, while $\min\left\{\sigma^{(\lambda)}(\boldsymbol{X}_{\mathrm{ctr}}), \sigma^{(\lambda)}(\boldsymbol{Z}_{\mathrm{ctr}})\right\}$ measures the strength of the "signal" retained in the $\lambda$-SVT of the design matrix. We observe that the error bound (15) increases proportionally to the normalized deviation of $x$ from the mean, $\mu_{\mathcal{D}_n}$, which is a reasonable outcome. For the complete version of Theorem 3 and its proof, refer to Appendix D.

**Remarks on Theorem 3.** We avoid imposing distributional assumptions on the noise $\varepsilon = Z - X$, to ensure broad applicability of the result. Also, the inequality (15) is sharp, as there is a worst-case noise instance that attains equality (up to a multiplicative constant). Despite its generality, this upper bound highlights effective error mitigation in specific scenarios. For instance, consider well-balanced, effectively low-rank covariates $\boldsymbol{X} \in \mathbb{R}^{n \times p}$ such that $|X_{ij}| = \Omega(1)$ for all $i, j$ and $\sigma_1(\boldsymbol{X}) \asymp \sigma_r(\boldsymbol{X}) \gg \sigma_{r+1}(\boldsymbol{X}) \asymp \sigma_{n \wedge p}(\boldsymbol{X}) = O(1)$, where $r \ll n \wedge p$ is the effective rank of $\boldsymbol{X}$. Then $\sigma_1(\boldsymbol{X})^2 \asymp \sigma_r(\boldsymbol{X})^2 \asymp \|\boldsymbol{X}\|_F^2 / r \gtrsim np/r$. Additionally, if $\boldsymbol{Z} = \boldsymbol{X} + \boldsymbol{E}$ where $\boldsymbol{E}$ is a random matrix with independent sub-Gaussian rows, then $\|\boldsymbol{Z} - \boldsymbol{X}\| \lesssim \sqrt{n} + \sqrt{p}$ with high probability. In the random design scenario where the rows of $\boldsymbol{X}$ and the test point $x$ are drawn IID from the same distribution, $\|x - \mu_{D_n}\|_{\Sigma} \approx 1$ with high probability. Consequently, the upper bound in (15) is bounded by $\sqrt{r/p} + \sqrt{r/n}$, which diminishes to 0 when $r \ll n \wedge p$.

### 4.3 Proof sketches

**Proof of Theorem 1.** We show that $R_{\mathcal{D}_n}^{(\lambda)}(y; x)$ weakly converges to $R_{\nu^*}^{(0)}(y; x)$ in the $\ell^\infty(\mathcal{M})$-sense. According to [55, Theorem 1.5.4], it suffices to show that (1) $R_{\mathcal{D}_n}^{(\lambda)}(y; x) - R_{\nu^*}^{(0)}(y; x) = o_p(1)$ for all $y \in \mathcal{M}$, and (2) $R_{\mathcal{D}_n}^{(\lambda)}$ is asymptotically equicontinuous in probability.

**Proof of Theorem 2.** We prove upper bounds for the bias and the variance separately.

*To control the bias* (Appendix C, Lemma 1), we show an upper bound for $R\left(\varphi^{(\lambda)}(x); x\right) - R\left(\varphi(x); x\right)$, and convert it to restrain the distance between the minimizers $d\left(\varphi^{(\lambda)}(x), \varphi(x)\right)$ using the `Growth` condition (C1). In this conversion, we employ the "peeling technique" in empirical process theory.

*To control the variance* (Appendix C, Lemma 2), we follow a similar strategy as in Lemma 1, but with additional technical considerations. Defining the 'fluctuation variable' $Z_n^{(\lambda)}(y; x) := R_{\mathcal{D}_n}^{(\lambda)}(y; x) - R_{\nu^*}^{(\lambda)}(y; x)$ parameterized by $y \in \mathcal{M}$, we derive a probabilistic upper bound for $R_{\nu^*}^{(\lambda)}(\varphi_{\mathcal{D}_n}^{(\lambda)}(x); x) - R_{\nu^*}^{(\lambda)}(\varphi_{\nu^*}^{(\lambda)}(x); x)$ by establishing a uniform upper bound for $Z_n^{(\lambda)}(y; x) - Z_n^{(\lambda)}(\varphi(x); x)$; here, the `Entropy` condition (C2) is used. Again, we use the `Growth` condition (C1) and the peeling technique to obtain a probabilistic upper bound for the distance $d(\varphi_{\mathcal{D}_n}^{(\lambda)}(x), \varphi_{\nu^*}^{(\lambda)}(x))$.

**Proof of Theorem 3.** Expressing the difference in the weights $w_{\widetilde{\mathcal{D}}_n}^{(\lambda)}(y; x) - w_{\mathcal{D}_n}^{(\lambda)}(y; x)$ in terms of $\boldsymbol{X}$ and $\boldsymbol{Z}$, we utilize classical matrix perturbation theory to control $R_{\widetilde{\mathcal{D}}_n}^{(\lambda)}(y; x) - R_{\mathcal{D}_n}^{(\lambda)}(y; x)$, and transform it to an upper bound on the distance $d(\varphi_{\widetilde{\mathcal{D}}_n}^{(\lambda)}(x), \varphi_{\mathcal{D}_n}^{(\lambda)}(x))$ using the `Growth` condition (C1).

# 5 Experiments

In this section, we present numerical simulation results to validate and support our theoretical findings. We focus on global Fréchet regression analysis for one-dimensional distribution functions (Example 2). These simulations cover various conditions, allowing us to evaluate and compare our methodology's performance with alternative approaches. For a summary of the experimental results, please refer to Figure 2 and Table 1. Further details about simulation settings and additional results are provided in Appendix E.

**Experimental setup.** We consider combinations of $p \in \{150, 300, 600\}$ and $n \in \{100, 200, 400\}$. The datasets $\mathcal{D}_n = \{(X_i, Y_i) : i \in [n]\}$ and $\widetilde{\mathcal{D}}_n = \{(Z_i, Y_i) : i \in [n]\}$ are generated as follows. Let $X_i \sim \mathcal{N}_p(\mathbf{0}_p, \Sigma)$ be IID multivariate Gaussian with mean $\mathbf{0}_p$ and covariance $\Sigma$ such that $\mathrm{spec}(\Sigma) = \{\kappa_j > 0 : j \in [p]\}$ is an exponentially decreasing sequence such that $\mathrm{tr}(\Sigma) = \sum_{j=1}^{p} \kappa_j = p$ and $\kappa_1 / \kappa_p = 10^3$. Note that $\sum_{j=1}^{\lfloor p/3 \rfloor} \kappa_j / \sum_{j'=1}^{p} \kappa_{j'} \approx 0.9$, and thus, $\Sigma$ is effectively low-rank. We generate $Z_i$ following (5) under two scenarios $\varepsilon_{ij} \overset{IID}{\sim} \mathcal{N}(0, \sigma_\varepsilon^2)$ and $\varepsilon_{ij} \overset{IID}{\sim} \mathrm{Laplace}(0, \sigma_\varepsilon)$, respectively. Lastly, given $X = x$, let $Y$ be the distribution function of $\mathcal{N}(\mu_{\alpha,\beta}(x) + \eta, \tau^2)$, where (i) $\mu_{\alpha,\beta}(x) = \alpha + \beta^\top x$ with $\alpha = 1$ and $\beta = p^{-1/2} \cdot \mathbf{1}_p$; (ii) $\eta \sim \mathcal{N}(0, \sigma_\eta^2)$; and (iii) $\tau^2 \sim \mathcal{IG}(s_1, s_2)$, an inverse gamma distribution with shape $s_1$ and scale $s_2$. We performed $B = 500$ Monte Carlo experiments by drawing $\mathcal{D}_n^{(b)}$ and $\widetilde{\mathcal{D}}_n^{(b)}$ as independent copies of $\mathcal{D}_n$ and $\widetilde{\mathcal{D}}_n$, respectively, for $b \in [B]$.

**Performance evaluation.** We assess the in-sample and out-of-sample performance of the Fréchet regression function estimator by using the mean squared error (MSE) and the mean squared prediction error (MSPE). To this end, we create a "test set" $\mathcal{D}_N^{\mathrm{new}} = \{(X_i^{\mathrm{new}}, Y_i^{\mathrm{new}}) : i \in [N]\}$, with $N = 1000$. The MSE and the MSPE are computed as the average of squared metric-distance residuals from the observed responses in the "training set" $\mathcal{D}_n$ and in the "test set" $\mathcal{D}_N^{\mathrm{new}}$), respectively:

$$\mathrm{MSE}(\varphi_\nu^{(\lambda)}) = \frac{1}{n} \sum_{i=1}^{n} d_W(Y_i, \varphi_\nu^{(\lambda)}(X_i))^2 \text{ and } \mathrm{MSPE}(\varphi_\nu^{(\lambda)}) = \frac{1}{N} \sum_{i=1}^{N} d_W(Y_i^{\mathrm{new}}, \varphi_\nu^{(\lambda)}(X_i^{\mathrm{new}}))^2.$$

We report the MSE averaged over $B = 500$ random trials: $\mathrm{MSE}(\varphi_\nu^{(\lambda)}) = B^{-1} \sum_{b=1}^{B} \mathrm{MSE}(\varphi_{\nu^{(b)}}^{(\lambda)})$, and likewise for MSPE. Furthermore, we evaluate the accuracy and efficiency of the estimator using bias and variance, with detailed definitions deferred to Appendix E.

**Simulation results.** Our numerical study demonstrates that the proposed SVT method consistently improves both estimation and prediction performance, especially in the errors-in-variables setting. Figure 2 highlights how the SVT estimator outperforms the naïve errors-in-variables (EIV) estimator, which corresponds to SVT with $\lambda = 0$. The naïve EIV suffers from an intrinsic model bias, called the attenuation effect [13], as it regresses responses on error-prone covariates. This leads to a misrepresentation of the association between responses and true covariates, potentially leading to statistical inference based on a mis-specified model.

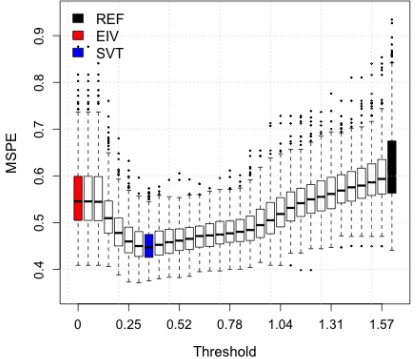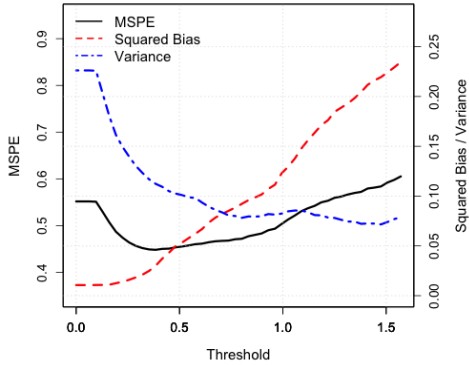

Figure 2: Comparison of the prediction performance of $\varphi_{\mathcal{D}_n}^{(0)}$ (REF), $\varphi_{\widetilde{\mathcal{D}}_n}^{(0)}$ (EIV), and $\varphi_{\widetilde{\mathcal{D}}_n}^{(\lambda)}$ (SVT) (left), and the trade-off between the bias and the variance (right) for $B = 500$, $p = 50$ and $n = 100$.

Table 1: Average performance of $\varphi_{\mathcal{D}_n}^{(0)}$ (REF), $\varphi_{\widetilde{\mathcal{D}}_n}^{(0)}$ (EIV), and $\varphi_{\widetilde{\mathcal{D}}_n}^{(\lambda)}$ (SVT) in various settings (**boldface** indicates the best). The choice of threshold $\lambda$ for SVT is detailed in Appendix E.

| Error distribution | Sample size | | $n = 100$ | | | $n = 200$ | | | $n = 400$ | | |
|---|---|---|---|---|---|---|---|---|---|---|---|
| | $p$ | Criterion | REF | EIV | SVT | REF | EIV | SVT | REF | EIV | SVT |
| Gaussian $N(0, 0.05^2)$ | 150 | Bias | **0.107** | 0.164 | 0.312 | **0.040** | 0.087 | 0.189 | **0.020** | 0.078 | 0.136 |
| | | $\sqrt{\mathrm{Var}}$ | 1.147 | 0.973 | **0.531** | 0.969 | 0.857 | **0.414** | 0.432 | 0.381 | **0.295** |
| | | MSE | **0.000** | 0.154 | 0.205 | **0.075** | 0.334 | 0.232 | **0.190** | 0.255 | 0.267 |
| | | MSPE | 1.613 | 1.267 | **0.686** | 1.245 | 1.042 | **0.513** | 0.492 | 0.456 | **0.412** |
| | 300 | Bias | **0.317** | 0.352 | 0.485 | **0.112** | 0.170 | 0.327 | 0.039 | 0.091 | 0.207 |
| | | $\sqrt{\mathrm{Var}}$ | 0.805 | 0.731 | **0.523** | 1.137 | 0.970 | **0.512** | 0.964 | 0.845 | **0.405** |
| | | MSE | **0.000** | 0.033 | 0.217 | **0.000** | 0.152 | 0.235 | **0.075** | 0.327 | 0.237 |
| | | MSPE | 1.045 | 0.956 | **0.808** | 1.602 | 1.267 | **0.667** | 1.232 | 1.025 | **0.509** |
| | 600 | Bias | **0.568** | 0.592 | 0.619 | **0.311** | 0.353 | 0.451 | **0.104** | 0.155 | 0.298 |
| | | $\sqrt{\mathrm{Var}}$ | 0.663 | 0.631 | **0.598** | 0.799 | 0.735 | **0.589** | 1.135 | 0.956 | **0.510** |
| | | MSE | **0.000** | 0.016 | 0.067 | **0.000** | 0.036 | 0.112 | **0.000** | 0.153 | 0.208 |
| | | MSPE | 1.075 | 1.062 | **1.054** | 1.039 | 0.968 | **0.858** | 1.602 | 1.242 | **0.657** |
| Laplacian $DE(0, 0.05)$ | 150 | Bias | **0.102** | 0.112 | 0.286 | **0.045** | 0.053 | 0.171 | **0.019** | 0.025 | 0.121 |
| | | $\sqrt{\mathrm{Var}}$ | 1.155 | 1.113 | **0.549** | 0.971 | 0.949 | **0.421** | 0.431 | 0.416 | **0.297** |
| | | MSE | **0.000** | 0.026 | 0.187 | **0.075** | 0.151 | 0.218 | **0.189** | 0.205 | 0.253 |
| | | MSPE | 1.633 | 1.538 | **0.688** | 1.254 | 1.207 | **0.513** | 0.489 | 0.477 | **0.406** |
| | 300 | Bias | **0.321** | 0.325 | 0.491 | **0.105** | 0.117 | 0.344 | **0.043** | 0.048 | 0.198 |
| | | $\sqrt{\mathrm{Var}}$ | 0.805 | 0.794 | **0.525** | 1.140 | 1.099 | **0.511** | 0.960 | 0.929 | **0.413** |
| | | MSE | **0.000** | 0.003 | 0.233 | **0.000** | 0.025 | 0.227 | **0.076** | 0.148 | 0.229 |
| | | MSPE | 1.049 | 1.034 | **0.814** | 1.610 | 1.521 | **0.677** | 1.226 | 1.169 | **0.514** |
| | 600 | Bias | **0.566** | 0.568 | 0.608 | **0.312** | 0.317 | 0.443 | **0.102** | 0.109 | 0.282 |
| | | $\sqrt{\mathrm{Var}}$ | 0.664 | 0.661 | **0.614** | 0.800 | 0.792 | **0.606** | 1.134 | 1.094 | **0.521** |
| | | MSE | **0.000** | 0.001 | 0.074 | **0.000** | 0.004 | 0.157 | **0.000** | 0.025 | 0.200 |
| | | MSPE | 1.073 | 1.071 | **1.059** | 1.045 | 1.035 | **0.872** | 1.602 | 1.515 | **0.662** |

Remarkably, the SVT estimator achieved a smaller MSPE even compared to the oracle estimator (REF) obtained from the error-free sample. Although the REF estimator had the smallest MSE due to its small bias, we observed its overfitting to the training sample, resulting in poor prediction performance. Notably, even the naïve EIV estimator outperformed the REF estimator in MSPE. We believe this is mainly because the true covariate matrix was nearly singular in our simulation setup, causing multicollinearity issues for the REF. In contrast, measurement errors introduced non-ignorable minimum singular values in the EIV covariate matrix, unintentionally mitigating multicollinearity for the naïve EIV and causing it to behave like ridge regression.

## 6 Discussion

This paper has addressed errors-in-variables regression of non-Euclidean response variables through the (global) Fréchet regression framework enhanced by low-rank approximation of covariates. Specifically, we introduce a novel *regularized (global) Fréchet regression* framework (Section 3), which combines the Fréchet regression with principal component regression. We also provide a comprehensive theoretical analysis in three main theorems (Section 4), and validate our theory through numerical experiments on simulated datasets. Moreover, our numerical experiments demonstrate empirical evidence of the effectiveness and superiority of our approach, reinforcing its practical relevance and potential impact in non-Euclidean regression analysis.

We conclude this paper by proposing several promising directions for future research. First, it would be worthwhile to explore the large sample theory for selecting the optimal threshold parameter $\lambda$ in the proposed SVT method, in order to characterize the theoretical phase transition of the bias-variance trade-off in the regularized (global) Fréchet regression. Second, we believe that our framework could be extended to errors-in-variables Fréchet regression for response variables in a broader class of metric spaces, e.g., by leveraging the quadruple inequality proposed by Schötz [44, 45]. Lastly, investigating the asymptotic distribution of the proposed SVT estimator would be highly appealing in the statistical literature, as it would enable us to make statistical inferences on the conditional Fréchet mean in non-Euclidean spaces [6, 8] with errors-in-variables covariates.

## Acknowledgments and Disclosure of Funding

DS acknowledges support from the National Science Foundation under Grant No. CCF-2212326.

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

# A Verification of the model assumptions

## A.1 Additional background

**Definition 6** ($\varepsilon$-net and covering number). *Let $(\mathcal{M}, d)$ be a metric space. Let $S \subseteq T$ be a subset and let $\varepsilon > 0$. A subset $\mathcal{N} \subseteq S$ is called an $\varepsilon$-net of $S$ if every point in $S$ is within distance $\varepsilon$ of some point $\mathcal{N}$, i.e.,*

$$\forall x \in S, \ \exists x_0 \in \mathcal{N} \text{ such that } d(x, x_0) \leq \varepsilon.$$

*The $\varepsilon$-covering number of $S$, denoted by $\mathfrak{N}(S, \varepsilon)$, is the smallest possible cardinality of an $\varepsilon$-net of $S$, i.e.,*

$$\mathfrak{N}(S, \varepsilon) := \min \left\{ k \in \mathbb{N} : \exists y_1, \ldots, y_k \in \mathcal{M} \text{ such that } S \subseteq \bigcup_{i=1}^{k} B_d(y_i, \varepsilon) \right\}, \tag{16}$$

*where $B_d(y, \varepsilon) = \{y' \in \mathcal{M} : d(y, y') \leq \varepsilon\}$ denotes the closed $\varepsilon$-ball centered at $y \in \mathcal{M}$.*

Let $B_2^r(0, 1) := \{x \in \mathbb{R}^r : \|x\|_2 \leq 1\}$ denote the unit $\ell^2$-norm ball in $\mathbb{R}^r$. It is well known[3] that for any $\varepsilon > 0$,

$$\left( \frac{1}{\varepsilon} \right)^r \leq \mathfrak{N}\big(B_2^r(0, 1), \varepsilon\big) \leq \left( \frac{2}{\varepsilon} + 1 \right)^r. \tag{17}$$

## A.2 Example 1: Euclidean space

**Proposition 1.** *The space $(\mathcal{H}, d_{\mathrm{HS}})$ defined in Example 1 satisfies Assumptions (C0), (C1), and (C2).*

*Proof of Proposition 1.* For any probability distribution $\nu$ and any $\lambda \in \mathbb{R}_+$, let $y_\nu^{(\lambda)} := \mathbb{E}_\nu\big[w_\nu^{(\lambda)}(X, x) \cdot Y\big]$. Then we observe that

$$
\begin{aligned}
R_\nu^{(\lambda)}(y; x) &= \mathbb{E}_\nu\big[w_\nu^{(\lambda)}(X, x) \cdot d^2(Y, y)\big] \\
&= \mathbb{E}_\nu\left[w_\nu^{(\lambda)}(X, x) \cdot \|Y - y\|^2\right] \\
&= \mathbb{E}_\nu\left[w_\nu^{(\lambda)}(X, x) \cdot \|Y - y_\nu^{(\lambda)}\|^2\right] + \|y - y_\nu^{(\lambda)}\|_{\mathrm{HS}}^2 \\
&\quad + 2 \left\langle \underbrace{\mathbb{E}_\nu\left[w_\nu^{(\lambda)}(X, x) \cdot \big(Y - y_\nu^{(\lambda)}\big)\right]}_{=0}, \ y_\nu^{(\lambda)} - y \right\rangle \\
&= R_\nu^{(\lambda)}(y_\nu^{(\lambda)}; x) + \|y - y_\nu^{(\lambda)}\|_{\mathrm{HS}}^2.
\end{aligned}
$$

As $R_\nu^{(\lambda)}(y; x)$ is a strictly convex and coercive function, there exists a unique minimizer, $\varphi_\nu^{(\lambda)}$. Thus, Condition (C0) is proved. Furthermore, Condition (C1) is also satisfied with $D_{\mathbf{g}} = \infty$, $C_{\mathbf{g}} = 1$, and $\alpha = 2$.

Lastly, for any $y \in \mathcal{H}$ and any $\varepsilon > 0$,

$$\mathfrak{N}\big(B_{d_{\mathrm{HS}}}(y, \delta), \delta\varepsilon\big) = \mathfrak{N}\big(B_{d_{\mathrm{HS}}}(y, 1), \varepsilon\big) \leq \left( \frac{2}{\varepsilon} + 1 \right)^r \leq C \cdot \varepsilon^{-r}$$

where $r = \dim \mathcal{H}$ and $C > 1$ is a constant that depends on $r$ only; see the covering number upper bound in (17). Thus, the integral (11) is bounded as follows:

$$
\begin{aligned}
\int_0^1 \sqrt{1 + \log \mathfrak{N}\big(B_d\big(\varphi(x), \delta\big), \delta\varepsilon\big)}\, \mathrm{d}\varepsilon &\leq \int_0^1 \sqrt{1 + \log C - r \log \varepsilon}\, \mathrm{d}\varepsilon \\
&\leq \sqrt{1 + \log C} + \sqrt{r} \int_0^1 \sqrt{-\log \varepsilon}\, \mathrm{d}\varepsilon \\
&= \sqrt{1 + \log C} + \sqrt{r} \int_0^\infty e^{-t} \sqrt{t}\, \mathrm{d}t \\
&= \sqrt{1 + \log C} + \frac{\sqrt{r\pi}}{2}
\end{aligned}
$$

---

[3]See [56, Corollary 4.2.13] for example.

using the change of variable $t = -\log \varepsilon$. Therefore, Assumption (C2) holds with $C_e = \sqrt{1 + \log C} + \frac{\sqrt{r\pi}}{2}$.

$\square$

### A.3 Example 2: Set of probability distributions

**Proposition 2.** *The space* $(\mathcal{W}, d_W)$ *defined in Example 2 satisfies Assumptions (C0), (C1), and (C2).*

*Proof of Proposition 2.* For a probability distribution function $y \in \mathcal{W}$ defined on $\mathbb{R}$, let $\mathcal{Q} = Q(\mathcal{W}) := \{Q(y) : y \in \mathcal{W}\}$ denote the collection of corresponding quantile functions, where $(Q(y))(u) = y^{-1}(u)$ for $u \in [0, 1]$.

We note that $f \mapsto \mathbb{E}_\nu\big[w_\nu^{(\lambda)}(X, x)\langle Q(Y), f\rangle\big]$ is a bounded linear functional defined on $L^2[0, 1]$ because $\mathbb{E}_\nu|w_\nu^{(\lambda)}(X, x)|^2 \le 2 + 2p\,\|(x - \mu_\nu)\|_{\Sigma_\nu}^2$ implies that $\mathbb{E}_\nu\big[w_\nu^{(\lambda)}(X, x)|\cdot\|Q(Y)\|_2\big]$ is bounded. It follows from the Riesz representation theorem that there exists $f_x^{(\lambda)} \in L^2[0, 1]$ such that

$$\mathbb{E}_\nu\big[w^{(\lambda)}(X, x)\langle Q(Y), g\rangle_2\big] = \langle f_x^{(\lambda)}, g\rangle_2 \tag{18}$$

for all $g \in L^2[0, 1]$. Then, we have

$$R_\nu^{(\lambda)}(y; x) = \mathbb{E}_\nu\big[w_\nu^{(\lambda)}(X, x)\,\|Q(Y) - f_x^{(\lambda)}\|_2^2\big] + \|Q(y) - f_x^{(\lambda)}\|_2^2, \tag{19}$$

which yields that

$$\varphi_\nu^{(\lambda)}(x) = Q^{-1}\bigg(\underset{Q \in \mathcal{Q}}{\arg\min}\,\|Q - f_x^{(\lambda)}\|_2^2\bigg). \tag{20}$$

The condition (C0) follows from the convexity of $(\mathcal{Q}, \|\cdot\|_2)$. Moreover, the convexity also gives $\big\langle Q(y) - Q(\varphi_\nu^{(\lambda)}(x)), f_x^{(\lambda)}(x) - Q(\varphi_\nu^{(\lambda)}(x))\big\rangle_2 \le 0$ for all $y \in \mathcal{W}$, so that

$$
\begin{aligned}
R_\nu^{(\lambda)}&(y; x) - R_\nu^{(\lambda)}(\varphi^{(\lambda)}(x); x) \\
&= \|Q(y) - f_x^{(\lambda)}(x)\|_2^2 - \|Q(\varphi_\nu^{(\lambda)}(x)) - f_x^{(\lambda)}(x)\|_2^2 \\
&= \|Q(y) - Q(\varphi_\nu^{(\lambda)}(x))\|_2 - 2\big\langle Q(y) - Q(\varphi_\nu^{(\lambda)}(x)), f_x^{(\lambda)}(x) - Q(\varphi_\nu^{(\lambda)}(x))\big\rangle_2 \\
&\ge \|Q(y) - Q(\varphi_\nu^{(\lambda)}(x))\|_2 \\
&= d_W^2(y, \varphi_\nu^{(\lambda)}(x)).
\end{aligned} \tag{21}
$$

Therefore, the condition (C1) holds for any arbitrary constant $D_g > 0$ with $C_g = 1$ and $\alpha = 2$.

Finally, we refer to [40, Proposition 1] to ensure that for any $\delta > 0$ and any $\varepsilon > 0$,

$$\sup_{y \in \mathcal{W}} \log \mathfrak{N}\big(B_{d_W}(y, \delta), D_e\varepsilon\big) \le \sup_{Q \in \mathcal{Q}} \log \mathfrak{N}\big(B_{d_2}(Q, \delta), \delta\varepsilon\big) \le C \cdot \varepsilon^{-1} \tag{22}$$

holds with an absolute constant $C > 0$. Technically, this fact comes from the covering number bound for a class of uniformly bounded and monotone functions in $L^2$. This confirms that the entropy condition (C2) holds. $\square$

### A.4 Example 3: Set of correlation matrices

**Proposition 3.** *The space* $(\mathcal{M}, d_F)$ *defined in Example 3 satisfies Assumptions (C0), (C1), and (C2).*

*Proof of Proposition 3.* First of all, we note that $\mathcal{M}$ is a convex subset of $\mathcal{S}^r := \{X \in \mathbb{R}^{r \times r} : X = X^\top\}$, which is the set of $r \times r$ symmetric matrices. It is because $\mathcal{M} = \mathcal{S}_+^r \cap H$ where $\mathcal{S}_+^r$ denotes the cone of $r \times r$ positive semidefinite matrices and $H := \{X \in \mathcal{S}^r : X_{ii} = 1, \forall i \in [r]\}$ denotes an affine subspace of $\mathcal{S}^r$, both of which are convex.

Next, we observe that $\mathcal{S}^r$ equipped with the Frobenius metric $d_F$ is isometrically isomorphic to $\mathbb{R}^{r(r+1)/2}$ equipped with the $\ell^2$-metric. Hence, $(\mathcal{M}, d_F)$ satisfies Assumptions (C0), (C1), and (C2), inheriting these properties from the ambient space $\mathcal{S}^r$, which is established by Proposition 1. We note that the inheritance of (C0), (C1) relies on the convexity of $\mathcal{M}$, while (C2) is inherited simply based on the inclusion $\mathcal{M} \subset \mathcal{S}^r$. $\square$

### A.5 Example 4: Bounded Riemannian manifold

**Proposition 4.** *The space $(\mathcal{M}, d_g)$ defined in Example 4 satisfies Assumption (C2) provided that the Riemannian metric is equivalent to the ambient Euclidean metric.*

*Furthermore, let $T_y\mathcal{M}$ be the tangent space of $\mathcal{M}$ at $y$, and $\mathrm{Exp}_y : T_y\mathcal{M} \to \mathcal{M}$ be the manifold exponential map at $y$. Let $g_\nu^{(\lambda)}(u; y, x) := R_\nu^{(\lambda)}\big(\mathbb{E}_y(u), x\big)$ for $u \in T_y\mathcal{M}$ If (C0) holds and the Hessian of $g_\nu^{(\lambda)}\big(u; \varphi_\nu^{(\lambda)}(x), x\big)$ is positive definite, then (C1) for some $D_g > 0$.*

*Proof of Proposition 3.* Since $\mathcal{M}$ has finite dimension and is bounded, the bounded entropy condition (C2) follows from the metric equivalence.

Suppose that (C0) holds, and let $\delta > 0$ be the injectivity radius at $\varphi_\nu^{(\lambda)}(x)$. Consider $y \in \mathcal{M}$ such that $d\big(y, \varphi_\nu^{(\lambda)}(x)\big) < \delta$, and let $u_y = \mathrm{Log}_{\varphi_\nu^{(\lambda)}(x)}(y)$. Then we have

$$R_\nu^{(\lambda)}(y; x) - R_\nu^{(\lambda)}\big(\varphi_\nu^{(\lambda)}(x); x\big) = g_\nu^{(\lambda)}\big(u_y; \varphi_\nu^{(\lambda)}(x), x\big) - g_\nu^{(\lambda)}\big(0; \varphi_\nu^{(\lambda)}(x), x\big) = u_y^\top \nabla^2 g_\nu^{(\lambda)}(\bar{u}_y)\, u_y$$

for some $\bar{u}_y$ between $0$ and $u_y$. Since $u_y^\top u_y = d\big(y, \varphi_\nu^{(\lambda)}(x)\big)^2$ and $g_\nu^{(\lambda)}$ is continuous, the positive definiteness of $\nabla^2 g_\nu^{(\lambda)}(\bar{u}_y)$ implies (C1) with $\alpha = 1$. $\qquad\square$

## B   Proof of Theorem 1

*Proof of Theorem 1.* Recall from Definition 4, cf. (9), that for any probability distribution $\nu$ on $\mathbb{R}^p$, any $\lambda \in \mathbb{R}_+$, and any $x \in \mathbb{R}^p$, the $\lambda$-regularized Fréchet regression function evaluated at $x$ is given as the minimizer of a function $R_\nu^{(\lambda)}$ as

$$\varphi_\nu^{(\lambda)}(x) = \underset{y \in \mathcal{M}}{\arg\min}\, R_\nu^{(\lambda)}(y; x)$$

where

$$R_\nu^{(\lambda)}(y; x) = \mathbb{E}_{(X,Y)\sim\nu}\left[ w_\nu^{(\lambda)}(X, x) \cdot d^2(Y, y) \right] \quad \text{and}$$

$$w_\nu^{(\lambda)}(x', x) = 1 + (x' - \mu_\nu)^\top \left[ \mathtt{SVT}^{(\lambda)}\big(\Sigma_\nu\big) \right]^\dagger (x - \mu_\nu).$$

In this proof, we follow a similar strategy to that in the proof of [40, Theorem 1]. Specifically, it suffices to show $\sup_{y\in\mathcal{M}} \big| R_{\mathcal{D}_n}^{(\lambda)}(y; x) - R_{\nu^*}^{(0)}(y; x) \big|$ converges to zero in probability, due to [55, Corollary 3.2.3]. To this end, we show $R_{\mathcal{D}_n}^{(\lambda)}(y; x)$ weakly converges to $R_{\nu^*}^{(0)}(y; x)$ in the $\ell^\infty(\mathcal{M})$-sense, and then apply [55, Theorem 1.3.6]. Again, according to [55, Theorem 1.5.4], this weak convergence can be proved by showing that

(S1)  $R_{\mathcal{D}_n}^{(\lambda)}(y; x) - R_{\nu^*}^{(0)}(y; x) = o_p(1)$ for all $y \in \mathcal{M}$, and

(S2)  $R_{\mathcal{D}_n}^{(\lambda)}$ is asymptotically equicontinuous in probability, i.e., for any $\varepsilon, \eta > 0$, there exists $\delta > 0$ such that

$$\limsup_n P\left( \sup_{y_1, y_2 \in \mathcal{M}:\, d(y_1, y_2) < \delta} \left| R_{\mathcal{D}_n}^{(\lambda)}(y_1; x) - R_{\mathcal{D}_n}^{(\lambda)}(y_2; x) \right| > \varepsilon \right) < \eta.$$

In what follows, we prove these two statements, (S1) and (S2), thereby completing the proof of Theorem 1.

**Step 1: proof of (S1).**   First of all, we observe that

$$R_{\mathcal{D}_n}^{(\lambda)}(y; x) - R_{\nu^*}^{(0)}(y; x) = \underbrace{\left( R_{\mathcal{D}_n}^{(\lambda)}(y; x) - R_{\mathcal{D}_n}^{(0)}(y; x) \right)}_{=:T_1} + \underbrace{\left( R_{\mathcal{D}_n}^{(0)}(y; x) - R_{\nu^*}^{(0)}(y; x) \right)}_{=:T_2}. \qquad (23)$$

We separately analyze the two terms $T_1$ and $T_2$ below to show $T_1 = o_p(1)$ and $T_2 = o_p(1)$ as $n \to \infty$.

(i) $T_1 = o_p(1)$.

Let $\mathcal{D}_n = \{(X_i, Y_i) : i \in [n]\}$, and we re-write

$$T_1 = \frac{1}{n} \sum_{i=1}^{n} \left( w_{\mathcal{D}_n}^{(\lambda)}(X_i, x) - w_{\mathcal{D}_n}^{(0)}(X_i, x) \right) \cdot d^2(Y_i, y).$$

Letting $\widehat{\mu}_n = \mu_{\mathcal{D}_n}$, $\widehat{\Sigma}_n = \Sigma_{\mathcal{D}_n}$, and $\widehat{\Sigma}_n^{(\lambda)} = \mathtt{SVT}^{(\lambda)}(\widehat{\Sigma}_n)$ for shorthand, we observe that

$$w_{\mathcal{D}_n}^{(\lambda)}(X_i, x) - w_{\mathcal{D}_n}^{(0)}(X_i, x) = (X_i - \widehat{\mu}_n)^{\top} \left[ \widehat{\Sigma}_n^{(\lambda),\dagger} - \widehat{\Sigma}_n^{\dagger} \right] (x - \widehat{\mu}_n).$$

Let $\boldsymbol{X} = \begin{bmatrix} X_1 & \cdots & X_n \end{bmatrix}^{\top} \in \mathbb{R}^{n \times p}$, and note that $\widehat{\Sigma}_n = \frac{1}{n} \left( \boldsymbol{X} - \mathbf{1}_n \widehat{\mu}_n^{\top} \right)^{\top} \left( \boldsymbol{X} - \mathbf{1}_n \widehat{\mu}_n^{\top} \right)$. Then it follows that

$$\frac{1}{n} \sum_{i=1}^{n} (X_i - \widehat{\mu}_n)^{\top} \left[ \widehat{\Sigma}_n^{(\lambda),\dagger} - \widehat{\Sigma}_n^{\dagger} \right] = \frac{1}{n} \mathbf{1}_n^{\top} \left( \boldsymbol{X} - \mathbf{1}_n \widehat{\mu}_n^{\top} \right) \left[ \widehat{\Sigma}_n^{(\lambda),\dagger} - \widehat{\Sigma}_n^{\dagger} \right]$$

Consider a singular value decomposition of $\boldsymbol{X} - \mathbf{1}_n \widehat{\mu}_n^{\top}$, namely,

$$\boldsymbol{X} - \mathbf{1}_n \widehat{\mu}_n^{\top} = \sum_{i=1}^{\min\{n,p\}} s_i \cdot u_i v_i^{\top},$$

and observe that $\widehat{\Sigma}_n = \sum_{i=1}^{\min\{n,p\}} s_i^2 \cdot v_i v_i^{\top}$ is an eigenvalue decomposition of $\widehat{\Sigma}_n$. Letting $\mathcal{V}_n^{(\lambda)} := \mathrm{span} \left\{ v_i : i \in [p], \ 0 < s_i \leq \sqrt{\lambda} \right\}$ be a subspace of $\mathbb{R}^p$ spanned by the eigenvectors of $\widehat{\Sigma}_n$ corresponding to the nonzero eigenvalues no greater than $\lambda$, we have

$$\begin{aligned}
\widehat{\Sigma}_n^{(\lambda),\dagger} - \widehat{\Sigma}_n^{\dagger} &= \sum_{i=1}^{p} \frac{1}{s_i^2} \cdot \mathbb{1}\{s_i > \sqrt{\lambda}\} \cdot v_i v_i^{\top} - \sum_{i=1}^{p} \frac{1}{s_i^2} \cdot \mathbb{1}\{s_i > 0\} \cdot v_i v_i^{\top} \\
&= \sum_{i=1}^{p} \frac{1}{s_i^2} \cdot \mathbb{1}\{0 < s_i \leq \sqrt{\lambda}\} \cdot v_i v_i^{\top} \\
&= \widehat{\Sigma}_n^{\dagger} \cdot \Pi_{\mathcal{V}_n^{(\lambda)}} \\
&= n \cdot \left( \boldsymbol{X} - \mathbf{1}_n \widehat{\mu}_n^{\top} \right)^{\dagger} \left( \boldsymbol{X} - \mathbf{1}_n \widehat{\mu}_n^{\top} \right)^{\dagger,\top} \cdot \Pi_{\mathcal{V}_n^{(\lambda)}} \qquad (24)
\end{aligned}$$

where $\Pi_{\mathcal{V}_n^{(\lambda)}}$ denotes the projection matrix onto the subspace $\mathcal{V}_n^{(\lambda)}$. Note that $\Pi_{\mathcal{V}_n^{(\lambda)}} = 0$ if and only if $\min \left\{ i \in [p] : 0 < s_i \leq \sqrt{\lambda} \right\} = \emptyset$.

Therefore, we have

$$\begin{aligned}
T_1 &= \frac{1}{n} \sum_{i=1}^{n} \left( w_{\mathcal{D}_n}^{(\lambda)}(X_i, x) - w_{\mathcal{D}_n}^{(0)}(X_i, x) \right) \cdot d^2(Y_i, y) \\
&\leq \frac{\mathrm{diam}\,(\mathcal{M})^2}{n} \mathbf{1}_n^{\top} \left( \boldsymbol{X} - \mathbf{1}_n \widehat{\mu}_n^{\top} \right) \left[ \widehat{\Sigma}_n^{(\lambda),\dagger} - \widehat{\Sigma}_n^{\dagger} \right] (x - \widehat{\mu}_n) \\
&= \mathrm{diam}\,(\mathcal{M})^2 \cdot \mathbf{1}_n^{\top} \left( \boldsymbol{X} - \mathbf{1}_n \widehat{\mu}_n^{\top} \right)^{\dagger,\top} \cdot \Pi_{\mathcal{V}_n^{(\lambda)}} \cdot (x - \widehat{\mu}_n) \qquad \because (24) \\
&= o_p(1).
\end{aligned}$$

The last line follows from the fact that $\sup_{i \in [p]} \left( \sigma_i(\widehat{\Sigma}_n) - \sigma_i(\Sigma_{\nu^*}) \right) \to 0$ in probability, and thus, $\Pi_{\mathcal{V}_n^{(\lambda)}} \to 0$ in probability.

(ii) $T_2 = o_p(1)$.

Letting $\tilde{R}_n(y;x) = \frac{1}{n}\sum_{i=1}^n w_{\nu^*}^{(0)}(X_i,x) \cdot d^2(Y_i,y)$, we decompose $T_2$ as follows:

$$T_2 = R_{\mathcal{D}_n}^{(0)}(y;x) - \tilde{R}_n(y;x) + \tilde{R}_n(y;x) - R_{\nu^*}^{(0)}(y;x)$$

$$= \underbrace{\frac{1}{n}\sum_{i=1}^n \left\{ w_{\mathcal{D}_n}^{(0)}(X_i,x) - w_{\nu^*}^{(0)}(X_i,x) \right\} \cdot d^2(Y_i,y)}_{=:T_{2A}}$$

$$+ \underbrace{\frac{1}{n}\sum_{i=1}^n \left\{ w_{\nu^*}^{(0)}(X_i,x) \cdot d^2(Y_i,y) - \mathbb{E}\left[ w_{\nu^*}^{(0)}(X_i,x) \cdot d^2(Y_i,y) \right] \right\}}_{=:T_{2B}}$$

Note that $T_{2B}$ converges to $0$ in probability by the weak law of large numbers.

Now it remains to show $T_{2A} = o_p(1)$. To this end, we note that

$$w_{\mathcal{D}_n}^{(0)}(X_i,x) - w_{\nu^*}^{(0)}(X_i,x) = V_n(x) + X_i^\top W_n(x)$$

$$\text{where} \quad \begin{cases} V_n(x) = -\widehat{\mu}_n^\top \widehat{\Sigma}_n^\dagger (x - \widehat{\mu}_n) + \mu^\top \Sigma^\dagger (x - \mu), \\ W_n(x) = \widehat{\Sigma}_n^\dagger (x - \widehat{\mu}_n) - \Sigma^\dagger (x - \mu). \end{cases} \tag{25}$$

Since $\widehat{\mu}_n$ and $\widehat{\Sigma}_n$ respectively converge to $\mu$ and $\Sigma$ in probability, it is possible to verify that $|V_n(x)|, \|W_n(x)\|$ converge to $0$ in probability. As a result, $T_2$ also converges to $0$ in probability.

All in all, we have $R_{\mathcal{D}_n}^{(\lambda)}(y;x) - R_{\nu^*}^{(0)}(y;x) = o_p(1)$, and thus, proved (S1).

**Step 2: proof of (S2).** For any $y_1, y_2 \in \mathcal{M}$,

$$\left| R_{\mathcal{D}_n}^{(\lambda)}(y_1;x) - R_{\mathcal{D}_n}^{(\lambda)}(y_2;x) \right| = \left| \frac{1}{n}\sum_{i=1}^n w_{\mathcal{D}_n}^{(\lambda)}(X_i,x) \cdot \left\{ d^2(Y_i,y_1) - d^2(Y_i,y_2) \right\} \right|$$

$$\leq \frac{1}{n}\sum_{i=1}^n \left| w_{\mathcal{D}_n}^{(\lambda)}(X_i,x) \right| \cdot |d(Y_i,y_1) + d(Y_i,y_2)| \cdot |d(Y_i,y_1) - d(Y_i,y_2)|$$

$$\leq 2\operatorname{diam}(\mathcal{M}) \cdot d(y_1,y_2) \cdot \left( \frac{1}{n}\sum_{i=1}^n \left| w_{\mathcal{D}_n}^{(\lambda)}(X_i,x) \right| \right)$$

$$= O_p\left( d(y_1,y_2) \right)$$

where the $O_p$ term is independent of $y_1, y_2 \in \mathcal{M}$. Therefore,

$$\sup_{y_1,y_2 \in \mathcal{M}:\, d(y_1,y_2) < \delta} \left| R_{\mathcal{D}_n}^{(\lambda)}(y_1;x) - R_{\mathcal{D}_n}^{(\lambda)}(y_2;x) \right| = O_p(\delta),$$

which proves (S2).

$\square$

## C   Proof of Theorem 2

In this section, we prove the two claims in Theorem 2. Specifically, in Section C.1, we present and prove a lemma that controls the bias in the population estimator (Lemma 1), and in Section C.2, we present and prove a lemma that controls the variance of the empirical estimator (Lemma 2).

### C.1   Bias in the population estimator

We recall the definition of Mahalanobis seminorm from Definition 5: $\|x\|_\Sigma := \left( x^\top \Sigma^\dagger x \right)^{1/2}$.

**Lemma 1.** *Suppose that Assumptions (C0) and (C1) hold. If*

$$\|x - \mu_{\nu^*}\|_{\Sigma_{\nu^*}} \leq \frac{C_g \cdot D_g^\alpha}{\operatorname{diam}(\mathcal{M})^2 \cdot \sqrt{\operatorname{rank}\Sigma_{\nu^*}}}, \tag{26}$$

*then for any $\lambda \in \mathbb{R}_+$,*

$$d\big(\varphi^{(\lambda)}(x), \varphi(x)\big) \leq 2^{K_0} \cdot \mathfrak{b}_\lambda(x)^{\frac{1}{\alpha-1}} = O\Big(\mathfrak{b}_\lambda(x)^{\frac{1}{\alpha-1}}\Big) \tag{27}$$

*where*

$$K_0 = \left\lfloor \frac{1}{(\alpha-1)\log 2} \cdot \log\left(\frac{4\operatorname{diam}(\mathcal{M})}{C_g \cdot \big(1 - 2^{-(\alpha-1)}\big)}\right) \right\rfloor + 1 \quad and$$

$$\mathfrak{b}_\lambda(x) = \sqrt{\operatorname{rank}\big(\Sigma_{\nu^*} - \Sigma_{\nu^*}^{(\lambda)}\big)} \cdot \|x - \mu_{\nu^*}\|_{\Sigma_{\nu^*} - \Sigma_{\nu^*}^{(\lambda)}}.$$

*Proof of Lemma 1.* For the sake of brevity, we write $\varphi^{(\lambda)}(x) = \varphi_{\nu^*}^{(\lambda)}(x)$ and $\varphi(x) = \varphi_{\nu^*}^{(0)}(x)$ throughout this proof, dropping the subscript $\nu^*$. Likewise, we simply write $\mu = \mu_{\nu^*}$ and $\Sigma = \Sigma_{\nu^*}$.

**Step 1: A naïve upper bound.** Observe that for any $\lambda \in \mathbb{R}_+$, $x \in \mathbb{R}^p$, and $y \in \mathcal{M}$,

$$\begin{aligned}
\big| R(y; x) &- R^{(\lambda)}(y; x) \big| \\
&= \left| \mathbb{E}_{\nu^*}\left[ (X - \mu)^\top \cdot \big(\Sigma^\dagger - \Sigma^{(\lambda),\dagger}\big) \cdot (x - \mu) \cdot d^2(Y, y) \right] \right| \\
&\leq \operatorname{diam}(\mathcal{M})^2 \cdot \mathbb{E}_{\nu^*}\left[\|X - \mu\|_{\Sigma - \Sigma^{(\lambda)}}\right] \cdot \|x - \mu\|_{\Sigma - \Sigma^{(\lambda)}} \qquad \because \text{Cauchy-Schwarz inequality} \\
&\leq \operatorname{diam}(\mathcal{M})^2 \cdot \left(\mathbb{E}_{\nu^*}\|X - \mu\|_{\Sigma - \Sigma^{(\lambda)}}^2\right)^{1/2} \cdot \|x - \mu\|_{\Sigma - \Sigma^{(\lambda)}} \quad \because \text{Jensen's inequality} \\
&= \operatorname{diam}(\mathcal{M})^2 \cdot \sqrt{\operatorname{rank}\big(\Sigma - \Sigma^{(\lambda)}\big)} \cdot \|x - \mu\|_{\Sigma - \Sigma^{(\lambda)}}, \tag{28}
\end{aligned}$$

where the last inequality follows from $\mathbb{E}_{\nu^*}\|X - \mu\|_{\Sigma - \Sigma^{(\lambda)}}^2 = \operatorname{rank}\big(\Sigma - \Sigma^{(\lambda)}\big)$.

We observe that the upper bound in (28) is monotone non-decreasing with respect to $\lambda \in \mathbb{R}_+$, and it converges to 0 as $\lambda \to 0$. To see this, for any $\lambda \in \mathbb{R}_+$, we let

$$\mathcal{V}^{(\lambda)} := \operatorname{span}\{v_i : i \in [p],\ 0 < \lambda_i \leq \lambda\}$$

where $\Sigma = \sum_{i=1}^p \lambda_i \cdot v_i v_i^\top$ is an eigenvdecomposition of $\Sigma$. Letting $\Pi_{\mathcal{V}^{(\lambda)}}$ denote the projection matrix onto the subspace $\mathcal{V}^{(\lambda)}$, we note that $\Sigma - \Sigma^{(\lambda)} = \Pi_{\mathcal{V}^{(\lambda)}} \Sigma \Pi_{\mathcal{V}^{(\lambda)}}$, and that $(\Sigma - \Sigma^{(\lambda)})^\dagger = \Pi_{\mathcal{V}^{(\lambda)}} \Sigma^\dagger \Pi_{\mathcal{V}^{(\lambda)}}$. Thus, $\operatorname{rank}\big(\Sigma - \Sigma^{(\lambda)}\big) = \dim \mathcal{V}^{(\lambda)}$, and furthermore, we notice that $\mathcal{V}^{(\lambda)} = \{0\}$ if and only if $\lambda < \lambda_{\min} := \min\{\lambda_i : \lambda_i > 0\}$. Therefore,

$$\lambda < \lambda_{\min} \qquad \Longrightarrow \qquad R^{(\lambda)}(y; x) - R(y; x) = 0 \qquad \Longrightarrow \qquad \varphi^{(\lambda)}(x) = \varphi(x),\ \forall x. \tag{29}$$

The observation (29), together with Assumption (C0), implies that $d\big(\varphi^{(\lambda)}(x), \varphi(x)\big) = o(1)$ as $\lambda \to 0$.

**Step 2: Controlling risk difference.** Next, we move on to determine the order of $d\big(\varphi^{(\lambda)}(x), \varphi(x)\big)$ — as a function of $\mathfrak{b}_\lambda(x)$ — for a fixed $\lambda \in \mathbb{R}$. We may assume $\lambda > \lambda_{\min}$ for the proof because the lemma is trivial otherwise, cf. (29). Assuming $\lambda > \lambda_{\min}$, we may decompose the difference in the population objective at $\varphi^{(\lambda)}(x)$ and $\varphi(x)$ as follows:

$$\begin{aligned}
R\big(\varphi^{(\lambda)}(x); x\big) - R\big(\varphi(x); x\big) = &\underbrace{\Big\{ R\big(\varphi^{(\lambda)}(x); x\big) - R^{(\lambda)}\big(\varphi^{(\lambda)}(x); x\big) + R^{(\lambda)}\big(\varphi(x); x\big) - R\big(\varphi(x); x\big) \Big\}}_{=:\mathfrak{R}_1} \\
&- \underbrace{\Big\{ R^{(\lambda)}\big(\varphi(x); x\big) - R^{(\lambda)}\big(\varphi^{(\lambda)}(x); x\big) \Big\}}_{=:\mathfrak{R}_2}.
\end{aligned}$$

We observe that both $\mathfrak{R}_1$ and $\mathfrak{R}_2$ are non-negative, due to the optimality of $\varphi(x)$ and $\varphi^{(\lambda)}(x)$. Then, we obtain an upper bound for $\mathfrak{R}_1$ using a similar argument as in (28). Specifically,

$$R\big(\varphi^{(\lambda)}(x);x\big) - R\big(\varphi(x);x\big) \leq \mathfrak{R}_1$$
$$= \mathbb{E}_{\nu^*}\left[\left\{w_{\nu^*}^{(0)}(X,x) - w_{\nu^*}^{(\lambda)}(X,x)\right\} \cdot \left\{d^2\big(Y,\varphi^{(\lambda)}(x)\big) - d^2\big(Y,\varphi(x)\big)\right\}\right]$$
$$\leq 2\,\mathrm{diam}\,(\mathcal{M}) \cdot \mathsf{b}_\lambda(x) \cdot d\big(\varphi^{(\lambda)}(x),\varphi(x)\big). \tag{30}$$

**Step 3: Converting risk difference to bias.** Lastly, we convert the upper bound (30) to an upper bound on the distance $d\big(\varphi^{(\lambda)}(x),\varphi(x)\big)$ using Assumption (C1). To this end, we begin by confirming that

$$R\big(\varphi^{(\lambda)}(x);x\big) - R\big(\varphi(x);x\big) = \mathbb{E}_{\nu^*}\left[(X-\mu)^\top \cdot \Sigma^\dagger \cdot (x-\mu) \cdot \left\{d^2\big(Y,\varphi^{(\lambda)}(x)\big) - d^2\big(Y,\varphi(x)\big)\right\}\right]$$
$$\leq \mathrm{diam}\,(\mathcal{M})^2 \cdot \left(\mathbb{E}_{\nu^*}\,\|X-\mu\|_\Sigma^2\right)^{1/2} \cdot \|x-\mu\|_\Sigma$$
$$= \mathrm{diam}\,(\mathcal{M})^2 \cdot \sqrt{\mathrm{rank}\,\Sigma} \cdot \|x-\mu\|_\Sigma$$
$$\leq C_{\mathsf{g}} \cdot D_{\mathsf{g}}^\alpha.$$

Thereafter, we choose an arbitrary $K \in \mathbb{N}$ and $r \in \mathbb{R}_+$ whose values will be determined later in this proof. Then we obtain the following inequality using the so-called peeling technique:

$$\mathbb{1}\left\{d\big(\varphi^{(\lambda)}(x),\varphi(x)\big) > 2^K \cdot \mathsf{b}_\lambda(x)^r\right\}$$
$$= \sum_{k=K}^{\infty} \mathbb{1}\left\{2^k \cdot \mathsf{b}_\lambda(x)^r < d\big(\varphi^{(\lambda)}(x),\varphi(x)\big) \leq 2^{k+1} \cdot \mathsf{b}_\lambda(x)^r\right\}$$
$$\leq \sum_{k=K}^{\infty} \mathbb{1}\left\{2^k \cdot \mathsf{b}_\lambda(x)^r < d\big(\varphi^{(\lambda)}(x),\varphi(x)\big) \leq 2^{k+1} \cdot \mathsf{b}_\lambda(x)^r\right\}$$
$$\leq \sum_{k=K}^{\infty} \frac{R\big(\varphi^{(\lambda)}(x);x\big) - R\big(\varphi(x);x\big)}{C_{\mathsf{g}} \cdot \big(2^k \cdot \mathsf{b}_\lambda(x)^r\big)^\alpha} \cdot \mathbb{1}\left\{d\big(\varphi^{(\lambda)}(x),\varphi(x)\big) \leq 2^{k+1} \cdot \mathsf{b}_\lambda(x)^r\right\}. \quad \because (C1) \tag{31}$$

Moreover, we decompose the numerator in the fraction appearing in the upper bound (31) as follows: Combining (30) with (31), we have

$$\mathbb{1}\left\{d\big(\varphi^{(\lambda)}(x),\varphi(x)\big) > 2^K \cdot \mathsf{b}_\lambda(x)^r\right\}$$
$$\leq \sum_{k=K}^{\infty} \frac{2\,\mathrm{diam}\,(\mathcal{M}) \cdot \mathsf{b}_\lambda(x) \cdot d\big(\varphi^{(\lambda)}(x),\varphi(x)\big)}{C_{\mathsf{g}} \cdot \big(2^k \cdot \mathsf{b}_\lambda(x)^r\big)^\alpha} \cdot \mathbb{1}\left\{d\big(\varphi^{(\lambda)}(x),\varphi(x)\big) \leq 2^{k+1} \cdot \mathsf{b}_\lambda(x)^r\right\}$$
$$\leq \frac{4\,\mathrm{diam}\,(\mathcal{M})}{C_{\mathsf{g}}} \cdot \mathsf{b}_\lambda(x)^{1-r(\alpha-1)} \sum_{k=K}^{\infty} \frac{1}{2^{k(\alpha-1)}}. \tag{32}$$

Note that $C := \frac{4\,\mathrm{diam}\,\mathcal{M}}{C_{\mathsf{g}}} > 0$ is a constant independent of $\lambda$. Let $r = 1/(\alpha-1)$, and observe that the upper bound in (32) becomes smaller than 1 for a sufficiently large $K$. Specifically,

$$K \geq \left\lfloor \frac{1}{(\alpha-1)\log 2} \cdot \log\left(\frac{4\,\mathrm{diam}\,(\mathcal{M})}{C_{\mathsf{g}} \cdot \big(1-2^{-(\alpha-1)}\big)}\right)\right\rfloor + 1 \quad \Longrightarrow \quad \frac{4\,\mathrm{diam}\,(\mathcal{M})}{C_{\mathsf{g}}} \cdot \sum_{k=K}^{\infty} \frac{1}{2^{k(\alpha-1)}} < 1.$$

As a result, the inequality "$d\big(\varphi^{(\lambda)}(x),\varphi(x)\big) > 2^{K_0} \cdot \mathsf{b}_\lambda(x)^r$" in the indicator function must be false, and we conclude that

$$d\big(\varphi^{(\lambda)}(x),\varphi(x)\big) \leq 2^{K_0} \cdot \mathsf{b}_\lambda(x)^{\frac{1}{\alpha-1}}.$$

$\square$

## C.2 Variance of the empirical estimator

**Lemma 2.** *Suppose that Assumptions (C0), (C1) and (C2) hold. For any $\lambda \in \mathbb{R}_+$ such that $\lambda \notin \mathrm{spec}\left(\Sigma_{\nu^*}\right)$, it holds that*

$$d\left(\varphi_{\mathcal{D}_n}^{(\lambda)}(x), \varphi_{\nu^*}^{(\lambda)}(x)\right) = O_P\left(n^{-\frac{1}{2(\alpha-1)}}\right).$$

*Proof of Lemma 2.* Recall from the definition of $\lambda$-regularized Fréchet regression (Definition 4) and (9) that

$$R_{\mathcal{D}_n}^{(\lambda)}(y;x) = \frac{1}{n}\sum_{i=1}^{n} w_{\mathcal{D}_n}^{(\lambda)}(X_i,x) \cdot d^2(Y_i,y) \quad \text{and} \quad R_{\nu^*}^{(\lambda)}(y;x) = \mathbb{E}_{(X,Y)\sim\nu^*}\left[w_{\nu^*}^{(\lambda)}(X,x) \cdot d^2(Y,y)\right].$$

Additionally, we define an auxiliary function $\tilde{R}_n(y;x)$ as the "empirical risk with population weight" such that

$$\tilde{R}_n(y;x) := \frac{1}{n}\sum_{i=1}^{n} w_{\nu^*}^{(\lambda)}(X_i,x) \cdot d^2(Y_i,y).$$

We present the rest of this proof in three steps, outlined as follows. In Step 1, we show the consistency of $\varphi_{\mathcal{D}_n}^{(\lambda)}(x)$, i.e., $d\left(\varphi_{\mathcal{D}_n}^{(\lambda)}(x), \varphi_{\nu^*}^{(\lambda)}(x)\right) = o_P(1)$ as $n \to \infty$. In Step 2, we define the discrepancy variable $Z_n^{(\lambda)}(y;x) := R_{\mathcal{D}_n}^{(\lambda)}(y;x) - R_{\nu^*}^{(\lambda)}(y;x)$ between the finite-sample and the population objectives, cf. (35), and prove a uniform upper bound for $Z_n^{(\lambda)}(y;x)$ that holds in a neighborhood of $\varphi_{\nu^*}^{(\lambda)}(y;x)$. Lastly, in Step 3, we utilize the peeling technique from empirical process theory to obtain the desired rate of convergence.

**Step 1: Consistency.** We first claim that $d\left(\varphi_{\mathcal{D}_n}^{(\lambda)}(x), \varphi_{\nu^*}^{(\lambda)}(x)\right) = o_P(1)$ by an argument similar to that used in the proof of Theorem 1. Specifcally, it suffices to show that

(S1') $R_{\mathcal{D}_n}^{(\lambda)}(y;x) - R_{\nu^*}^{(\lambda)}(y;x) = o_P(1)$, and

(S2') $R_{\mathcal{D}_n}^{(\lambda)}(\cdot;x) : \mathcal{M} \to \mathbb{R}$ is asymptotically equicontinuous in probability.

Note that we already showed the asymptotic equicontinuity in the proof of Theorem 1; see (S2). Thus, it remains to show the pointwise convergence in probability. To show (S1'), we decompose $R_{\mathcal{D}_n}^{(\lambda)}(y;x) - R_{\nu^*}^{(\lambda)}(y;x)$ as follows.

$$R_{\mathcal{D}_n}^{(\lambda)}(y;x) - R_{\nu^*}^{(\lambda)}(y;x) = \left\{R_{\mathcal{D}_n}^{(\lambda)}(y;x) - \tilde{R}_n(y;x)\right\} + \left\{\tilde{R}_n(y;x) - R_{\nu^*}^{(\lambda)}(y;x)\right\}$$

$$= \underbrace{\frac{1}{n}\sum_{i=1}^{n}\left\{w_{\mathcal{D}_n}^{(\lambda)}(X_i,x) - w_{\nu^*}^{(\lambda)}(X_i,x)\right\} \cdot d^2(Y_i,y)}_{:=A_n^{(\lambda)}(y;x)}$$

$$+ \underbrace{\frac{1}{n}\sum_{i=1}^{n}\left(w_{\nu^*}^{(\lambda)}(X_i,x) \cdot d^2(Y_i,y) - \mathbb{E}_{\nu^*}\left[w_{\nu^*}^{(\lambda)}(X_i,x) \cdot d^2(Y_i,y)\right]\right)}_{:=B_n^{(\lambda)}(y;x)}.$$

Next, we show that $A_n^{(\lambda)}(y;x)$ and $B_n^{(\lambda)}(y;x)$ respectively converge to 0 in probability.

- Letting $\widehat{\mu}_n = \mu_{\mathcal{D}_n}$, $\widehat{\Sigma}_n = \Sigma_{\mathcal{D}_n}$, and $\widehat{\Sigma}_n^{(\lambda)} = \mathtt{SVT}^{(\lambda)}(\widehat{\Sigma}_n)$ for shorthand, we can write
$$w_{\mathcal{D}_n}^{(\lambda)}(X_i,x) - w_{\nu^*}^{(\lambda)}(X_i,x) = V_n^{(\lambda)}(x) + X_i^\top W_n^{(\lambda)}(x),$$
similarly to (25), where
$$V_n^{(\lambda)}(x) = -\widehat{\mu}_n^\top\left[\widehat{\Sigma}_n^{(\lambda)}\right]^\dagger(x-\widehat{\mu}_n) + \mu^\top\left[\Sigma^{(\lambda)}\right]^\dagger(x-\mu),$$
$$W_n^{(\lambda)}(x) = \left[\widehat{\Sigma}_n^{(\lambda)}\right]^\dagger(x-\widehat{\mu}_n) - \left[\Sigma^{(\lambda)}\right]^\dagger(x-\mu). \tag{33}$$

Since $\|\widehat{\mu}_n - \mu\|_2 = O_P(n^{-1/2})$ and $\|\widehat{\Sigma}_n^{(\lambda)} - \Sigma^{(\lambda)}\| = O_P(n^{-1/2})$ (if $\lambda \notin \operatorname{spec}\Sigma$) independent of $\lambda > 0$, we also have $|V_n^{(\lambda)}(x)| = O_P(n^{-1/2})$ and $\|W_n^{(\lambda)}(x)\|_2 = O_P(n^{-1/2})$. This implies that $A_n^{(\lambda)}(y;x) = o_P(1)$.

- Moreover, we note that if $\|x - \mu\|_\Sigma < \infty$, then the random variable $w_{\nu^*}^{(\lambda)}(X,x)$ has finite second moment

$$
\begin{aligned}
\mathbb{E}_{\nu^*}\left[w_{\nu^*}^{(\lambda)}(X,x)^2\right] &\leq 2\left(1 + \mathbb{E}_{\nu^*}\left[\left|(X-\mu)^\top[\Sigma^{(\lambda)}]^\dagger(x-\mu)\right|^2\right]\right) \\
&\leq 2\left(1 + \mathbb{E}_{\nu^*}\left[\|X-\mu\|_{\Sigma^{(\lambda)}}^2 \cdot \|x-\mu\|_{\Sigma^{(\lambda)}}^2\right]\right) \\
&\leq 2\{1 + p\,\|x-\mu\|_\Sigma^2\},
\end{aligned}
\tag{34}
$$

regardless of the value of $\lambda > 0$. When $\operatorname{diam}(\mathcal{M}) < \infty$, the product $w_{\nu^*}^{(\lambda)}(X,x) \cdot d^2(Y,y)$ also has finite second moment. Since $B_n^{(\lambda)}(y;x)$ is the sample mean of IID random variables with mean zero and finite variance, it follows that

$$
B_n^{(\lambda)}(y;x) = O_P\left(\sqrt{\frac{\operatorname{Var}\left[w_{\nu^*}^{(\lambda)}(X_1,x) \cdot d^2(Y_1,y)\right]}{n}}\right) = O_P\left(n^{-1/2}\right).
$$

**Step 2: Uniform control of the fluctuation in objective discrepancy.** For any $\lambda \in \mathbb{R}_+$ and any $(x,y) \in \mathbb{R}^p \times \mathcal{M}$, we let $Z_n^{(\lambda)}(y;x)$ denote the random variable defined as

$$
Z_n^{(\lambda)}(y;x) := R_{\mathcal{D}_n}^{(\lambda)}(y;x) - R_{\nu^*}^{(\lambda)}(y;x)
\tag{35}
$$

We observed that

$$
\begin{aligned}
&Z_n^{(\lambda)}\big(y;x\big) - Z_n^{(\lambda)}\big(\varphi_{\nu^*}^{(\lambda)}(x);x\big) \\
&= \left\{R_{\mathcal{D}_n}^{(\lambda)}(y;x) - R_{\nu^*}^{(\lambda)}(y;x)\right\} - \left\{R_{\mathcal{D}_n}^{(\lambda)}\big(\varphi_{\nu^*}^{(\lambda)}(x);x\big) - R_{\nu^*}^{(\lambda)}\big(\varphi_{\nu^*}^{(\lambda)}(x);x\big)\right\} \\
&= \left[\left\{R_{\mathcal{D}_n}^{(\lambda)}(y;x) - \tilde{R}_n(y;x)\right\} - \left\{R_{\mathcal{D}_n}^{(\lambda)}\big(\varphi_{\nu^*}^{(\lambda)}(x);x\big) - \tilde{R}_n\big(\varphi_{\nu^*}^{(\lambda)}(x);x\big)\right\}\right] \\
&\quad + \left[\left\{\tilde{R}_n(y;x) - R_{\nu^*}^{(\lambda)}(y;x)\right\} - \left\{\tilde{R}_n\big(\varphi_{\nu^*}^{(\lambda)}(x);x\big) - R_{\nu^*}^{(\lambda)}\big(\varphi_{\nu^*}^{(\lambda)}(x);x\big)\right\}\right] \\
&= \underbrace{\frac{1}{n}\sum_{i=1}^n \left\{w_{\mathcal{D}_n}^{(\lambda)}(X_i,x) - w_{\nu^*}^{(\lambda)}(X_i,x)\right\} \cdot \ell_i^{(\lambda)}(y;x)}_{=:\mathfrak{A}_n^{(\lambda)}(y;x)} \\
&\quad + \underbrace{\frac{1}{n}\sum_{i=1}^n \left(w_{\nu^*}^{(\lambda)}(X_i,x) \cdot \ell_i^{(\lambda)}(y;x) - \mathbb{E}_{\nu^*}\left[w_{\nu^*}^{(\lambda)}(X_i,x) \cdot \ell_i^{(\lambda)}(y;x)\right]\right)}_{=:\mathfrak{B}_n^{(\lambda)}(y;x)}
\end{aligned}
\tag{36}
$$

where $\ell_i^{(\lambda)}(y;x) := d^2\big(Y_i,y\big) - d^2\big(Y_i,\varphi_{\nu^*}^{(\lambda)}(x)\big)$.

Next, we analyze the asymptotic behavior of the two terms, $\mathfrak{A}_n^{(\lambda)}(y;x)$ and $\mathfrak{B}_n^{(\lambda)}(y;x)$. Specifically, we establish upper bounds on their magnitudes that hold uniformly over a $\delta$-neighborhood of $\varphi^{(\lambda)}(x) = \varphi_{\nu^*}^{(\lambda)}(x)$, which will be used later in Step 3 of this proof.

- Firstly, we observe that for any $\delta > 0$,

$$\sup_{y \in B_d\left(\varphi_{\nu^*}^{(\lambda)}(x);\,\delta\right)} \left|\mathfrak{A}_n^{(\lambda)}(y;x)\right|$$

$$\leq \frac{1}{n}\sum_{i=1}^{n}\left|w_{\mathcal{D}_n}^{(\lambda)}(X_i,x) - w_{\nu^*}^{(\lambda)}(X_i,x)\right| \cdot \sup_{y \in B_d\left(\varphi_{\nu^*}^{(\lambda)}(x);\,\delta\right)} \left|d^2(Y_i,y) - d^2(Y_i,\varphi_{\nu^*}^{(\lambda)}(x))\right|$$

$$\leq 2\operatorname{diam}(\mathcal{M}) \cdot \left\{ \frac{1}{n}\sum_{i=1}^{n}\left\{ \left|V_n^{(\lambda)}(x)\right| + \|X_i\|_2\, \|W_n^{(\lambda)}(x)\|_2 \right\} \right\}$$

$$\times \sup_{y \in B_d\left(\varphi_{\nu^*}^{(\lambda)}(x);\,\delta\right)} d\left(y,\varphi_{\nu^*}^{(\lambda)}(x)\right)$$

$$= O_P\left(\delta \cdot n^{-1/2}\right), \tag{37}$$

where we used the property of $V_n^{(\lambda)}(x)$ and $W_n^{(\lambda)}(x)$ discussed in the paragraph following (33). Since the stochastic magnitudes of $V_n^{(\lambda)}(x)$ and $W_n^{(\lambda)}(x)$ are independent of $\delta$, (37) implies that there exists $C_1^{(\lambda)} = C_1^{(\lambda)}(x) > 0$ such that for any $\delta > 0$,

$$\liminf_{n\to\infty} P\left( \sup_{y \in \mathcal{M}} \left\{ |\mathfrak{A}_n^{(\lambda)}(y;x)| : d\left(y,\varphi_{\nu^*}^{(\lambda)}(x)\right) < \delta \right\} \leq C_1^{(\lambda)} \cdot \delta \cdot n^{-1/2} \right) = 1. \tag{38}$$

Furthermore, for any $\gamma, \delta \in \mathbb{R}_+$ such that $0 \leq \gamma < \delta$, let $\mathfrak{E}_n^{(\lambda)}(\gamma,\delta;x)$ be defined as an event such that

$$\mathfrak{E}_n(\gamma,\delta;x) = \left( \sup_{y \in \mathcal{M}} \left\{ |\mathfrak{A}_n^{(\lambda)}(y;x)| : d\left(y,\varphi_{\nu^*}^{(\lambda)}(x)\right) \in [\gamma,\delta) \right\} \leq C_1^{(\lambda)} \cdot \delta \cdot n^{-1/2} \right). \tag{39}$$

For any $\gamma \in [0,\delta]$, we have $\mathfrak{E}_n(0,\delta;x) \subseteq \mathfrak{E}_n(\gamma,\delta;x)$, and thus, $\liminf_{n\to\infty} P\left(\mathfrak{E}_n(\gamma,\delta;x)\right) = 1$.

- Next, we note that

$$\left|w_{\nu^*}^{(\lambda)}(X_i,x) \cdot \ell_i^{(\lambda)}(y;x)\right| \leq 2\operatorname{diam}(\mathcal{M}) \cdot d\left(y,\varphi_{\nu^*}^{(\lambda)}(x)\right) \cdot \left|w_{\nu^*}^{(\lambda)}(X_i,x)\right|.$$

Observe that $d\left(y,\varphi_{\nu^*}^{(\lambda)}(x)\right) \leq \operatorname{diam}(\mathcal{M}) < \infty$ and recall that $\mathbb{E}_{\nu^*}\left[w_{\nu^*}^{(\lambda)}(X,x)^2\right] \leq 2\left\{1 + p\,\|x-\mu\|_\Sigma^2\right\}$ as shown in Step 1 of this proof, cf. (34). It follows from the uniform entropy condition (C2), Theorem 2.7.11, and Theorem 2.14.2 in [55] that there exists $D_e = D_e(x) > 0$ such that for all $\delta \in [0, D_e)$,

$$\mathbb{E}\left[ \sup_{y \in \mathcal{M}} \left\{ |\mathfrak{B}_n^{(\lambda)}(y;x)| : d\left(y,\varphi_{\nu^*}^{(\lambda)}(x)\right) < \delta \right\} \right]$$

$$\leq 2\operatorname{diam}(\mathcal{M}) \cdot \delta \cdot n^{-1/2}\sqrt{1 + p\,\|x-\mu\|_\Sigma^2} \int_0^1 \sqrt{1 + \log \mathfrak{N}\left(B_d(\varphi^{(\lambda)}(x);\delta), \delta\epsilon\right)}\, d\epsilon$$

$$\leq C_2^{(\lambda)} \cdot \delta \cdot n^{-1/2} \tag{40}$$

where $C_2^{(\lambda)} = 2\,(C_e + 1) \cdot \operatorname{diam}(\mathcal{M}) \cdot \sqrt{1 + p\,\|x-\mu\|_\Sigma^2}$ is independent of $\delta > 0$ and $n \geq 1$.

**Step 3: Concluding the proof.** Lastly, we combine the results from Steps 1-2 to show that, for any $\eta > 0$, there exist $K = K(\eta) > 0$ and $N = N(\eta) \geq 1$ such that $P\left(d\left(\varphi_{\mathcal{D}_n}^{(\lambda)}(x), \varphi_{\nu^*}^{(\lambda)}(x)\right) > 2^K\, n^{-\beta}\right) < \eta$ for any $n \geq N$, where $\beta > 0$ is an absolute constant that will be determined later in this proof. We prove this claim using the peeling technique, in a similar manner as we did in the proof of Lemma 1. To avoid cluttered notation, we let $\Delta(x) = d\left(\varphi_{\mathcal{D}_n}^{(\lambda)}(x), \varphi_{\nu^*}^{(\lambda)}(x)\right)$ in the rest of this proof.

For any fixed $K \in \mathbb{N}$ and a sufficiently large $n = n(K) \geq 1$ satisfying $2^K n^{-\beta} < D_* := D_{\mathrm{g}} \wedge D_{\mathrm{e}}$, we observe that

$$P\Big(\Delta(x) > 2^K n^{-\beta}\Big) = P\Big(\Delta(x) \geq D_*\Big) + P\Big(2^K n^{-\beta} \leq \Delta(x) < D_*\Big) \tag{41}$$

where we used $P(A) \leq P(B^c) + P(A \cap B)$ to get the inequality. As we know that $P\Big(\Delta(x) \geq D_*\Big) = o(1)$ by Step 1 of this proof, we focus on showing an upper bound for the other term, $P\big(2^K n^{-\beta} \leq \Delta(x) < D_*\big)$.

*Step 3-A: Decomposition of $P\big(2^K n^{-\beta} \leq \Delta(x) < D_*\big)$.* For each $n, k \in \mathbb{N}$, we define

$$
\begin{aligned}
\mathfrak{F}_{n,k} &= \bigcap_{k'=K}^{k} \mathfrak{E}_n^{(\lambda)}\big(2^{k'} n^{-\beta}, 2^{k'+1} n^{-\beta} \wedge D_*; x\big), \\
\mathfrak{G}_{n,k} &= \Bigg( \bigcap_{k'=K}^{k-1} \mathfrak{E}_n^{(\lambda)}\big(2^{k'} n^{-\beta}, 2^{k'+1} n^{-\beta} \wedge D_*; x\big) \Bigg) \cap \mathfrak{E}_n^{(\lambda)}\big(2^{k} n^{-\beta}, 2^{k+1} n^{-\beta} \wedge D_*; x\big)^c,
\end{aligned}
\tag{42}
$$

where we set $\mathfrak{F}_{n,K-1}$ to be the entire event space so that $\mathfrak{G}_{n,K} = \big(\mathfrak{F}_{n,K}\big)^c$. It is worth mentioning that $\mathfrak{G}_{n,k}$ and $\mathfrak{G}_{n,k'}$ are mutually exclusive for any $k \neq k' \geq K$, and we will use this property when concluding the proof in Step 3-C below.

Now, we observe that

$$
\begin{aligned}
P\Big(2^K &n^{-\beta} \leq \Delta(x) < D_*\Big) \\
&\leq P\Big(\mathfrak{E}_n^{(\lambda)}\big(2^K n^{-\beta}, 2^{K+1} n^{-\beta} \wedge D_*; x\big)^c\Big) \\
&\quad + P\bigg(\Big(2^K n^{-\beta} \leq \Delta(x) < D_*\Big) \cap \mathfrak{E}_n^{(\lambda)}\big(2^{k'} n^{-\beta}, 2^{k'+1} n^{-\beta} \wedge D_*; x\big)\bigg) \\
&= P\big(\mathfrak{G}_{n,K}\big) + P\bigg(\Big(2^K n^{-\beta} \leq \Delta(x) < D_*\Big) \cap \mathfrak{F}_{n,K}\bigg) \\
&= P\big(\mathfrak{G}_{n,K}\big) + P\bigg(\Big(2^K n^{-\beta} \leq \Delta(x) < 2^{K+1} n^{-\beta} \wedge D_*\Big) \cap \mathfrak{F}_{n,K}\bigg) \\
&\quad + P\bigg(\Big(2^{K+1} n^{-\beta} \leq \Delta(x) < D_*\Big) \cap \mathfrak{F}_{n,K}\bigg)
\end{aligned}
$$

and that for every $k \geq K$,

$$P\bigg(\Big(2^{k+1} n^{-\beta} \leq \Delta(x) < D_*\Big) \cap \mathfrak{F}_{n,k}\bigg) \leq P\bigg(\Big(2^{k+1} n^{-\beta} \leq \Delta(x) < D_*\Big) \cap \mathfrak{F}_{n,k+1}\bigg) + P\big(\mathfrak{G}_{n,k+1}\big).$$

As a result, we have

$$P\Big(2^K n^{-\beta} \leq \Delta(x) < D_*\Big) = \sum_{k=K}^{\infty} P\big(\mathfrak{G}_{n,k}\big) + \sum_{k=K}^{\infty} \underbrace{P\bigg(\Big(2^k n^{-\beta} \leq \Delta(x) < 2^{k+1} n^{-\beta} \wedge D_*\Big) \cap \mathfrak{F}_{n,k}\bigg)}_{=: \mathfrak{C}_{n,k}}.$$

$$\tag{43}$$

*Step 3-B: Controlling $\mathfrak{C}_{n,k}$.* Next, we show an upper bound for $\mathfrak{C}_{n,k}$. Suppose that $2^k n^{-\beta} \leq \Delta(x) < 2^{k+1} n^{-\beta} \wedge D_*$ and the event $\mathfrak{F}_{n,k}$ occurs. Then it follows from Assumption (C1) that

$$
\begin{aligned}
C_{\mathbf{g}} &\cdot \Delta(x)^\alpha \\
&\leq R_{\nu^*}^{(\lambda)}\big(\varphi_{\mathcal{D}_n}^{(\lambda)}(x); x\big) - R_{\nu^*}^{(\lambda)}\big(\varphi_{\nu^*}^{(\lambda)}(x); x\big) \\
&\leq \left\{ R_{\nu^*}^{(\lambda)}\big(\varphi_{\mathcal{D}_n}^{(\lambda)}(x); x\big) - R_{\nu^*}^{(\lambda)}\big(\varphi_{\nu^*}^{(\lambda)}(x); x\big) \right\} + \underbrace{\left\{ R_{\mathcal{D}_n}^{(\lambda)}\big(\varphi_{\nu^*}^{(\lambda)}(x); x\big) - R_{\mathcal{D}_n}^{(\lambda)}\big(\varphi_{\mathcal{D}_n}^{(\lambda)}(x); x\big) \right\}}_{\geq 0} \\
&= Z_n^{(\lambda)}\big(\varphi_{\nu^*}^{(\lambda)}; x\big) - Z_n^{(\lambda)}\big(\varphi_{\mathcal{D}_n}^{(\lambda)}(x); x\big) \qquad\qquad\qquad\qquad\quad \text{cf. (35)} \\
&\leq \left| \mathfrak{A}_n^{(\lambda)}\big(\varphi_{\nu^*}^{(\lambda)}(x); x\big) \right| + \left| \mathfrak{B}_n^{(\lambda)}\big(\varphi_{\nu^*}^{(\lambda)}(x); x\big) \right| \qquad\qquad\qquad\quad \because \text{(36)} \\
&\leq \sup_{y \in \mathcal{M}} \left\{ \left| \mathfrak{A}_n^{(\lambda)}(y; x) \right| + \left| \mathfrak{B}_n^{(\lambda)}(y; x) \right| : 2^k n^{-\beta} \leq d\big(y, \varphi_{\nu^*}^{(\lambda)}(x)\big) < 2^{k+1} n^{-\beta} \wedge D_* \right\} \\
&\leq C_1^{(\lambda)} \cdot \big(2^{k+1} n^{-\beta} \wedge D_*\big) \cdot n^{-1/2} + \sup_{y \in \mathcal{M}} \left\{ \left| \mathfrak{B}_n^{(\lambda)}(y; x) \right| : d\big(y, \varphi_{\nu^*}^{(\lambda)}(x)\big) < 2^{k+1} n^{-\beta} \wedge D_* \right\}. \quad \because \text{(39)}
\end{aligned}
$$

$$\tag{44}$$

Therefore, we obtain that for each $k \geq K$,

$$
\begin{aligned}
\mathfrak{C}_{n,k} &= P\left( \Big( 2^k n^{-\beta} \leq \Delta(x) < 2^{k+1} n^{-\beta} \wedge D_* \Big) \cap \mathfrak{F}_{n,k} \right) \\
&\leq P\left( \Big( \Delta(x)^\alpha \geq \big(2^k n^{-\beta}\big)^\alpha \Big) \cap \mathfrak{F}_{n,k} \right) \\
&\leq \frac{C_1^{(\lambda)} \cdot \big(2^{k+1} n^{-\beta} \wedge D_*\big) \cdot n^{-1/2} + \mathbb{E}\left[ \sup_{y \in \mathcal{M}} \left\{ \left| \mathfrak{B}_n^{(\lambda)}(y; x) \right| : d\big(y, \varphi_{\nu^*}^{(\lambda)}(x)\big) < 2^{k+1} n^{-\beta} \wedge D_* \right\} \right]}{C_{\mathbf{g}} \cdot \big(2^k n^{-\beta}\big)^\alpha} \\
&\qquad\qquad\qquad\qquad\qquad\qquad\qquad \because \text{(44) \& Markov's inequality} \\
&\leq \frac{\big(C_1^{(\lambda)} + C_2^{(\lambda)}\big) \cdot \big(2^{k+1} n^{-\beta} \wedge D_*\big) \cdot n^{-1/2}}{C_{\mathbf{g}} \cdot \big(2^k n^{-\beta}\big)^\alpha} \qquad\qquad \because \text{(40)} \qquad\qquad\qquad \text{(45)}
\end{aligned}
$$

*Step 3-C: Concluding Step 3.* Combining (41), (43), and (45), we have

$$
P\Big( \Delta(x) > 2^K n^{-\beta} \Big) \leq \frac{2\big(C_1^{(\lambda)} + C_2^{(\lambda)}\big)}{C_{\mathbf{g}}} n^{-\frac{1}{2} + \beta(\alpha-1)} \sum_{k=K}^{\infty} 2^{-k(\alpha-1)}
$$

$$
+ \underbrace{P\Big( \Delta(x) \geq D_* \Big)}_{=o(1) \ \because \text{Step 1 of this proof}} + \sum_{k=K}^{\infty} P\Big( \mathfrak{G}_{n,k} \Big).
$$

Moreover, $\mathfrak{G}_{n,k}$ are mutually exclusive, and thus,

$$
\sum_{k=K}^{\infty} P\Big( \mathfrak{G}_{n,k} \Big) = P\left( \bigcup_{k=K}^{\infty} \mathfrak{G}_{n,k} \right) = P\left( \left( \bigcup_{k=K}^{\infty} \mathfrak{E}_n^{(\lambda)}\big( 2^k n^{-\beta}, 2^{k+1} n^{-\beta} \wedge D_*; x \big) \right)^c \right) \to 0 \quad \because \text{(39)}
$$

Finally, we obtain the desired result by letting $\beta = \frac{1}{2(\alpha-1)}$.

$$\square$$

## D  Proof of Theorem 3

In this section, we prove Theorem 4 that establishes an upper bound on $d\big(\varphi_{\widehat{\mathcal{D}}_n}^{(\lambda)}(x), \varphi_{\mathcal{D}_n}^{(\lambda)}(x)\big)$. This section is organized as follows. Firstly, in Section D.1, we present several useful results from matrix perturbation theory as lemmas. Next, in Section D.2, we provide a key lemma (Lemma 6) that establishes the stability of the weight function when there is covariate noise. Lastly, in Section D.3, we state and prove Theorem 4, from which Theorem 3 can be easily derived.

### D.1 Useful lemmas

**Definition 7.** *Let $n, p \in \mathbb{N}$ and let $M \in \mathbb{R}^{n \times p}$. The* row projection matrix *for $M$, denoted by $\Pi_M^{\text{row}} \in \mathbb{R}^{p \times p}$, is a matrix such that*

$$\Pi_M^{\text{row}} := M^\dagger \cdot M. \tag{46}$$

*and the* column projection matrix *for $M$, denoted by $\Pi_M^{\text{col}} \in \mathbb{R}^{n \times n}$, is a matrix such that*

$$\Pi_M^{\text{col}} := M \cdot M^\dagger. \tag{47}$$

We recall from (6) that for any $\lambda \in \mathbb{R}_+$, the singular value thresholding (SVT) operator $\text{SVT}^{(\lambda)}$ is defined such that

$$M = \sum_{i=1}^{\min\{n,p\}} s_i \cdot u_i v_i^\top \text{ is a SVD} \qquad \mapsto \qquad \text{SVT}^{(\lambda)}(M) = \sum_{i=1}^{\min\{n,p\}} s_i \cdot \mathbb{1}\{s_i > \lambda\} \cdot u_i v_i^\top.$$

In the rest of this section, we let $M^{(\lambda)} := \text{SVT}^{(\lambda)}(M)$ for shorthand.

**Lemma 3** (Properties of the row/column projection matrices). *Let $n, p \in \mathbb{N}$, and $M \in \mathbb{R}^{n \times p}$. For any $\lambda \in \mathbb{R}_+$, the following statements are true.*

1. $\Pi_{M^{(\lambda)}}^{\text{row}}$ *defines a projection in $\mathbb{R}^p$ and* $\text{rank } \Pi_{M^{(\lambda)}}^{\text{row}} = \text{rank } M^{(\lambda)}$.

2. $\Pi_{M^{(\lambda)}}^{\text{col}}$ *defines a projection in $\mathbb{R}^n$ and* $\text{rank } \Pi_{M^{(\lambda)}}^{\text{col}} = \text{rank } M^{(\lambda)}$.

3. $M \Pi_{M^{(\lambda)}}^{\text{row}} M^\dagger = \Pi_{M^{(\lambda)}}^{\text{col}}$ *and* $M^\dagger \Pi_{M^{(\lambda)}}^{\text{col}} M = \Pi_{M^{(\lambda)}}^{\text{row}}$.

*Proof.* Let $r = \text{rank } M$ and consider a compact singular value decomposition (SVD) of $M$:

$$M = \sum_{i=1}^{r} s_i \cdot u_i v_i^\top$$

where $s_1, \ldots, s_r$ are non-zero singular values of $M$. Noticing that

$$M^{(\lambda)} = \text{SVT}^{(\lambda)}(M) = \sum_{i=1}^{r} \mathbb{1}\{s_i > \lambda\} \cdot u_i v_i^\top$$

and that $M^\dagger = \sum_{i=1}^{r} s_i^{-1} \cdot v_i u_i^\top$, the three conclusions of the lemma follow straightforwardly from the orthonormality of singular vectors.

- $\Pi_{M^{(\lambda)}}^{\text{row}} = \sum_{i=1}^{r} v_i v_i^\top \cdot \mathbb{1}\{s_i > \lambda\}$ is the projection onto the row space of $M^{(\lambda)}$.

- $\Pi_{M^{(\lambda)}}^{\text{col}} = \sum_{i=1}^{r} u_i u_i^\top \cdot \mathbb{1}\{s_i > \lambda\}$ is the projection onto the column space of $M^{(\lambda)}$.

- Due to the orthonormality of singular vectors,

$$M \Pi_{M^{(\lambda)}}^{\text{row}} M^\dagger = \left( \sum_{i=1}^{r} s_i \cdot u_i v_i^\top \right) \left( \sum_{i=1}^{r} v_i v_i^\top \cdot \mathbb{1}\{s_i > \lambda\} \right) \left( \sum_{i=1}^{r} s_i^{-1} \cdot v_i u_i^\top \right)$$

$$= \sum_{i=1}^{r} u_i u_i^\top \cdot \mathbb{1}\{s_i > \lambda\}$$

$$= \Pi_{M^{(\lambda)}}^{\text{col}},$$

and likewise, $M^\dagger \Pi_{M^{(\lambda)}}^{\text{col}} M = \Pi_{M^{(\lambda)}}^{\text{row}}$.

$\square$

In addition, we collect two classical results from matrix perturbation theory and state them as lemmas.

**Lemma 4** ([50, Theorem 3.2]). *Let $\boldsymbol{X}, \boldsymbol{Z} \in \mathbb{R}^{n\times p}$. Then the following equation is true:*

$$\boldsymbol{Z}^\dagger - \boldsymbol{X}^\dagger = -\boldsymbol{Z}^\dagger \Pi_{\boldsymbol{Z}}^{\mathrm{col}}(\boldsymbol{Z} - \boldsymbol{X})\Pi_{\boldsymbol{X}}^{\mathrm{row}}\boldsymbol{X}^\dagger + \boldsymbol{Z}^\dagger \Pi_{\boldsymbol{Z}}^{\mathrm{col}}\Pi_{\boldsymbol{X}}^{\mathrm{col}\perp} - \Pi_{\boldsymbol{Z}}^{\mathrm{row}\perp}\Pi_{\boldsymbol{X}}^{\mathrm{row}}\boldsymbol{X}^\dagger \tag{48}$$

*where $\Pi_{\boldsymbol{X}}^{\mathrm{col}\perp} = \boldsymbol{I}_n - \Pi_{\boldsymbol{X}}^{\mathrm{col}}$ and $\Pi_{\boldsymbol{Z}}^{\mathrm{row}\perp} = \boldsymbol{I}_p - \Pi_{\boldsymbol{Z}}^{\mathrm{row}}$.*

**Lemma 5** ([15, Theorems 2.4 & 2.5]). *Let $\boldsymbol{X}, \boldsymbol{Z} \in \mathbb{R}^{n\times p}$. Then*

$$\left\|\Pi_{\boldsymbol{Z}}^{\mathrm{col}} - \Pi_{\boldsymbol{X}}^{\mathrm{col}}\right\| \leq \max\left\{\left\|(\boldsymbol{Z} - \boldsymbol{X})\boldsymbol{X}^\dagger\right\|, \left\|(\boldsymbol{Z} - \boldsymbol{X})\boldsymbol{Z}^\dagger\right\|\right\}. \tag{49}$$

*Moreover, if* $\operatorname{rank}\boldsymbol{X} = \operatorname{rank}\boldsymbol{Z}$, *then*

$$\left\|\Pi_{\boldsymbol{Z}}^{\mathrm{col}} - \Pi_{\boldsymbol{X}}^{\mathrm{col}}\right\| \leq \min\left\{\left\|(\boldsymbol{Z} - \boldsymbol{X})\boldsymbol{X}^\dagger\right\|, \left\|(\boldsymbol{Z} - \boldsymbol{X})\boldsymbol{Z}^\dagger\right\|\right\}. \tag{50}$$

## D.2 Stability of the weights under (small) perturbation in covariates

Let $\mathcal{D}_n = \{(x_i, y_i) \in \mathbb{R}^p \times \mathcal{M} : i \in [n]\}$ and $\widetilde{\mathcal{D}}_n = \{(z_i, y_i) \in \mathbb{R}^p \times \mathcal{M} : i \in [n]\}$ be two sets in $\mathbb{R}^p \times \mathcal{M}$. We may identify these sets with their empirical distributions. Recall the definition of $w_\nu^{(\lambda)}$ from (9): for any probability measure $\nu$ on $\mathbb{R}^p \times \mathcal{M}$, any $\lambda \in \mathbb{R}_+$, and any $x, x' \in \mathbb{R}^p$,

$$w_\nu^{(\lambda)}(x', x) = 1 + (x' - \mu_\nu)^\top \left[\mathtt{SVT}^{(\lambda)}(\Sigma_\nu)\right]^\dagger (x - \mu_\nu)$$

where $\mu_\nu = \mathbb{E}_{(X,Y)\sim\nu}(X)$ and $\Sigma_\nu = \operatorname{Var}_{(X,Y)\sim\nu}(X)$, cf. (7). We define the *weight vectors* induced by $\mathcal{D}_n$ and $\widetilde{\mathcal{D}}_n$ as follows: for any $\lambda \in \mathbb{R}_+$ and any $x \in \mathbb{R}^p$,

$$\begin{aligned}
\vec{w}_{\mathcal{D}_n}^{(\lambda)}(x) &:= \left[w_{\mathcal{D}_n}^{(\lambda)}(x_1, x) \quad \cdots \quad w_{\mathcal{D}_n}^{(\lambda)}(x_n, x)\right] \in \mathbb{R}^n, \\
\vec{w}_{\widetilde{\mathcal{D}}_n}^{(\lambda)}(x) &:= \left[w_{\widetilde{\mathcal{D}}_n}^{(\lambda)}(z_1, x) \quad \cdots \quad w_{\widetilde{\mathcal{D}}_n}^{(\lambda)}(z_n, x)\right] \in \mathbb{R}^n.
\end{aligned} \tag{51}$$

**Lemma 6** (Stability of weights). *Let $\mathcal{D}_n = \{(x_i, y_i) \in \mathbb{R}^p \times \mathcal{M} : i \in [n]\}$ and $\widetilde{\mathcal{D}}_n = \{(z_i, y_i) \in \mathbb{R}^p \times \mathcal{M} : i \in [n]\}$. Let $\boldsymbol{X} = [x_1 \ \cdots \ x_n]^\top \in \mathbb{R}^{n\times p}$ and $\boldsymbol{Z} = [z_1 \ \cdots \ z_n]^\top \in \mathbb{R}^{n\times p}$. For any $\lambda \in \mathbb{R}_+$, if $x \in \mathbb{R}^p$ satisfies $x - \mu_{\mathcal{D}_n} \in \operatorname{rowsp}(\boldsymbol{X}_{\mathrm{ctr}})$, then*

$$\left\|\vec{w}_{\widetilde{\mathcal{D}}_n}^{(\lambda)}(x) - \vec{w}_{\mathcal{D}_n}^{(\lambda)}(x)\right\| \leq \frac{\sqrt{n} \cdot \|\boldsymbol{Z} - \boldsymbol{X}\|}{\min\left\{\sigma^{(\lambda)}(\boldsymbol{X}_{\mathrm{ctr}}),\ \sigma^{(\lambda)}(\boldsymbol{Z}_{\mathrm{ctr}})\right\}} \cdot \left(2 \cdot \left\|x - \mu_{\mathcal{D}_n}\right\|_{\Sigma_{\mathcal{D}_n}} + 1\right) \tag{52}$$

*where $\boldsymbol{X}_{\mathrm{ctr}} = \left(\boldsymbol{I}_n - \frac{1}{n}\mathbf{1}_n\mathbf{1}_n^\top\right)\boldsymbol{X}$ and $\sigma^{(\lambda)}(\boldsymbol{X}) := \inf\{\sigma_i(\boldsymbol{X}) > \lambda : i \in \mathbb{N}\}$ (likewise for $\boldsymbol{Z}$).*

*Proof of Lemma 6.* This proof consists of three steps. In Step 1, we express the weight discrepancy $\vec{w}_{\widetilde{\mathcal{D}}_n}^{(\lambda)}(x) - \vec{w}_{\mathcal{D}_n}^{(\lambda)}(x)$ as a sum of matrix products using projections. In Step 2, we establish upper bounds on the norm of the expression obtained in Step 1. In Step 3, we collect intermediate results together and conclude the proof.

**Step 1: Decomposition of the weight discrepancy.** First of all, we rewrite $\vec{w}_{\widetilde{\mathcal{D}}_n}^{(\lambda)}(x) - \vec{w}_{\mathcal{D}_n}^{(\lambda)}(x)$ in a compact matrix representation that is presented in (60) at the end of this step. To this end, we begin by observing that

$$\mu_{\mathcal{D}_n} = \frac{1}{n}\boldsymbol{X}^\top\mathbf{1}_n, \qquad \text{and} \qquad \Sigma_{\mathcal{D}_n} = \frac{1}{n}\left(\boldsymbol{X} - \mathbf{1}_n\mu_{\mathcal{D}_n}^\top\right)^\top\left(\boldsymbol{X} - \mathbf{1}_n\mu_{\mathcal{D}_n}^\top\right) = \frac{1}{n}\boldsymbol{X}_{\mathrm{ctr}}^\top\boldsymbol{X}_{\mathrm{ctr}}. \tag{53}$$

For given $\lambda \in \mathbb{R}_+$, we let $\boldsymbol{X}_{\mathrm{ctr}}^{(\lambda)} := \mathtt{SVT}^{(\lambda)}(\boldsymbol{X}_{\mathrm{ctr}})$, and observe that

$$\Sigma_{\mathcal{D}_n}^{(\lambda)} = \Pi_{\boldsymbol{X}_{\mathrm{ctr}}^{(\lambda)}}^{\mathrm{row}} \cdot \left(\frac{1}{n}\boldsymbol{X}_{\mathrm{ctr}}^\top\boldsymbol{X}_{\mathrm{ctr}}\right) \cdot \Pi_{\boldsymbol{X}_{\mathrm{ctr}}^{(\lambda)}}^{\mathrm{row}} = \frac{1}{n} \cdot \boldsymbol{X}_{\mathrm{ctr}}^{(\lambda)\top} \cdot \boldsymbol{X}_{\mathrm{ctr}}^{(\lambda)}. \tag{54}$$

Then it follows that

$$\left[\Sigma_{\mathcal{D}_n}^{(\lambda)}\right]^\dagger = n \cdot \left[\boldsymbol{X}_{\mathrm{ctr}}^{(\lambda)\top} \cdot \boldsymbol{X}_{\mathrm{ctr}}^{(\lambda)}\right]^\dagger = n \cdot \left[\boldsymbol{X}_{\mathrm{ctr}}^{(\lambda)}\right]^\dagger \cdot \left[\boldsymbol{X}_{\mathrm{ctr}}^{(\lambda)\top}\right]^\dagger = n \cdot \Pi_{\boldsymbol{X}_{\mathrm{ctr}}^{(\lambda)}}^{\mathrm{row}} \cdot \boldsymbol{X}_{\mathrm{ctr}}^\dagger \cdot \left(\boldsymbol{X}_{\mathrm{ctr}}^\top\right)^\dagger \cdot \Pi_{\boldsymbol{X}_{\mathrm{ctr}}^{(\lambda)}}^{\mathrm{row}}.$$

Therefore, we have

$$\vec{w}_{\mathcal{D}_n}^{(\lambda)}(x) = \mathbf{1}_n + \left( \boldsymbol{X} - \mathbf{1}_n \mu_{\mathcal{D}_n}^\top \right) \cdot \left[ \Sigma_{\mathcal{D}_n}^{(\lambda)} \right]^\dagger \cdot (x - \mu_{\mathcal{D}_n})$$

$$= \mathbf{1}_n + n \cdot \boldsymbol{X}_{\mathrm{ctr}} \cdot \Pi_{\boldsymbol{X}_{\mathrm{ctr}}^{(\lambda)}}^{\mathrm{row}} \cdot \boldsymbol{X}_{\mathrm{ctr}}^\dagger \cdot \left( \boldsymbol{X}_{\mathrm{ctr}}^\top \right)^\dagger \cdot \Pi_{\boldsymbol{X}_{\mathrm{ctr}}^{(\lambda)}}^{\mathrm{row}} \cdot (x - \mu_{\mathcal{D}_n})$$

$$= \mathbf{1}_n + n \cdot \Pi_{\boldsymbol{X}_{\mathrm{ctr}}^{(\lambda)}}^{\mathrm{col}} \cdot \left( \boldsymbol{X}_{\mathrm{ctr}}^\top \right)^\dagger \cdot \Pi_{\boldsymbol{X}_{\mathrm{ctr}}^{(\lambda)}}^{\mathrm{row}} \cdot (x - \mu_{\mathcal{D}_n}), \tag{55}$$

where the equality in the last line follows from Lemma 3: $\boldsymbol{X}_{\mathrm{ctr}} \Pi_{\boldsymbol{X}_{\mathrm{ctr}}^{(\lambda)}}^{\mathrm{row}} \boldsymbol{X}_{\mathrm{ctr}}^\dagger = \Pi_{\boldsymbol{X}_{\mathrm{ctr}}^{(\lambda)}}^{\mathrm{col}}$.

Likewise, we repeat the above for $\widetilde{\mathcal{D}}_n$ and $\boldsymbol{Z}$ to write

$$\mu_{\widetilde{\mathcal{D}}_n} = \frac{1}{n} \boldsymbol{Z}^\top \mathbf{1}_n \qquad \text{and} \qquad \Sigma_{\widetilde{\mathcal{D}}_n} = \frac{1}{n} \boldsymbol{Z}_{\mathrm{ctr}}^\top \boldsymbol{Z}_{\mathrm{ctr}}.$$

Then, we obtain an expression for $\vec{w}_{\widetilde{\mathcal{D}}_n}^{(\lambda)}(x)$ in a similar form to (55), namely,

$$\vec{w}_{\widetilde{\mathcal{D}}_n}^{(\lambda)}(x) = \mathbf{1}_n + n \cdot \Pi_{\boldsymbol{Z}_{\mathrm{ctr}}^{(\lambda)}}^{\mathrm{col}} \cdot \left( \boldsymbol{Z}_{\mathrm{ctr}}^\top \right)^\dagger \cdot \Pi_{\boldsymbol{Z}_{\mathrm{ctr}}^{(\lambda)}}^{\mathrm{row}} \cdot (x - \mu_{\widetilde{\mathcal{D}}_n}). \tag{56}$$

Thereafter, we define $c_x, \tilde{c}_x \in \mathbb{R}^{n \times 1}$ so that

$$c_x = \|x - \mu_{\mathcal{D}_n}\|_{\Sigma_{\mathcal{D}_n}} = \left( \frac{1}{\sqrt{n}} \boldsymbol{X}_{\mathrm{ctr}}^\top \right)^\dagger \cdot (x - \mu_{\mathcal{D}_n}) \qquad \text{and}$$

$$\tilde{c}_x = \|x - \mu_{\mathcal{D}_n}\|_{\Sigma_{\widetilde{\mathcal{D}}_n}} = \left( \frac{1}{\sqrt{n}} \boldsymbol{Z}_{\mathrm{ctr}}^\top \right)^\dagger \cdot \left( x - \mu_{\widetilde{\mathcal{D}}_n} \right). \tag{57}$$

Then we observe that for any $x \in \mathbb{R}^p$,

$$n \cdot \Pi_{\boldsymbol{X}_{\mathrm{ctr}}^{(\lambda)}}^{\mathrm{row}} \cdot (x - \mu_{\mathcal{D}_n}) = n \cdot \Pi_{\boldsymbol{X}_{\mathrm{ctr}}^{(\lambda)}}^{\mathrm{row}} \cdot \frac{1}{\sqrt{n}} \boldsymbol{X}_{\mathrm{ctr}}^\top \cdot \left( \frac{1}{\sqrt{n}} \boldsymbol{X}_{\mathrm{ctr}}^\top \right)^\dagger \cdot (x - \mu_{\mathcal{D}_n}) = \sqrt{n} \cdot \Pi_{\boldsymbol{X}_{\mathrm{ctr}}^{(\lambda)}}^{\mathrm{row}} \cdot \boldsymbol{X}_{\mathrm{ctr}}^\top \cdot c_x. \tag{58}$$

Likewise,

$$n \cdot \Pi_{\boldsymbol{Z}_{\mathrm{ctr}}^{(\lambda)}}^{\mathrm{row}} \cdot \left( x - \mu_{\widetilde{\mathcal{D}}_n} \right) = n \cdot \Pi_{\boldsymbol{Z}_{\mathrm{ctr}}^{(\lambda)}}^{\mathrm{row}} \cdot \frac{1}{\sqrt{n}} \boldsymbol{Z}_{\mathrm{ctr}}^\top \cdot \left( \frac{1}{\sqrt{n}} \boldsymbol{Z}_{\mathrm{ctr}}^\top \right)^\dagger \cdot (x - \mu_{\widetilde{\mathcal{D}}_n}) = \sqrt{n} \cdot \Pi_{\boldsymbol{Z}_{\mathrm{ctr}}^{(\lambda)}}^{\mathrm{row}} \cdot \boldsymbol{Z}_{\mathrm{ctr}}^\top \cdot \tilde{c}_x. \tag{59}$$

Consequently, for any $x \in \mathbb{R}^p$, we obtain from (55) and (56) with aid of (58) and (59) that

$$\vec{w}_{\widetilde{\mathcal{D}}_n}^{(\lambda)}(x) - \vec{w}_{\mathcal{D}_n}^{(\lambda)}(x)$$

$$= \sqrt{n} \cdot \Pi_{\boldsymbol{Z}_{\mathrm{ctr}}^{(\lambda)}}^{\mathrm{col}} \cdot \left( \boldsymbol{Z}_{\mathrm{ctr}}^\top \right)^\dagger \cdot \Pi_{\boldsymbol{Z}_{\mathrm{ctr}}^{(\lambda)}}^{\mathrm{row}} \cdot \boldsymbol{Z}_{\mathrm{ctr}}^\top \cdot \tilde{c}_x - \sqrt{n} \cdot \Pi_{\boldsymbol{X}_{\mathrm{ctr}}^{(\lambda)}}^{\mathrm{col}} \cdot \left( \boldsymbol{X}_{\mathrm{ctr}}^\top \right)^\dagger \cdot \Pi_{\boldsymbol{X}_{\mathrm{ctr}}^{(\lambda)}}^{\mathrm{row}} \cdot \boldsymbol{X}_{\mathrm{ctr}}^\top \cdot c_x$$

$$= \sqrt{n} \cdot \Pi_{\boldsymbol{Z}_{\mathrm{ctr}}^{(\lambda)}}^{\mathrm{col}} \cdot \tilde{c}_x - \sqrt{n} \cdot \Pi_{\boldsymbol{X}_{\mathrm{ctr}}^{(\lambda)}}^{\mathrm{col}} \cdot c_x \qquad\qquad \because \text{Lemma 3}$$

$$= \sqrt{n} \cdot \Pi_{\boldsymbol{Z}_{\mathrm{ctr}}^{(\lambda)}}^{\mathrm{col}} \cdot (\tilde{c}_x - c_x) + \sqrt{n} \cdot \left( \Pi_{\boldsymbol{Z}_{\mathrm{ctr}}^{(\lambda)}}^{\mathrm{col}} - \Pi_{\boldsymbol{X}_{\mathrm{ctr}}^{(\lambda)}}^{\mathrm{col}} \right) \cdot c_x. \tag{60}$$

By triangle inequality, we obtain the following upper bound:

$$\left\| \vec{w}_{\widetilde{\mathcal{D}}_n}^{(\lambda)}(x) - \vec{w}_{\mathcal{D}_n}^{(\lambda)}(x) \right\| \leq \sqrt{n} \cdot \left\| \Pi_{\boldsymbol{Z}_{\mathrm{ctr}}^{(\lambda)}}^{\mathrm{col}} \cdot (\tilde{c}_x - c_x) \right\| + \sqrt{n} \cdot \left\| \left( \Pi_{\boldsymbol{Z}_{\mathrm{ctr}}^{(\lambda)}}^{\mathrm{col}} - \Pi_{\boldsymbol{X}_{\mathrm{ctr}}^{(\lambda)}}^{\mathrm{col}} \right) \cdot c_x \right\|. \tag{61}$$

**Step 2: Upper bounding the norm.** Next, we establish separate upper bounds for the two terms in (61).

**(1) The first term in** (61). First of all, we observe from the definition of $c_x$ and $\tilde{c}_x$, cf. (57), that

$$\tilde{c}_x - c_x = \left( \frac{1}{\sqrt{n}} \boldsymbol{Z}_{\mathrm{ctr}}^\top \right)^\dagger \cdot \left( x - \mu_{\widetilde{\mathcal{D}}_n} \right) - \left( \frac{1}{\sqrt{n}} \boldsymbol{X}_{\mathrm{ctr}}^\top \right)^\dagger \cdot (x - \mu_{\mathcal{D}_n})$$

$$= \sqrt{n} \cdot \left( \boldsymbol{Z}_{\mathrm{ctr}}^{\top \, \dagger} - \boldsymbol{X}_{\mathrm{ctr}}^{\top \, \dagger} \right) \cdot (x - \mu_{\mathcal{D}_n}) + \sqrt{n} \cdot \left[ \boldsymbol{Z}_{\mathrm{ctr}}^\top \right]^\dagger \cdot \left( \mu_{\widetilde{\mathcal{D}}_n} - \mu_{\mathcal{D}_n} \right).$$

Then we can upper bound the first term in (61) as follows:

$$\left\| \Pi_{\boldsymbol{Z}_{\mathrm{ctr}}^{(\lambda)}}^{\mathrm{col}} \cdot (\tilde{c}_x - c_x) \right\| = \sqrt{n} \cdot \left\| \Pi_{\boldsymbol{Z}_{\mathrm{ctr}}^{(\lambda)}}^{\mathrm{col}} \cdot \left( \boldsymbol{Z}_{\mathrm{ctr}}^{\top \, \dagger} - \boldsymbol{X}_{\mathrm{ctr}}^{\top \, \dagger} \right) \cdot (x - \mu_{\mathcal{D}_n}) \right\| + \sqrt{n} \cdot \left\| \Pi_{\boldsymbol{Z}_{\mathrm{ctr}}^{(\lambda)}}^{\mathrm{col}} \cdot \left[ \boldsymbol{Z}_{\mathrm{ctr}}^{\top} \right]^{\dagger} \cdot \left( \mu_{\widetilde{\mathcal{D}}_n} - \mu_{\mathcal{D}_n} \right) \right\|.$$
(62)

Next, we consider the orthogonal decomposition of $x - \mu_{\mathcal{D}_n}$:

$$x - \mu_{\mathcal{D}_n} = \Pi_{\boldsymbol{X}_{\mathrm{ctr}}}^{\mathrm{row}} (x - \mu_{\mathcal{D}_n}) + \Pi_{\boldsymbol{X}_{\mathrm{ctr}}}^{\mathrm{row} \, \perp} \cdot (x - \mu_{\mathcal{D}_n}) = \frac{1}{\sqrt{n}} \boldsymbol{X}_{\mathrm{ctr}}^{\top} \cdot c_x + \Pi_{\boldsymbol{X}_{\mathrm{ctr}}}^{\mathrm{row} \, \perp} \cdot (x - \mu_{\mathcal{D}_n}). \quad (63)$$

If $x - \mu_{\mathcal{D}_n} \in \mathrm{rowsp}\,(\boldsymbol{X}_{\mathrm{ctr}})$, then we obtain the following upper bound for the first term in (62):

$$\sqrt{n} \cdot \left\| \Pi_{\boldsymbol{Z}_{\mathrm{ctr}}^{(\lambda)}}^{\mathrm{col}} \cdot \left( \boldsymbol{Z}_{\mathrm{ctr}}^{\top \, \dagger} - \boldsymbol{X}_{\mathrm{ctr}}^{\top \, \dagger} \right) \cdot (x - \mu_{\mathcal{D}_n}) \right\|$$

$$\leq \left\| \Pi_{\boldsymbol{Z}_{\mathrm{ctr}}^{(\lambda)}}^{\mathrm{col}} \cdot \left( \boldsymbol{Z}_{\mathrm{ctr}}^{\top \, \dagger} - \boldsymbol{X}_{\mathrm{ctr}}^{\top \, \dagger} \right) \cdot \boldsymbol{X}_{\mathrm{ctr}}^{\top} \cdot c_x \right\|$$

$$\qquad + \sqrt{n} \cdot \left\| \Pi_{\boldsymbol{Z}_{\mathrm{ctr}}^{(\lambda)}}^{\mathrm{col}} \cdot \left( \boldsymbol{Z}_{\mathrm{ctr}}^{\top \, \dagger} - \boldsymbol{X}_{\mathrm{ctr}}^{\top \, \dagger} \right) \cdot \underbrace{\Pi_{\boldsymbol{X}_{\mathrm{ctr}}}^{\mathrm{row} \, \perp} \cdot (x - \mu_{\mathcal{D}_n})}_{=0} \right\| \qquad \because (63)$$

$$\leq \left\| \Pi_{\boldsymbol{Z}_{\mathrm{ctr}}^{(\lambda)}}^{\mathrm{col}} \cdot \left\{ -\boldsymbol{Z}_{\mathrm{ctr}}^{\top \, \dagger} \cdot \Pi_{\boldsymbol{Z}_{\mathrm{ctr}}}^{\mathrm{row}} \cdot \left( \boldsymbol{Z}_{\mathrm{ctr}}^{\top} - \boldsymbol{X}_{\mathrm{ctr}}^{\top} \right) \cdot \Pi_{\boldsymbol{X}_{\mathrm{ctr}}}^{\mathrm{col}} \cdot \boldsymbol{X}_{\mathrm{ctr}}^{\top \, \dagger} \right\} \cdot \boldsymbol{X}_{\mathrm{ctr}}^{\top} \cdot c_x \right\| \qquad \because \text{Lemma 4}$$

$$\leq \left\| \left[ \boldsymbol{Z}_{\mathrm{ctr}}^{(\lambda) \top} \right]^{\dagger} \right\| \cdot \left\| \Pi_{\boldsymbol{Z}_{\mathrm{ctr}}}^{\mathrm{row}} \cdot (\boldsymbol{Z} - \boldsymbol{X})^{\top} \cdot \Pi_{\boldsymbol{1}_n^{\perp}}^{\mathrm{row}} \cdot \Pi_{\boldsymbol{X}_{\mathrm{ctr}}}^{\mathrm{col}} \right\| \cdot \|c_x\|$$

$$\leq \frac{\|\boldsymbol{Z} - \boldsymbol{X}\|}{\sigma^{(\lambda)}\,(\boldsymbol{Z}_{\mathrm{ctr}})} \cdot \|c_x\|.$$

Similarly, the second term in (62) can be bounded by

$$\sqrt{n} \cdot \left\| \Pi_{\boldsymbol{Z}_{\mathrm{ctr}}^{(\lambda)}}^{\mathrm{col}} \cdot \left[ \boldsymbol{Z}_{\mathrm{ctr}}^{\top} \right]^{\dagger} \cdot \left( \mu_{\widetilde{\mathcal{D}}_n} - \mu_{\mathcal{D}_n} \right) \right\| \leq \frac{1}{\sqrt{n}} \cdot \left\| \left[ \boldsymbol{Z}_{\mathrm{ctr}}^{(\lambda) \top} \right]^{\dagger} \right\| \cdot \left\| \boldsymbol{1}_n^{\top} \cdot (\boldsymbol{Z} - \boldsymbol{X}) \right\|$$

$$\leq \frac{1}{\sqrt{n}} \cdot \frac{\|\boldsymbol{Z} - \boldsymbol{X}\|}{\sigma^{(\lambda)}\,(\boldsymbol{Z}_{\mathrm{ctr}})} \cdot \|\boldsymbol{1}_n\|.$$

All in all, we obtain

$$\sqrt{n} \cdot \left\| \Pi_{\boldsymbol{Z}_{\mathrm{ctr}}^{(\lambda)}}^{\mathrm{col}} \cdot (\tilde{c}_x - c_x) \right\| \leq \frac{\|\boldsymbol{Z} - \boldsymbol{X}\|}{\sigma^{(\lambda)}\,(\boldsymbol{Z}_{\mathrm{ctr}})} \cdot \left( \sqrt{n} \cdot \|c_x\| + \|\boldsymbol{1}_n\| \right) \qquad (64)$$

**(2) The second term in** (61). Letting $\boldsymbol{E}^{(\lambda)} := \boldsymbol{Z}_{\mathrm{ctr}}^{(\lambda)} - \boldsymbol{X}_{\mathrm{ctr}}^{(\lambda)}$, we observe that

$$\left\| \Pi_{\boldsymbol{Z}_{\mathrm{ctr}}^{(\lambda)}}^{\mathrm{col}} - \Pi_{\boldsymbol{X}_{\mathrm{ctr}}^{(\lambda)}}^{\mathrm{col}} \right\| \leq \max \left\{ \left\| \boldsymbol{E}^{(\lambda)} \cdot \boldsymbol{X}_{\mathrm{ctr}}^{(\lambda) \dagger} \right\|, \ \left\| \boldsymbol{E}^{(\lambda)} \cdot \boldsymbol{Z}_{\mathrm{ctr}}^{(\lambda) \dagger} \right\| \right\} \qquad \because \text{Lemma 5}$$

$$\leq \left\| \boldsymbol{E}^{(\lambda)} \right\| \cdot \max \left\{ \left\| \boldsymbol{X}_{\mathrm{ctr}}^{(\lambda) \dagger} \right\|, \ \left\| \boldsymbol{Z}_{\mathrm{ctr}}^{(\lambda) \dagger} \right\| \right\}$$

$$\leq \frac{\|\boldsymbol{Z} - \boldsymbol{X}\|}{\min \left\{ \sigma^{(\lambda)}(\boldsymbol{X}_{\mathrm{ctr}}), \ \sigma^{(\lambda)}(\boldsymbol{Z}_{\mathrm{ctr}}) \right\}}.$$

All in all, we obtain the following upper bound:

$$\sqrt{n} \left\| \left( \Pi_{\boldsymbol{Z}_{\mathrm{ctr}}^{(\lambda)}}^{\mathrm{col}} - \Pi_{\boldsymbol{X}_{\mathrm{ctr}}^{(\lambda)}}^{\mathrm{col}} \right) \cdot c_x \right\| \leq \left\| \Pi_{\boldsymbol{Z}_{\mathrm{ctr}}^{(\lambda)}}^{\mathrm{col}} - \Pi_{\boldsymbol{X}_{\mathrm{ctr}}^{(\lambda)}}^{\mathrm{col}} \right\| \cdot \|c_x\| \leq \frac{\|\boldsymbol{Z} - \boldsymbol{X}\|}{\min \left\{ \sigma^{(\lambda)}(\boldsymbol{X}_{\mathrm{ctr}}), \ \sigma^{(\lambda)}(\boldsymbol{Z}_{\mathrm{ctr}}) \right\}} \cdot \sqrt{n} \cdot \|c_x\|.$$
(65)

**Step 3: Concluding the proof.** We conclude this proof by inserting the upper bounds (64) and (65) from Step 2 into the upper bound (61) in Step 1. Specifically, we obtain

$$\left\| \vec{w}_{\widetilde{\mathcal{D}}_n}^{(\lambda)}(x) - \vec{w}_{\mathcal{D}_n}^{(\lambda)}(x) \right\| \leq \frac{\|\boldsymbol{Z} - \boldsymbol{X}\|}{\sigma^{(\lambda)}\,(\boldsymbol{Z}_{\mathrm{ctr}})} \cdot \left( \sqrt{n} \cdot \|c_x\| + \|\boldsymbol{1}_n\| \right) + \frac{\|\boldsymbol{Z} - \boldsymbol{X}\|}{\min \left\{ \sigma^{(\lambda)}(\boldsymbol{X}_{\mathrm{ctr}}), \ \sigma^{(\lambda)}(\boldsymbol{Z}_{\mathrm{ctr}}) \right\}} \cdot \sqrt{n} \cdot \|c_x\|$$

$$\leq \frac{\|\boldsymbol{Z} - \boldsymbol{X}\|}{\min \left\{ \sigma^{(\lambda)}(\boldsymbol{X}_{\mathrm{ctr}}), \ \sigma^{(\lambda)}(\boldsymbol{Z}_{\mathrm{ctr}}) \right\}} \cdot \left( 2\sqrt{n} \cdot \|c_x\| + \|\boldsymbol{1}_n\| \right).$$

Lastly, we note that $\|c_x\| = \sqrt{(x - \mu_{\mathcal{D}_n})^{\top} \Sigma_{\mathcal{D}_n}^{\dagger}\,(x - \mu_{\mathcal{D}_n})} = \left\| x - \mu_{\mathcal{D}_n} \right\|_{\Sigma_{\mathcal{D}_n}}$ and $\|\boldsymbol{1}_n\| = \sqrt{n}$. $\quad \square$

## D.3 Completing the proof of Theorem 3

Recall that given a set $\mathcal{D}_n = \{(x_i, y_i) : i \in [n]\}$, we let $\boldsymbol{X}_{\mathcal{D}_n} := [x_1 \ \cdots \ x_n]^\top \in \mathbb{R}^{n \times p}$. In addition, we let

$$\forall y \in \mathcal{M}, \ \vec{d}_{\mathcal{D}_n}^2(y) := \begin{bmatrix} d^2(y_1, y) & \cdots & d^2(y_n, y) \end{bmatrix} \in \mathbb{R}^n. \tag{66}$$

Recall that we let $\boldsymbol{X} = \boldsymbol{X}_{\mathcal{D}_n}$ and $\boldsymbol{Z} = \boldsymbol{X}_{\widetilde{\mathcal{D}}_n}$ for shorthand, and further, we let $\boldsymbol{X}_{\text{ctr}} = \left(\boldsymbol{I}_n - \frac{1}{n}\boldsymbol{1}_n\boldsymbol{1}_n^\top\right)\boldsymbol{X}$ and $\boldsymbol{Z}_{\text{ctr}} = \left(\boldsymbol{I}_n - \frac{1}{n}\boldsymbol{1}_n\boldsymbol{1}_n^\top\right)\boldsymbol{Z}$ denote the 'row-centered' matrices. Here we present and prove the complete version of Theorem 3.

**Theorem 4** (De-noising covariates)**.** *Suppose that Assumptions (C0) and (C1) hold. For any $\lambda \in \mathbb{R}_+$, if $x \in \mu_{\mathcal{D}_n} + \text{rowsp}\,\boldsymbol{X}_{\text{ctr}}$ and*

$$\|x - \mu_{\mathcal{D}_n}\|_{\Sigma_{\mathcal{D}_n}} \le \frac{1}{2}\left(\frac{C_g \cdot D_g^\alpha}{2\,\text{diam}\,(\mathcal{M})} \cdot \frac{\min\left\{\sigma^{(\lambda)}(\boldsymbol{X}_{\text{ctr}}),\ \sigma^{(\lambda)}(\boldsymbol{Z}_{\text{ctr}})\right\}}{\|\boldsymbol{Z} - \boldsymbol{X}\|} - 1\right), \tag{67}$$

*then*

$$d\left(\varphi_{\widetilde{\mathcal{D}}_n}^{(\lambda)}(x), \varphi_{\mathcal{D}_n}^{(\lambda)}(x)\right)$$

$$\le \left(\frac{\|\boldsymbol{Z} - \boldsymbol{X}\|}{\min\left\{\sigma^{(\lambda)}(\boldsymbol{X}_{\text{ctr}}),\ \sigma^{(\lambda)}(\boldsymbol{Z}_{\text{ctr}})\right\}} \cdot \frac{2 \cdot \|x - \mu_{\mathcal{D}_n}\|_{\Sigma_{\mathcal{D}_n}} + 1}{C_g} \cdot \frac{\left\|\vec{d}_{\mathcal{D}_n}^2(\tilde{\varphi}_n)\right\| + \left\|\vec{d}_{\mathcal{D}_n}^2(\varphi_n)\right\|}{\sqrt{n}}\right)^{\frac{1}{\alpha}}. \tag{68}$$

*Proof of Theorem 4.* First of all, we recall from (51) that

$$\vec{w}_{\mathcal{D}_n}^{(\lambda)}(x) = \begin{bmatrix} w_{\mathcal{D}_n}^{(\lambda)}(x_1, x) & \cdots & w_{\mathcal{D}_n}^{(\lambda)}(x_n, x) \end{bmatrix} \qquad \text{and} \qquad \vec{w}_{\widetilde{\mathcal{D}}_n}^{(\lambda)}(x) = \begin{bmatrix} w_{\widetilde{\mathcal{D}}_n}^{(\lambda)}(z_1, x) & \cdots & w_{\widetilde{\mathcal{D}}_n}^{(\lambda)}(z_n, x) \end{bmatrix}.$$

In addition, recall that we let for any $y \in \mathcal{M}$,

$$\vec{d}_{\mathcal{D}_n}^2(y) = \begin{bmatrix} d^2(y_1, y) & \cdots & d^2(y_n, y) \end{bmatrix} \in \mathbb{R}^n.$$

Thereafter, we observe that for any $y \in \mathcal{M}$ and any $x \in \left(\mu_{\mathcal{D}_n} + \text{rowsp}\,\boldsymbol{X}_{\text{ctr}}\right)$,

$$\left|R_{\widetilde{\mathcal{D}}_n}^{(\lambda)}(y; x) - R_{\mathcal{D}_n}^{(\lambda)}(y; x)\right| = \frac{1}{n}\left|\sum_{i=1}^n \left(w_{\widetilde{\mathcal{D}}_n}^{(\lambda)}(z_i, x) - w_{\mathcal{D}_n}^{(\lambda)}(x_i, x)\right) \cdot d^2(y_i, y)\right|$$

$$= \frac{1}{n}\left|\left\langle \vec{w}_{\widetilde{\mathcal{D}}_n}^{(\lambda)}(x) - \vec{w}_{\mathcal{D}_n}^{(\lambda)}(x),\ \vec{d}_{\mathcal{D}_n}^2(y)\right\rangle\right|$$

$$\overset{(a)}{\le} \frac{1}{n}\left\|\vec{w}_{\widetilde{\mathcal{D}}_n}^{(\lambda)}(x) - \vec{w}_{\mathcal{D}_n}^{(\lambda)}(x)\right\| \cdot \left\|\vec{d}_{\mathcal{D}_n}^2(y)\right\|$$

$$\overset{(b)}{\le} \frac{\|\boldsymbol{Z} - \boldsymbol{X}\|}{\min\left\{\sigma^{(\lambda)}(\boldsymbol{X}_{\text{ctr}}),\ \sigma^{(\lambda)}(\boldsymbol{Z}_{\text{ctr}})\right\}} \cdot \left(2 \cdot \|x - \mu_{\mathcal{D}_n}\|_{\Sigma_{\mathcal{D}_n}} + 1\right) \cdot \frac{\left\|\vec{d}_{\mathcal{D}_n}^2(y)\right\|}{\sqrt{n}} \tag{69}$$

where (a) is due to Cauchy-Schwarz inequality, and (b) follows from Lemma 6.

Using shorthand notation $R_n = R_{\mathcal{D}_n}^{(\lambda)}$, $\tilde{R}_n = R_{\widetilde{\mathcal{D}}_n}^{(\lambda)}$, $\varphi_n = \varphi_{\mathcal{D}_n}^{(\lambda)}(x)$, and $\tilde{\varphi}_n = \varphi_{\widetilde{\mathcal{D}}_n}^{(\lambda)}(x)$, we observe that

$$R_n(\tilde{\varphi}_n) - R_n(\varphi_n)$$

$$= R_n(\tilde{\varphi}_n) - \tilde{R}_n(\tilde{\varphi}_n) + \tilde{R}_n(\tilde{\varphi}_n) - R_n(\varphi_n)$$

$$\overset{(a)}{\le} R_n(\tilde{\varphi}_n) - \tilde{R}_n(\tilde{\varphi}_n) + \tilde{R}_n(\varphi_n) - R_n(\varphi_n)$$

$$\overset{(b)}{\le} \frac{\|\boldsymbol{Z} - \boldsymbol{X}\|}{\min\left\{\sigma^{(\lambda)}(\boldsymbol{X}_{\text{ctr}}),\ \sigma^{(\lambda)}(\boldsymbol{Z}_{\text{ctr}})\right\}} \cdot \left(2 \cdot \|x - \mu_{\mathcal{D}_n}\|_{\Sigma_{\mathcal{D}_n}} + 1\right) \cdot \frac{\left\|\vec{d}_{\mathcal{D}_n}^2(\tilde{\varphi}_n)\right\| + \left\|\vec{d}_{\mathcal{D}_n}^2(\varphi_n)\right\|}{\sqrt{n}} \tag{70}$$

where (a) follows from the optimality of $\tilde{\varphi}_n$, i.e., $\tilde{R}_n(\varphi_n) \geq \tilde{R}_n(\tilde{\varphi}_n)$, and (b) is due to (69).

Finally, we note that if

$$\|x - \mu_{\mathcal{D}_n}\|_{\Sigma_{\mathcal{D}_n}} \leq \frac{1}{2}\left(\frac{C_{\mathbf{g}} \cdot D_{\mathbf{g}}^{\alpha}}{2\operatorname{diam}(\mathcal{M})} \cdot \frac{\min\left\{\sigma^{(\lambda)}(\boldsymbol{X}_{\mathrm{ctr}}), \, \sigma^{(\lambda)}(\boldsymbol{Z}_{\mathrm{ctr}})\right\}}{\|\boldsymbol{Z} - \boldsymbol{X}\|} - 1\right),$$

then the upper bound in (70) certifies that $R_n(\tilde{\varphi}_n) - R_n(\varphi_n) < C_{\mathbf{g}} \cdot D_{\mathbf{g}}^{\alpha}$. Thus, we can use Assumption (C1) to convert the risk bound (70) to derive a distance bound between the minimizers:

$$d(\tilde{\varphi}_n, \varphi_n) \leq \left(\frac{R_n(\tilde{\varphi}_n) - R_n(\varphi_n)}{C_{\mathbf{g}}}\right)^{\frac{1}{\alpha}},$$

which completes the proof. $\qquad\square$

# E   Further details on the experiments

**Experimental setup in Section 5.** We consider combinations of $p \in \{150, 300, 600\}$ and $n \in \{100, 200, 400\}$. The datasets $\mathcal{D}_n = \{(X_i, Y_i) : i \in [n]\}$ and $\widetilde{\mathcal{D}}_n = \{(Z_i, Y_i) : i \in [n]\}$ are generated as follows.

(True covariate $X$) Let $X_i \sim \mathcal{N}_p(\mathbf{0}_p, \Sigma)$ be IID multivariate Gaussian with mean $\mathbf{0}_p$ and covariance $\Sigma$ such that $\operatorname{spec}(\Sigma) = \{\kappa_j > 0 : j \in [p]\}$ is an exponentially decreasing sequence such that $\operatorname{tr}(\Sigma) = \sum_{j=1}^{p} \kappa_j = p$. To be specific, for each $p$, we consider an exponentially decreasing sequence $1 = a_1 > \cdots > a_p = 10^{-3}$, and then set $\kappa_j = p \cdot a_j / (\sum_{j'=1} a_{j'})$ for each $j \in [p]$. Note that $\sum_{j=1}^{\lfloor p/3 \rfloor} \kappa_j / \sum_{j'=1}^{p} \kappa_{j'} \approx 0.9$, and thus, $\Sigma$ is effectively low-rank.

(Noisy covariate $Z$) For the error-prone covariate $Z = X + \varepsilon$, we consider two scenarios $\varepsilon_j \overset{IID}{\sim} \mathcal{N}(0, \sigma_\varepsilon^2)$ and $\varepsilon_j \overset{IID}{\sim} \operatorname{Laplace}(0, \sigma_\varepsilon)$. Note that in this setting, we have the signal-to-noise ratio $\mathbb{E}(\|X\|_2^2)/\mathbb{E}(\|\varepsilon\|_2^2) = 1/\sigma_\varepsilon^2$. We set $\sigma_\varepsilon^2 = 0.05^2$.

(Response $Y$) Given $X = x$, let $Y$ be the distribution function of $\mathcal{N}(\mu_{\alpha,\beta}(x) + \eta, \tau^2)$, where

- $\mu_{\alpha,\beta}(x) = \alpha + \beta^\top x$ with $\alpha = 1$ and $\beta = p^{-1/2} \cdot \mathbf{1}_p$,
- $\eta \sim \mathcal{N}(0, \sigma_\eta^2)$,
- $\tau^2 \sim \mathcal{IG}(s_1, s_2)$, an inverse gamma distribution with shape $s_1$ and scale $s_2$.

We note that $\mathbb{E}(\tau^2) = \frac{s_2}{s_1 - 1}$ and $\operatorname{Var}(\tau^2) = \frac{s_2^2}{(s_1-1)^2(s_1-2)}$. In particular, when $\tau^2 = 0$, this setting corresponds to the classical linear regression model for scalar responses. We set $\sigma_\eta^2 = 0.5^2$, and $(s_1, s_2) = (18, 17)$. In this setting, we have

- $\mathbb{E}(\mu_{\alpha,\beta}(X)) = 1$ and $\operatorname{Var}(\mu_{\alpha,\beta}(X)) = \beta^\top \Sigma \beta \approx 1$,
- $\mathbb{E}(\tau^2) = 1$ and $\operatorname{Var}(\tau^2) = 0.25^2$.

(Tuning parameter $\lambda$) For simplicity, we chose a universal threshold value as

$$\hat{\lambda}_n = \arg\min_{\lambda \in \Lambda} \operatorname{MSPE}(\varphi_{\widetilde{\mathcal{D}}_n}^{(\lambda)}),$$

where $\Lambda$ is a fine grid on $(0, \sqrt{\lambda_1 \cdot p/n})$. Then the same threshold $\hat{\lambda}_n$ was used to evaluate $\operatorname{Bias}^2(\varphi_{\widetilde{\mathcal{D}}^{(b)}}^{(\lambda)}(x))$, $\operatorname{Var}(\varphi_{\widetilde{\mathcal{D}}^{(b)}}^{(\lambda)}(x))$, and $\operatorname{MSE}(\varphi_{\widetilde{\mathcal{D}}^{(b)}}^{(\lambda)}(x))$ for all $b = 1, \ldots, B$. Therefore, we claim that the performance of the SVT estimator reported in Table 1 has further room for improvement if one substitute $\hat{\lambda}_n^{(b)} = \arg\min_{\lambda \in \Lambda} \operatorname{MSPE}(\varphi_{\nu^{(b)}}^{(\lambda)})$ for each Monte Carlo experiment. Although suboptimal results are reported, we note that the proposed SVT outperforms both the oracle estimator and the naive EIV estimator in our simulation study. In practice, one may employ cross-validation for better performance. For the MSPE in Table 1, we reported $\min_{\lambda \in \Lambda} \operatorname{MSPE}(\varphi_{\widetilde{\mathcal{D}}_n}^{(\lambda)})$.

**Evaluation metrics: bias and variance.** We evaluate the accuracy and efficiency of the Fréchet regression function estimator using the bias and the variance. For any given $x$, we define

$$\mathrm{Bias}_x\big(\varphi_\nu^{(\lambda)}\big) := d_W\Big(\overline{\varphi}_\nu^{(\lambda)}(x),\, \varphi_{\nu*}^{(0)}(x)\Big) \quad \text{and} \quad \mathrm{Var}_x\big(\varphi_\nu^{(\lambda)}\big) := \frac{1}{B}\sum_{b=1}^{B} d_W\Big(\varphi_{\nu^{(b)}}^{(\lambda)}(x),\, \overline{\varphi}_\nu^{(\lambda)}(x)\Big)^2,$$

where $\nu \in \{\mathcal{D}_n, \widetilde{\mathcal{D}}_n\}$ and $\overline{\varphi}_\nu^{(\lambda)}(x) := \arg\min_y \sum_{b=1}^{B} d_W\big(\varphi_{\nu^{(b)}}^{(\lambda)}(x),\, y\big)^2$ is the Fréchet mean of $\varphi_{\nu^{(1)}}^{(\lambda)}(x), \ldots, \varphi_{\nu^{(B)}}^{(\lambda)}(x)$. Note that these definitions are a generalization of the standard bias and variance of the regression function estimator in Euclidean spaces. We evaluate the global performance of the estimator by considering a fixed set of evaluation points, $\mathcal{G}_M = \{x_m : m = 1, \ldots, M\}$, and compute

$$\mathrm{Bias}^2\big(\varphi_\nu^{(\lambda)}\big) := \frac{1}{M}\sum_{m=1}^{M} \mathrm{Bias}_{x_m}^2\big(\varphi_\nu^{(\lambda)}\big) \quad \text{and} \quad \mathrm{Var}\big(\varphi_\nu^{(\lambda)}\big) := \frac{1}{M}\sum_{m=1}^{M} \mathrm{Var}_{x_m}\big(\varphi_\nu^{(\lambda)}\big).$$

In our experiment, we generate the set $\mathcal{G}_M$ by drawing $x_1, \ldots, x_M$ IID from the same distribution as $X$, with $M = 500$.

**Additional experiment with linear regression models** We also conducted an additional experiment for the standard linear regression models with three different metric metrics.

(Model) The linear regression model for $(X, Y) \in \mathbb{R}^p \times \mathbb{R}^d$ is defined as $Y = \alpha + X\beta + \eta$ and the covariate is contaminated as $Z = X + \varepsilon$. We generate $X$ using the effective low-rank model with a geometrically decaying spectrum and condition number $10^3$ (see Appendix E in the original submission).

(Parameters) Here, we let $\alpha = \mathbf{1}_d + 0.1 \cdot \boldsymbol{g}$, $\beta = d^{-1/2} \cdot \mathbf{1}_{p\times d} + 0.1 \cdot \boldsymbol{G}$, $\eta \sim \mathcal{N}(0, 0.5^2 \cdot I_d)$ and $\varepsilon \sim \mathcal{N}(0, 0.5^2 \cdot I_p)$, with $\boldsymbol{g}$, $\boldsymbol{G}$ being standard Gaussians.

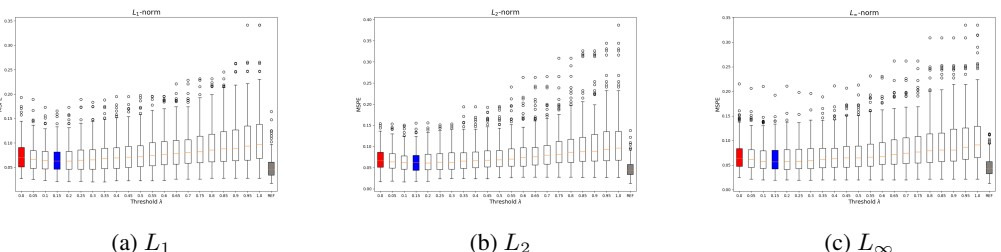

(a) $L_1$          (b) $L_2$          (c) $L_\infty$

Figure 3: Normalized mean squared prediction error (NMSPE) versus threshold $\lambda$ for vector-valued linear regression with three different metrics for $\mathcal{Y}$: (a) $\ell_1$-metric; (b) $\ell_2$-metric; and (c) $\ell_\infty$-metric. **[SVT]**: the regularized Fréchet regression estimator $\varphi_{\widetilde{\mathcal{D}}_n}^{(\lambda)}$ with best NMSPE; **[EIV]**: the unregularized Fréchet regression estimator $\varphi_{\widetilde{\mathcal{D}}_n}^{(0)}$ with errors-in-variables covariates $Z$; **[REF]**: the unregularized Fréchet regression estimator $\varphi_{\mathcal{D}_n}^{(0)}$ with error-free covariates $X$ (oracle).

