# OpenReview forum: "Errors-in-variables Fr\'echet Regression with Low-rank Covariate Approximation"
_NeurIPS.cc/2023/Conference — NeurIPS 2023 poster_

### Official Review · Reviewer_urUC · 2023-07-01

**Soundness:** 3 good
**Presentation:** 3 good
**Contribution:** 3 good
**Rating:** 5
**Confidence:** 4

**Summary:**

The paper attempts to extend regression models by combining error-in-variables in high-dimensional predictors and responses in a metric space. The main method is principal component regression whereby principal components are formed in the predictor space and serve as dimension-reduced predictors. It ties into existing approaches for global object regression and is a valid attempt at a very challenging problem but unfortunately none of the problems is solved: The predictors are not high-dimensional as their dimension is fixed; the errors-in-variables are not mitigated: While a bound is provided where the errors appear, the error-in-variables model is not shown to allow for consistent estimation and why principal components would be a reasonable approach at error mitigation is not elucidated; tuning parameter choice is not investigated.


**Strengths:**

This paper addresses two difficult topics, errors-in-variables and Frechet regression. Errors-in-variables is a major problem in regression settings and is an important issue for many applications. While much research has been done on errors-in-variables in standard regression settings, it remains a challenging issue and especially so when the responses are in metric spaces. Such new types of responses are also clearly relevant for modern data. An approach through principal component regression would be of interest if it leads to consistent estimation and is implemented in a data-adaptive way. So the topic of this paper is timely from several perspectives.



**Weaknesses:**

(1) The background is not well balanced. It emphasizes Frechet regression while it is very superficial about error-in-variables. Since principal component regression is used to address error-in-variables it is then of interest to report how PCR mitigates/addresses or is expected to mitigate error-in-variables even in standard regression setting.

(2) The proposed method does not lead to convergence for errors-in-variables according to (16), but this is what is needed for a method to succeed in this setting.

(3) There is no data-adaptive choice (along with theory justifying it) for the tuning parameter $\lambda$, however this parameter may be a crucial feature for principal component regression implementations as it determines the number of components to be included as predictors. It is not clear how it should be chosen to mitigate errors-in-variables or when responses are in metric space.


**Questions:**

Is the bound in (16) sharp?

High-dimensional predictors are mentioned but the dimension p is fixed throughout. Can you handle the case of large p where p would increase with sample size ?


**Limitations:**

See the previous comments.

---

> ### Author Rebuttal · Authors · 2023-08-09
>
> We value the insightful feedback and comments received. In response, we have prepared a comprehensive Author Rebuttal report and individual rebuttals to address the highlighted concerns and questions. Our response is structured to first address the highlighted weaknesses and then provide detailed point-by-point responses to the specific questions.
>
> ### **Weaknesses**
> #### 1\. Background of study:
> In this paper, we tackle EIV problems within the Fr'echet regression framework. While previous works have explored regression analysis in non-Euclidean metric spaces, addressing EIV issues in this context remains uncharted. This study aims to fill this gap, responding to real-world situations where latent measurement errors impact observed covariates.
>
> We adopt PCR as a concrete, practical solution to EIV models in non-Euclidean regression, driven by two compelling considerations. Firstly, the prevalence of (approximate) low-rank structures in real-world datasets enhances the practically relevance of our approach. Secondly, we intentionally opt for an approach with minimal assumptions regarding covariate errors to ensure broad applicability. Notably, PCR aligns well with these considerations, due to its inherent utilization of effective low-rank structures and its demonstrated capacity to alleviate covariate errors in the standard regression setting [2]. This has prompted our anticipation that PCR can similarly address EIV challenges in non-Euclidean settings.
>
> While there is an extensive EIV literature, conventional techniques often rely on knowledge of error distributions or require prior information to eliminate measurement errors. In response to the reviewer's insightful comment, we focus on "mitigating" (possibly non-stochastic) covariate perturbations by retaining only essential assumptions necessary for Fr'echet regression analysis, instead of attempting to completely "denoise" EIV effects that may depend on impractical information. Importantly, PCR's advantage lies in its error-mitigation capabilities without necessitating prior knowledge of error distributions. Our method demonstrates effectiveness with theoretical guarantees and practical applicability.
>
> We value the reviewer's input and acknowledge that these nuances may not have been fully conveyed in our initial presentation. In particular, we recognize the need to elaborate on how/why PCR is expected to mitigate/address EIV challenges. To address this, we plan to enrich the "Errors-in-variables-regression" and "Principal component regression" paragraphs in the Related Work, providing a more comprehensive overview of literature and our motivations in the upcoming camera-ready version after acceptance.
>
> #### 2\. Convergence/mitigation of EIV implied by Eq.(16):
> Here we reiterate our response in the Author Rebuttal report. Despite its generality, this upper bound highlights the effective error mitigation in specific scenarios. Consider the following:
>
> 1. **Well-balanced, effectively low-rank covariates:**
> Suppose that $X\in R^{n\times p}$ satisfies (1) $|X_{ij}|=\Omega(1)$ for all i,j, and (2) $\sigma_1(X)\asymp\sigma_r(X)\gg\sigma_{r+1}(X)\asymp\sigma_{n\wedge p} = O(1)$, where $r\ll n\wedge p$ denotes the effective rank of $X$. Then we have $\sigma_1(X)^2\asymp\sigma_r(X)^2\asymp\|X\|_F^2/r\gtrsim np/r$.
>
> 2. **Independent sub-gaussian noise:**
> Next, suppose that $Z=X+E$ where $E$ is a random matrix with independent sub-Gaussian rows. Then $\|Z-X\|\lesssim\sqrt{n}+\sqrt{p}$ with high probability due to concentration inequality.
>
> In the random design scenario where $X$ and $x$ have i.i.d. rows drawn from the same distribution, $\|x-\mu_{D_n}\|_{\Sigma}\approx 1$ with high probability. As a result, the upper bound in Eq.(16) is bounded by $\sqrt{r/p}+\sqrt{r/n}$, which diminishes to 0 when $r\ll n\wedge p$.
>
> #### 3\. Principled choice of $\lambda$:
> In our numerical study, we utilized a uniform search grid for $\lambda$ selection, opting for the value that minimizes the MSPE. Similarly, in practical scenarios, cross-validation can be employed to identify the optimal $\lambda$.
>
> Regarding the concern about lacking a data-adaptive principle (and theoretical justification), it's worth noting that the effective estimation of $X$ from $Z$ drives the error-mitigating capability, irrespective of $\mathcal{Y}$'s metric, under Assumptions (C0)-(C2). The low-rank estimation of $X$ via SVT (Eq. 6) is closely connected to nuclear-norm-regularized low-rank matrix estimation (hard SVT vs. soft SVT). We conjecture based on these observations that the well-studied principles for selecting $\lambda$ in low-rank matrix estimation could be extended with minor adjustments.
>
> ### **Questions**
> #### 1\. Sharpness of the bound in Eq.(16):
> The error bound in Eq.(16) is sharp (up to a multiplicative constant) due to the existence of a worst-case noise instance ($E=Z-X$) that achieves equality. This sharpness is inherited from the tightness of the subspace perturbation bound (Davis-Kahan/Wedin). While it might be feasible to achieve a more potent error-mitigating bound by adopting a stricter model assumption, we have refrained from doing so to maintain generality. Thus, Eq. (16) remains sharp within our framework.
>
> #### 2\. Handling high-dimensional case ($p>n$):
> Our approach, involving low-rank matrix estimation via SVT, effectively addresses the high-dimensional case where $p$ grows with $n$, under the condition that the (effective) rank of $X$ remains appreciably smaller than $n\wedge p$, ensuring control over $|Z-X|/\sigma_r(X)$ as seen in Eq. (16).
>
> Furthermore, we have supplemented our response with additional numerical simulation results in the Author Rebuttal report. These results underscore the superior performance of our proposed method compared to the naive Fr\'echet in EIV settings. Our method consistently achieves better predictive accuracy across diverse scenarios, including the cases with $p > n$.

---

> > ### Comment · Reviewer_urUC · 2023-08-14
> >
> > Thank you for your response. I have just reviewed the reference paper [2] regarding the role of the low-rank covariate matrix and the significance of large p in their work. Essentially, the findings presented in [2] support the validity of the principal component regression approach in your research. I am content with your explanations and have adjusted the score to 5 (borderline accept), as the results are accurate; however, the novelty beyond what is presented in [2] appears to be limited.

---

> > > ### Author Response · Authors · 2023-08-14
> > >
> > > Thank you once again for your thoughtful feedback on our research paper and for revisiting the reference paper [2] regarding the role of low-rank covariance matrix and the significance of large $p$. We also appreciate your diligence in re-evaluating the nuances of our contribution, and are pleased to learn that our explanations have met your satisfaction.
> > >
> > > Nonetheless, we wish to take this opportunity to address your concerns about the perceived limited novelty beyond [2]. In this regard, we aim to elucidate the unique aspects of our work, highlighting its novelty and significance.
> > >
> > > 1. **Addressing Errors-in-Variables Regression with Non-Euclidean Response Variables:**
> > > While our approach shares some similarities with Agarwal et al.'s [2] strategy for addressing errors-in-variables (EIV), it goes beyond these similarities by tackling the unique challenges posed by non-Euclidean regression settings. Unlike the standard linear regression scenario considered by Agarwal et al. [2], where a hypothesis relation $f: x \mapsto y$ can be explicitly represented as a regression coefficient vector $\beta \in R^P$, the distinctive nature of non-Euclidean response variables makes such a representation unattainable. The absence of an explicit form of the solution -- e.g., the global Fr\'echet regression function in Eq. (3) of the original submission -- profoundly limits the mathematical tools available for analyzing the theoretical properties of the Fr\'echet regression function. To the best of our knowledge, the analysis of Fr\'echet regression is only available through the generalized $M$-estimation theory, but EIV problems have not been addressed in the recent literature, including pioneering works such as [38], [41], and [46]. By effectively integrating Fr\'echet regression with covariate data cleansing (via SVT), our method adeptly handles a broad spectrum of non-Euclidean regression scenarios,  subject to the specified regularity assumptions (C0)-(C2). This wide-ranging applicability stands as a notable and innovative advancement beyond the scope of [2].
> > >
> > > 2. **Advancements in Theoretical Proof and Bounds:**
> > > While our proof of Theorem 3 shares common arguments and mathematical tools with [2], we emphasize our proof enhances clarity and precision. Notably, our analysis employs a cleaner and more straightforward argument, evident in the comparison between the proof of our Theorem 3 and that of Agarwal et al.'s Theorems 3.1 \& 3.2. Moreover, our approach eliminates dependency on exogenous factors, e.g., $\| \beta^* \|_1$ as seen in Eq. (5) of [2], providing a more intuitive understanding of prediction error control as shown in Eq. (16) of the original submission.
> > >
> > > Considering these elucidations, we believe that our research makes a substantial contribution towards addressing challenges in errors-in-variables regression with non-Euclidean response variables. We appreciate your recognition of the accuracy and validity of our results.
> > > We hear your perceived concerns on the novelty beyond [2], and we would like to assure you that the universal applicability of our method, coupled with the advancements in theoretical proofs, substantiates its significance.

---

### Official Review · Reviewer_BqGr · 2023-07-03

**Soundness:** 3 good
**Presentation:** 2 fair
**Contribution:** 3 good
**Rating:** 6
**Confidence:** 2

**Summary:**

This paper discusses a new method for Fréchet regression that relates non-Euclidean response variable to multivariate predictors. The proposed method leverages the low-rank structure of the covariate matrix to improve the efficiency and accuracy of the estimator, particularly in high-dimensional and errors-in-variables regression settings. The authors provide a theoretical analysis of the estimator's properties and demonstrate its superior performance through numerical experiments.

**Strengths:**

1. The authors propose the regularized global Fréchet regression framework that can effectively utilize the low-rank structure inherent in the covariate matrix.

2. The proposed method is straightforward to implement and works well in the errors-in variables regression setting.

3. The authors have provided both theoretical analysis and numerical experiments to demonstrate the effectiveness of the proposed method.





**Weaknesses:**

1. The paper looks interesting, but it does not appear to be significantly advancing the state of the art in the field, e.g. [38, 41]

2. In Theorem 2, it is not clear how $\lambda$ affects the convergence rate in general.

3. No discussions regarding the performance of the three algorithms are given after Table 1 in Section 5.


**Questions:**

1. In the numerical experiments, does the method still work for the high-dimensional case when $p>n$?

2. How to choose the regularization parameter $\lambda$ in practice?


**Limitations:**

The authors have adequately addressed the limitations.

---

> ### Author Rebuttal · Authors · 2023-08-09
>
> We appreciate your valuable feedback and comments on our work. With gratitude for the positive evaluation, we are committed to further elucidating our contributions by addressing the concerns and questions you have raised. To facilitate this process, our response is structured to first address highlighted weaknesses, followed by comprehensive point-by-point responses to your specific questions.
>
> ### **Weaknesses**
> #### 1\. Lack of significant advances:
> We acknowledge the pioneering nature of [38] and [41] in the realm of Fr\'echet regression, as highlighted in our Introduction section. Yet, it is crucial to underscore that our paper's primary focus is on addressing Errors-in-Variables (EIV) challenges within the Fr\'echet regression framework.
>
> While there have been commendable strides in statistical and machine learning literature, including notable works such as [10], [38], [41], [46], and [54], that delve into regression analysis for response variables within general metric spaces, we are not aware of any prior investigations into addressing EIV problems in the non-Euclidean regression setting. Our work bridges this gap adeptly, addressing the EIV challenge in the non-Euclidean regression context, relevant to real-world scenarios with latent measurement errors impacting observed covariates.
>
> Our paper also offers substantial theoretical advancements. An inherent challenge we tackled involves analyzing the asymptotic behavior of the $M$-estimator in the global Fr'echet regression model when coupled with the singular value thresholding (SVT) technique. This task is made considerably intricate due to the absence of an explicit form for the global Fr\'echet regression function in generic metric spaces, where it can be only identified pointwisely. Thus, our endeavor necessitates a detailed examination of risk differences involving weight function perturbations, influenced by the threshold $\lambda$ and covariate errors. As such, deriving the bias-variance decomposition in Theorem 2 is not straightforward; intricate technical development, exemplified by Lemmas 2 and 3, was essential. Additionally, the derivation of Theorem 3 diverges from conventional PCR analyses in [2] for similar reasons, highlighting our unique approach. While rooted in some proof techniques from [41] and [2], our work employs more sophisticated arguments to address intricate nuances and challenges.
>
> Considering these factors, we firmly assert that our work significantly advances this field.
>
> #### 2\. Effects of $\lambda$ in Theorem 2:
> As $\lambda$ increases, the bias (reflected on $b_{\lambda}$) grows while the variance diminishes. Our analysis captures the bias trend, although the variance aspect may be slightly unclear due to the asymptotic nature of our analysis. Refining our analysis into the finite-sample properties in future research could provide further insights.
>
> #### 3\. Missing discussions on numerical experiments:
> A thorough analysis of the three algorithms' performance is available in Appendix E. We've incorporated technical details like simulation settings, implementation specifics, evaluation metrics, and further result discussions in Appendix E due to page limitations. We acknowledge the reviewer's recommendation to succinctly describe these aspects in the Experiments section for clarity. In the camera-ready version post-acceptance, we will provide clear references to facilitate easy access to the technical details.
>
> ### **Questions**
> #### 1\. Numerical experiments for high-dimensional cases:
> We've performed extra simulations to reaffirm the consistency of our numerical results with those in the original manuscript. Specifically, we have extended our investigation to include nine additional combinations, with $n\in${100, 200, 400} and $p\in${150, 300, 600}, as presented in Table R.1 attached to the Author Rebuttal. These encompass high-dimensional scenarios, aiding readers in extrapolating numerical trends to even larger-scale experiments. For comprehensive information, please refer to our responses in the Author Rebuttal report.
>
> #### 2\. Choice of $\lambda$ in practice:
> In our numerical analysis, we utilized an evenly spaced search grid in the range $(0, \lambda_1 \sqrt{p/n})$, taking into account the maximum eigenvalue of the covariance matrix ($\lambda_1$), model complexity ($p$), and sample size ($n$). After evaluating mean squared prediction errors (MSPE) across this grid, we selected the parameter value that minimized MSPE.
>
> In practical contexts, cross-validation could be employed to estimate MSPE, and it is viable to use the sample covariance matrix of noisy covariates to approximate $\lambda_1$. Importantly, it is noteworthy that certain settings within the penalized regression literature provide theoretically-guided principles for selecting $\lambda$.

---

### Official Review · Reviewer_S37y · 2023-07-05

**Soundness:** 3 good
**Presentation:** 3 good
**Contribution:** 2 fair
**Rating:** 6
**Confidence:** 2

**Summary:**

This paper proposes a new method in Frechet regression of non-Euclidean response variables, with a particular focus on high-dimensional, errors-in-variables regression. The idea is to combine the original Frechet regression with Principle Component Regression (PCR). In this way, the low-rank structure in the matrix of (Euclidean) covariates is utilized by extracting its principal components via low-rank matrix approximation. Theoretical analysis of consistency, convergence etc of the proposed method have been provided. Numerical experiments on simulated datasets have also been presented.

**Strengths:**

(1) The proposed method tackles limitations of Frechet estimation such as reliance on ideal scenarios with abundant and noiseless
covariate data;

(2) The method utilizes the low-rank structure so that it can be applied to high-dimensional settings;

(3) both theoretical analysis and empirical results are demonstrated.

**Weaknesses:**

(1) underlines some terms and references which are rarely seen in papers;

(2) line 98: after "Frechet regression.", there is an additional phrase "in this study."

**Questions:**

N/A

**Limitations:**

no limitations provided

---

> ### Author Rebuttal · Authors · 2023-08-09
>
> We appreciate your feedback and positive evaluation. In response, we have prepared both a comprehensive Author Rebuttal report and individual rebuttal to address highlighted weaknesses and questions. We believe these efforts will strengthen our contributions while effectively addressing the concerns raised.
>
> ### **Regarding your specific points**
> #### 1\. Atypical underlines:
> Thank you for bringing this to our attention. We will ensure the removal of the underlines in the revision.
>
> #### 2\. Typo in Line 98:
> We are grateful for pointing out the typo. During the preparation of the camera-ready revision upon acceptance, we will meticulously review the manuscript and make necessary corrections, including those highlighted by the reviewer.

---

> > ### Comment · Reviewer_S37y · 2023-08-12
> > **keep the score as is**
> >
> > Thanks for the rebuttal and I will keep my score as is.

---

### Official Review · Reviewer_ZuE1 · 2023-07-07

**Soundness:** 3 good
**Presentation:** 3 good
**Contribution:** 3 good
**Rating:** 6
**Confidence:** 4

**Summary:**

The proposed method leverages the low-rank structure inherent in the covariate matrix to improve efficiency and accuracy. It combines global Fréchet regression with principal component regression to enable more effective modeling and estimation, especially in high-dimensional and errors-in-variables regression settings. The paper provides a theoretical analysis of the proposed estimator's properties in large samples, including bias, variance, and variations due to measurement errors. Empirical experiments support the theoretical findings, demonstrating the superior performance of the approach.

**Strengths:**

Strengths of this paper include:

The paper introduces a new framework, called regularized (global) Fréchet regression, which combines Fréchet regression and principal component regression. This framework effectively utilizes the low-rank structure in the covariate matrix by extracting principal components through low-rank matrix approximation.

The paper provides a thorough theoretical analysis with three main theorems. Firstly, it proves the consistency of the proposed estimator for the true global Fréchet regression model. Secondly, it investigates the convergence rate of the estimator's bias and variance. Lastly, it derives an upper bound for the distance between estimates obtained with error-free covariates and those with errors-in-variables covariates. These results establish the effectiveness of the proposed framework in addressing model mis-specification and achieving more efficient model estimation.

Numerical experiments conducted on simulated datasets validate the theoretical findings. The results demonstrate that the proposed method provides more accurate estimates of regression parameters, particularly in high-dimensional settings. The experiments highlight the importance of incorporating the low-rank structure of covariates in Fréchet regression and provide empirical evidence that aligns with the theoretical analysis.

**Weaknesses:**

The authors should have performed experiments in diverse settings using multiple distance metrics. It would have been interesting to see the relation of MSPE against threshold in different settings.

**Questions:**

What is the difference between MSE and MSPE, does MSPE stand for Mean Square Prediction Error? The abbreviations in the experiments section have not been explained properly like REV, EIV, SVT, etc. Line 147 can be rewritten as " .. conditional distribution of Y given X = x is normally distributed .. "

**Limitations:**

Yes.

---

> ### Author Rebuttal · Authors · 2023-08-08
>
> Thank you for the valuable feedback on our work. With gratitude for the positive evaluation, we are dedicated to clarifying and enhancing our contributions by addressing the raised concerns and questions. To facilitate this process, our response is organized to first tackle highlighted weaknesses, followed by a detailed response to the question.
>
> ### **Weaknesses**
> #### 1\. Experiments in more diverse settings:
> We value the reviewer's suggestion and have extended our numerical investigation to encompass a wider range of scenarios. This includes diverse problem parameters and the introduction of non-Gaussian noise (Laplacian) conditions for the Wasserstein space example. Furthermore, we have examined the standard linear regression model for Euclidean responses, along with variations involving different metrics ($\ell_1$ and $\ell_{\infty}$) within the response space $\mathcal{Y}=R^d$. We present a summary of these findings in Table R.1 and Figure R.1, included in the attachment to our Author Rebuttal.
>
> Specifically, Figure R.1 demonstrates consistent trends in mean squared prediction error (MSPE) across all metrics, resembling those observed in Figure 2 of our original submission. Notably, our proposed method (represented in blue) consistently outperforms the naive EIV approach (depicted in red) across all three metrics. It is important to highlight that while the $\ell_2$ metric enables an explicit linear regression model form, such an explicit form is not available for $\ell_1$ or $\ell_{\infty}$ metrics alike Fr\'echet regression for the Wasserstein space.
>
> ### **Questions**
> #### 1\. Acronyms / paraphrasing of Line 147:
> We acknowledge the reviewer's accurate observation regarding the MSPE acronym. The formal definition of Mean Squared Prediction Error can be found in Appendix E (Line 822). Due to page constraints, we deferred the definitions of evaluation metrics and acronyms, including Mean Squared Error (MSE) and Mean Squared Prediction Error (MSPE), to Appendix E of the original manuscript. Other technical details, such as simulation settings, implementation specifics, and evaluation metrics, are also provided in Appendix E.
>
> We appreciate the reviewer's suggestion and recognize the value of providing concise explanations of these terms within the Experiments section to enhance readers' comprehension. During the revision process for the camera-ready version upon acceptance, we will incorporate these explanations either in the caption of Table 1 or within the main text. Moreover, we will revise line 147, as per the suggestion, for improved clarity.

---

> > ### Comment · Reviewer_ZuE1 · 2023-08-14
> >
> > I thank the authors for the detailed rebuttal and clarifications. I will keep the positive score.

---

### Official Review · Reviewer_Jnac · 2023-07-25

**Soundness:** 2 fair
**Presentation:** 3 good
**Contribution:** 2 fair
**Rating:** 6
**Confidence:** 3

**Summary:**

The submission 5822 makes a step forward in exploring the Fréchet regression, which is a significant approach for non-Euclidean response variables. Compared to existing work, this research has specific focuses on 1) high-dimensional, 2) errors-in-variables settings, and designed  a novel framework that combines the Fréchet regression and the principal component regression. The analysis in paper 1) proved the consistency of the proposed estimator, 2) investigated the convergence rate, and 3) derived an upper bound for the distance between the the estimates from error-free and errors-in-variables covariates. Some numerical simulations on synthetic datasets are provided.



**Strengths:**


- this work is well-written and provides results that will I believe be of interest to the community. The idea to design such a regression scheme is novel to me. In particular, the framework and main theorems described in Sec. 3 and Sec. 4 are convincing and well presented. The literature (from what I know) is globally well discussed.

- claims and mathematical derivations seem to be sound and correct, and it clearly state its contributions, notation and results.

- also the R implementation is offered in supplement, which allows for reproducing the key results using the README file



**Weaknesses:**


- I understand this paper is a theory-based work. However, I still think the numerical simulations are too weak and might not be convincing.
  - the scale of the experiment is too small. The authors are suggested to consider larger $p$ and $n$, as well as more concrete examples/models as mentioned in Sec. 4.1
  - the synthesized data ($p<n$) in implementation is *not* high-dimensional, which is the main setting of this work
  - lack of experiments on real-world benchmarks, e.g., UCI, libsvm datasets

- maybe the motivations of this research should be reconsidered.
  - why the authors want to combine the *Fréchet regression* and *errors-in-variables setting*, along with the specific *PCR*? The authors only mentioned some advances in PCR research in line 96 and expressed their inspiration from this. Infact, there is whole lot of literature on high dimensional statistical learning and robust modeling. Why not borrow relevant ideas?
  - compared to the existing work of Fréchet  regression, what are the core challenges (difficulties) of this work, and did the authors draw on some existing frameworks / proof techniques from them?

**Questions:**


- in experiments, the Gaussian noise added to covariate may be too ideal. Although this assumption is common, could you provide results of other forms of noise (for example, some work on adversarial perturbations)?

- how would proposed theorems extend to concrete models/objectives? providing a practical guide is better for helping readers understand the contribution of this paper

- line 826: ''$\Lambda$ is a fine grid on ...''  adding more details about how to define the searching grid and the final chosen value of $\lambda$ are appreciated

- some lines have small typos, e.g., in line 523, it should be Proof of Proposition **4**

I'm not an expert of non-Euclidean regression analysis and would be happy to revise the rating if I missed some key aspects




**Limitations:**

- authors did not discuss in detail the limitations of the proposed regression framework (while they said "Yes" in checklist), but given that this work is based on theoretical analysis, I believe that all Assumptions in main body could clarify the technical limitations

- there's no need to discuss potential negative societal impact

---

> ### Author Rebuttal · Authors · 2023-08-08
>
> We value the insightful comments and feedback provided. Our response is organized to address the highlighted weaknesses first, followed by comprehensive point-by-point responses to the specific questions.
>
> ### **Weaknesses**
> #### 1\. Numerical simulations:
> *(A) Scale of experiments, high-dimensional settings, and more concrete examples:*
> We expanded simulations, incorporating results in the Author Rebuttal report; we added nine configurations in Table R.1, encompassing various dimensions ($n\in${100, 200, 400} and $p\in${150, 300, 600}) for easy extrapolation to larger-scale experiments.
>
> Additionally, we extended investigations, analyzing the proposed method's prediction performance in standard regression analysis with Euclidean responses (Example 1 in Sect. 4.1) and other metrics ($\ell_1$ and $\ell_{\infty}$) in $R^d$. Consistent MSPE trends were observed, akin to Figure 2 in the original submission; see Figure R.1 in the Author Rebuttal report.
>
> Lastly, our focus on the Wasserstein space (Example 2 in Sect. 4.1) as the primary simulation instance in the original submission aligns with growing interest in machine learning for random objects, enhancing insight into errors-in-variables challenges within related work.
>
> *(B) Absence of real-world benchmarks:*
> As the reviewer mentioned, this paper primarily focuses on establishing theoretical guarantees for employing singular value thresholding (SVT) in errors-in-variables Fr\'echet regression analysis for metric-space-valued responses, with a comprehensive analysis. While real-world benchmark experiments are absent, our mathematically rigorous analysis and numerical experiments on synthesized data consistently demonstrate SVT's superior finite-sample performance over the naive EIV estimator, even without leveraging prior knowledge of measurement error distributions.
>
> We acknowledge the reviewer's point and concur that investigating SVT's performance on real-world datasets offers exciting prospects for future research. Additionally, the inherent flexibility of our two-step EIV Fr\'echet regression framework (allowing covariate cleansing through means other than SVT) suggests potential for practical improvements, making these avenues enticing for further exploration.
>
> #### 2\. Reconsidering the motivations:
> *(A) Why PCR?*
> Indeed, there exists a diverse body of literature in high-dimensional learning and robust regression modeling, as acknowledged by the reviewer and elaborated upon in the Author Rebuttal. However, much of this literature assumes response spaces to be vector spaces or endowed with inner product structures. Additionally, prior statistical analyses of EIV problems often assume known or estimable noise distributions.
>
> In this paper, we propose a two-step approach involving covariate cleansing followed by followed by Fréchet regression. Specifically, we opt for PCR/SVT for the covariate cleansing step, among other methods, to address these challenges in scenarios when distributional knowledge is absent. We appreciate the reviewer's suggestion of exploring other ideas in the literature and agree that this could offer promising avenues for future research.
>
> *(B) Core challenges:*
> One of the core challenges in our work was analyzing the asymptotic behavior of the $M$-estimator in the global Fr\'echet regression model combined with SVT. Unlike settings with an algebraic structure on the response space, the global Fr\'echet regression model in generic metric spaces lacks an explicit form and is defined only pointwisely. Thus, controlling the risk difference involves scrutinizing perturbations of weight functions, influenced by the threshold $\lambda$ and covariate errors. Consequently, obtaining the bias-variance decomposition in Theorem 2 is not straightforward from existing works; it necessitated intricate technical development, as exemplified by Lemmas 2 and 3. The derivation of Theorem 3 also departs from conventional PCR analyses in [2] due to similar reasons. While drawing on some proof techniques in [41] and [2], we employed more sophisticated arguments to address these unique intricacies and challenges within our framework.
>
> ### **Questions**
> #### 1\. Non-Gaussian noise:
> Please refer to the Author Rebuttal and its attachment for extra experimental outcomes using Laplacian noise. It's important to note that our proposed method isn't explicitly designed for active robustness, yet it can withstand (sparse) adversarial noise if $|Z-X|$ is significantly smaller than the signal singular value, as indicated by Eq. (16).
>
> #### 2\. Illustration of Theorems with concrete models:
> In our paper, we used a basic linear regression model with scalar responses to offer an accessible illustration for readers unfamiliar with metric-spaced-valued responses. For instance, consider the linear regression $Y = \alpha + \beta X + \eta$, where $X$ is confined to a compact interval in $\mathbb{R}$ (page 4). Then, Theorem 2 demonstrates that bias asymptotically relies on eigenvalues below threshold $\lambda$, while variance follows a $\sqrt{n}$ rate. This extends standard principal component regression to metric-spaced-valued regression. Remark 1 on page 7 discusses the application of Theorem 2 in Examples 1, 2, and 3, and Theorem 3's implications are highlighted in the Author Rebuttal report.
>
> #### 3\. Details for searching grid:
> We used an 100 evenly spaced search grid in the interval $(0, \lambda_1 \sqrt{p/n})$, where $\lambda_1$ is the maximum eigenvalue of the covariance matrix. MSPE was calculated over this grid, and the value minimizing MSPE was selected. For instance, we obtained $\lambda=0.3584$ for n=100, p=50 in Figure 2.
>
> Due to page limits, full technical details, simulation settings, implementation, metrics, and result discussions are deferred to Appendix E. However, we recognize the value of summarizing these details in the Experiments section for reader clarity. Upon revision, we'll ensure a clear pointer to technical details.

---

> > ### Comment · Reviewer_Jnac · 2023-08-15
> >
> > Thanks for your response to each question and hope to see these clarifications in the final version. I have improved my rating to 6.

---

### Author Rebuttal · Authors · 2023-08-08

We appreciate the valuable feedback from the reviewers and their constructive comments. Here, we address common themes raised by the reviewers, providing clarity on our methodological and theoretical contributions. We also present supplementary numerical results to reinforce our findings. Detailed responses to each reviewer's comments can be found in dedicated individual rebuttals.

### **Summary of contributions**
We address the errors-in-variables (EIV) regression problem in a non-Euclidean response context through a composite approach, combining Fr\'echet regression (for non-Euclidean responses) and principal component regression (for covariate error mitigation).

Unlike conventional EIV literature that relies on distributional knowledge for covariate error ($\varepsilon_i$ in Eq.(5)), we avoid such assumptions, distinguishing our approach. Rather than aiming for complete elimination of covariate noise -- a challenging task without stringent distributional assumptions -- we focus on "mitigating" errors. This is achieved by estimating the design matrix through low-rank matrix estimation, particularly via principal component regression (PCR), which approximates the design matrix and alleviates errors. A similar approach was explored in standard linear regression setting [2].

The paper underscores efficient use of the design matrix's low-rank structure (covariates), even in arbitrary non-Euclidean metric spaces for the response variable.  Our "regularized" Fr\'echet regression estimator's superiority is supported by three theorems (consistency, convergence rate, and error reduction in variables) and also corroborated by numerical experiments.

### **More background & Rationale for PCR**
A vast literature addresses EIV models. In this work, we adopt PCR for a concrete, practical solution to EIV models in non-Euclidean regression, driven by two compelling reasons. Firstly, the prevalence of (approximate) low-rank structures in real-world datasets renders our approach practically relevant. Secondly, we purposefully choose an approach with minimal assumptions concerning covariate errors for broad applicability.

PCR aligns with these considerations, leveraging inherent low-rank structures effectively and demonstrating error-mitigating capabilities [2]. Notably, PCR stands apart from conventional EIV techniques by not requiring a priori knowledge of measurement error distributions. Moreover, PCR has an extensive presence in the high-dimensional statistics and dimensionality reduction literature.

We appreciate Reviewer 1 (Jnac) and Reviewer 5 (urUC) for their invaluable insights. Acknowledging that these nuances may not have been fully elucidated in our initial presentation, we plan to augment the "Errors-in-variables-regression" paragraph within the Related Work section in the upcoming camera-ready version after acceptance. This revision will offer a more comprehensive overview of high-dimensional and robust modeling, clarifying our motivations and methodology.

### **Implications of Theorem 3**
We refrain from imposing distributional assumptions on $\varepsilon=Z-X$, ensuring the broad applicability of inequality Eq.(16) across diverse scenarios. Also, this bound in Eq.(16) is sharp, exemplified by a worst-case noise instance that attains equality (up to a multiplicative constant).

Despite its generality, this upper bound highlights the effective error mitigation in specific scenarios. Consider the following:
1. *Well-balanced, effectively low-rank covariates:*
Suppose that $X\in R^{n\times p}$ satisfies (1) $|X_{ij}|=\Omega(1)$ for all i,j, and (2) $\sigma_1(X)\asymp\sigma_r(X)\gg\sigma_{r+1}(X)\asymp\sigma_{n\wedge p} = O(1)$, where $r\ll n\wedge p$ denotes the effective rank of $X$. Then we have $\sigma_1(X)^2\asymp\sigma_r(X)^2\asymp\|X\|_F^2/r\gtrsim np/r$.

2. *Independent sub-gaussian covariate noise:*
Next, suppose that $Z=X+E$ where $E$ is a random matrix with independent sub-Gaussian rows. Then $\|Z-X\|\lesssim\sqrt{n}+\sqrt{p}$ with high probability due to concentration inequality.

In the random design scenario where $X$ and $x$ have i.i.d. rows drawn from the same distribution, $\|x-\mu_{D_n}\|_{\Sigma}\approx 1$ with high probability. As a result, the upper bound in Eq.(16) is bounded by $\sqrt{r/p}+\sqrt{r/n}$, which diminishes to 0 when $r\ll n\wedge p$.

### **Additional numerical experiments**
Our extended numerical investigation encompasses non-Gaussian (Laplacian) covariate noise  and includes larger-scale experiments, including $p>n$ cases, enabling straightforward extrapolation to larger scenarios. The summarized results in the attachment (Table \ref{tab:sim-wasserstein-high-dim}) affirm the superiority of our proposed method (SVT), evident from consistently lower mean squared prediction errors (MSPE). This added experiment  further underscores SVT's ability to effectively address EIV challenges, enhancing prediction accuracy for EIV Fr\'echet regression, even without prior knowledge of measurement errors.

Significantly, SVT even surpasses error-free-covariate Fr\'echet regression (REF) in MSPE within this experiment. Standard Fr\'echet regression generally falters in high-dimensional settings due to issues like non-invertibility of sample covariance matrix (when $n<p$), or high covariate correlations. Our background study (not detailed here) identified the ill-posed nature of the M-estimation for REF, leading to instability and escalated mean squared errors (MSE). Hence, Table R.1 presents REF's finite-sample performance achieved through pseudo-inverse of the sample covariance matrix. Nevertheless, SVT proves a more dependable predictive model than even this stabilized version of REF.

Furthermore, we conducted experiments in linear regression settings ($\cal{Y}=R^d$) using three metrics: $\ell_1$, $\ell_2$, and $\ell_{\infty}$. Similar trends in MSPE, observed in Figure 2 of the original submission, persist across all these metrics; see Figure R.1 attached.

---

### Decision · Program_Chairs · 2023-09-21

**Decision:**

Accept (poster)

**Comment:**

All reviewers voted for acceptance. They (more or less) all argue that this work contains concepts and results that will be of interest to the community. Even the more critical reviewers acknowledge that during the rebuttal and discussion phase several critical remarks could be addressed in a meaningful way. I fully share this positive over-all impression of the paper, and for me the methodological strengths clearly outweigh possible weaknesses regarding the motivation and some design choices in the experiments. So after going again over the paper, all reviews an, the rebuttal and the discussions, I am convinced that a final version of this paper that takes into account the results of the discussions would make a nice contribution. Therefore, I recommend acceptance of this paper.